# Understanding In-Context Learning on Structured Manifolds: Bridging Attention to Kernel Methods

**Zhaiming Shen**[1]**, Alexander Hsu**[2]**, Rongjie Lai**[2]**, Wenjing Liao**[1]
[1]School of Mathematics, Georgia Institute of Technology
[2]Department of Mathematics, Purdue University
[1]{zshen49,wliao60}@gatech.edu
[2]{hsu297,lairj}@purdue.edu

## Abstract

While in-context learning (ICL) has achieved remarkable success in natural language and vision domains, its theoretical understanding—particularly in the context of structured geometric data—remains unexplored. This paper initiates a theoretical study of ICL for regression of Hölder functions on manifolds. We establish a novel connection between the attention mechanism and classical kernel methods, demonstrating that transformers effectively perform kernel-based prediction at a new query through its interaction with the prompt. This connection is validated by numerical experiments, revealing that the learned query–prompt scores for Hölder functions are highly correlated with the Gaussian kernel. Building on this insight, we derive generalization error bounds in terms of the prompt length and the number of training tasks. When a sufficient number of training tasks are observed, transformers give rise to the minimax regression rate of Hölder functions on manifolds, which scales exponentially with respect to the prompt length with the exponent depending on the intrinsic dimension of the manifold, rather than the ambient space dimension. Our result also characterizes how the generalization error scales with the number of training tasks, shedding light on the complexity of transformers as in-context kernel algorithm learners. Our findings provide foundational insights into the role of geometry in ICL and novels tools to study ICL of nonlinear models.

## 1 Introduction

The Transformer architecture, first introduced by Vaswani et al. (2017), has fundamentally reshaped machine learning, driving significant advancements in natural language processing (NLP), computer vision, and other domains. Unlike traditional feedforward and convolutional neural networks, transformers employ an attention mechanism that allows each token to interact with others and selectively aggregate information based on learned relevance scores. This mechanism enables more flexible and context-aware representation learning. Transformers now serve as the foundational architecture for large language and video generation models, such as GPT (Achiam et al., 2023), BERT (Devlin, 2018), SORA (Brooks et al., 2024) and their successors.

These empirical successes have demonstrated the in-context learning (ICL) capability of transformers, in which models can perform learning tasks by conditioning on a given set of examples, known as a prompt, provided at inference time, without any additional parameter updates Brown et al. (2020); Radford et al. (2019); Liu et al. (2023); Garg et al. (2022). The ICL phenomenon of transformers has also sparked substantial research interest in developing theoretical explanations of its underlying mechanisms. In Bai et al. (2023); Zhang et al. (2024); Von Oswald et al. (2023); Akyürek et al. (2022); Cole et al. (2024), transformers are proved for ICL of linear models, including least squares, ridge regression, Lasso, generalized linear models and linear inverse problems.

Beyond linear models, transformers are studied for nonlinear models in Yun et al. (2019); Takakura & Suzuki (2023); Gurevych et al. (2022); Havrilla & Liao (2024); Shen et al. (2025), with the goal of learning a single function, classifier, or sequence-to-sequence mapping. Specifically, Yun et al.

(2019) proved that transformer models can universally approximate continuous sequence-to-sequence functions with compact support, while the network size grows exponentially with respect to the sequence dimension. Takakura & Suzuki (2023) studied the approximation and estimation ability of transformers for sequence-to-sequence functions with anisotropic smoothness on infinite-dimensional inputs. Gurevych et al. (2022) studied binary classification with transformers when the posterior probability function exhibits a hierarchical composition model with Hölder smoothness. In Havrilla & Liao (2024); Shen et al. (2025), transformers are proved to leverage low-dimensional geometric structures of data (Havrilla & Liao, 2024) or machine learning tasks (Shen et al., 2025). While these works focus on a single learning task, ICL involves multiple learning tasks performed within the same model by leveraging prompts to adapt to each task on the fly, highlighting a form of task generalization without explicit retraining. It is worth noting that such task generalization also appear in the operator learning setting such as solving partial differential equations (Yang et al., 2023; Cole et al., 2024) and learning on the space of probability measures (Huang & Lai, 2025; Cole et al., 2026). While in-context operator learning has been explored with feedforward neural networks (Chiu et al., 2025), transformer-based in-context learning (ICL) has received increasing attention due to its strong empirical performance.

A theoretical understanding of ICL in transformers—especially in settings involving structured data with a geometric prior—remains limited and largely unexplored. In this work, we initiate a theoretical study of ICL for manifold regression. A manifold hypothesis is incorporated into our regression model to leverage low-dimensional geometric structures of data. Recent works have demonstrated that, under a manifold hypothesis of data, feedforward and convolutional residual networks give rise to a sample complexity depending on the intrinsic dimension (Shaham et al., 2018; Chen et al., 2022; 2019; Liu et al., 2021; Nakada & Imaizumi, 2020; Schmidt-Hieber, 2019). Empirical evidence has shown that the neural scaling laws of transformers depend on the intrinsic dimension of data (Kaplan et al., 2020; Sharma & Kaplan, 2022), while theoretical justifications, especially for ICL, are limited.

A central insight of this paper is the interpretation of transformers as learning kernel methods for function regression. Our study establishes a novel connection between the attention mechanism and classical kernel methods, showing that token interactions within attention can be interpreted as constructing an interaction kernel used to perform regression. Based on this connection, we construct a transformer neural network to exactly implement kernel regression, which builds an approximation theory of transformers for in-context manifold regression. To be more precise, let $\mathfrak{s} = \{\mathbf{x}_1, f(\mathbf{x}_1), \mathbf{x}_2, f(\mathbf{x}_2), \ldots, \mathbf{x}_n, f(\mathbf{x}_n); \mathbf{x}_{n+1}\}$ be a prompt, we explicitly construct a transformer $\mathrm{T}_h^*$ to exactly implement the kernel regression estimator $\mathcal{K}_h(\mathfrak{s})$ such that

$$\mathrm{T}_h^*(\mathfrak{s}) = \mathcal{K}_h(\mathfrak{s}) := \frac{\sum_{i=1}^n \exp\left(-\|\mathbf{x}_{n+1} - \mathbf{x}_i\|^2/h^2\right) f(\mathbf{x}_i)}{\sum_{i=1}^n \exp\left(-\|\mathbf{x}_{n+1} - \mathbf{x}_i\|^2/h^2\right)}, \tag{1}$$

where the Gaussian kernel of bandwidth $h > 0$ is used. Our construction shows that transformer-based ICL can implement kernel regression with *zero approximation error*. A formal statement can be found in Lemma 1. This perspective not only illuminates the internal workings of transformers in the in-context regression setting, but also motivates a theoretical framework for analyzing their generalization performance. Moreover, this connection is validated by numerical experiments on the regression of Hölder functions, revealing that the learned query–prompt scores in the last transformer layer are highly correlated with the Gaussian kernel.

Based on this key insight, our theoretical contribution for the generalization error of the transformer-based ICL can be summarized as follows: Let $\mathcal{M}$ be a $d$-dimensional compact Riemannian manifold in $\mathbb{R}^D$. We consider the ICL of $\alpha$-Hölder ($0 < \alpha \leq 1$) functions on $\mathcal{M}$ given a prompt of length $n$. During training, one observes the regression of $\Gamma$ functions/tasks, where each function/task is provided on a prompt of length $n$. At inference time, a prompt of length $n$ is given for a new $\alpha$-Hölder function on $\mathcal{M}$, and the goal is to predict the function value at a new input. Under this setting, we prove that the squared generalization error of transformer-based ICL is upper bounded by

$$C_1 \left(nD^3\Gamma^{-\frac{1}{2}}\sqrt{\log(nD\Gamma)}\right) + C_2 \left(n^{-\frac{2\alpha}{2\alpha+d}}[\log n]^{1+\frac{3d}{2}}\right), \tag{2}$$

with constants $C_1, C_2$. A formal statement of our result can be found in Theorem 1. Our result sheds light on theoretical understandings of transformer-based ICL in the following aspects:

• **Scaling Law of Transformers as Algorithm Learners.** The first error term in (2) characterizes the scaling law of transformers as in-context kernel algorithm learners. When a transformer is trained

on $\Gamma$ regression tasks, it can learn a kernal regression algorithm and generalize to a new task, with the generalization error given in the first term in (2).

• **Minimax Regression Error with a Prompt of Length** $n$**.** The second error term in (2) indicates the scaling law of transformers to make predictions based on a Prompt of Length $n$. It matches the lower bound of $n^{-\frac{2\alpha}{2\alpha+d}}$ (Györfi et al., 2006) for the regression of Hölder functions up to a $\log$ factor , and thereby demonstrating that transformers can achieve near-optimal performance if $\Gamma$ is large. Specifically, if $\Gamma \gtrsim n^{\frac{4\alpha}{2\alpha+d}+\delta}n^2 D^6 \log(nD)$ for some $\delta > 0$, then the second error term in (2) dominates the first term.

• **Dependence on the Intrinsic Dimension.** By leveraging low-dimensional geometric structures of data, the error in (2) has an exponential dependence on $d$ rather than the ambient dimension $D$. This improvement offers foundational insight into the role of geometry in ICL.

**Organization.** In this paper, we present some preliminaries in Section 2 and the problem setup in Section 3. We bridge attention to kernel methods in Section 4 and present the generalization error bound in Section 5. Related works are discussed in Section 7. Finally, we make conclusion and discuss the limitation of our paper in Section 8.

**Notation.** Throughout this paper, vectors are denoted by boldface letters, while scalars and matrices are denoted by standard (non-bold) letters. For a vector $\mathbf{x} \in \mathbb{R}^D$, we use $\|\mathbf{x}\|$ to denote its Euclidean norm. For a function $f : \Omega \to \mathbb{R}$, we denote its $L^\infty$ norm as $\|f\|_{L^\infty(\Omega)} := \sup_{\mathbf{x} \in \Omega} |f(\mathbf{x})|$.

## 2 PRELIMINARIES

In this section, we introduce preliminary definitions about manifolds, Hölder functions on manifolds, and the transformer neural networks used in this paper.

**Manifolds and Hölder Functions on Manifolds.** In this paper, we consider that data are sampled in a compact $d$-dimensional Riemannian manifold $\mathcal{M}$ isometrically embedded in $\mathbb{R}^D$. Mathematically, a $d$-dimensional *manifold* $\mathcal{M}$ is a topological space where each point has a neighborhood that is homeomorphic to an open subset of $\mathbb{R}^d$. Furthermore, distinct points in $\mathcal{M}$ can be separated by disjoint neighborhoods, and $\mathcal{M}$ has a countable basis for its topology. A formal definition of manifold and more definitions on geodesic distance and the reach of manifold are provided in Appendix B.1.

This work considers in-context regression of Hölder functions on $\mathcal{M}$.

**Definition 1** (Hölder function on a manifold). *A function $f : \mathcal{M} \to \mathbb{R}$ is Hölder continuous with Hölder exponent $\alpha \in (0, 1]$ and Hölder constant $L > 0$ if*

$$|f(\mathbf{x}) - f(\mathbf{x}')| \leq L d_{\mathcal{M}}^\alpha(\mathbf{x}, \mathbf{x}') \text{ for all } \mathbf{x}, \mathbf{x}' \in \mathcal{M}.$$

**Attention and Transformer Blocks.** We consider ICL using transformer-based networks structure Vaswani et al. (2017) in this paper. We briefly review attention and multi-head attention here.

**Definition 2** (Attention and Multi-head Attention). *Attention with the Query, Key, Value matrices $Q, K, V \in \mathbb{R}^{d_{embed} \times d_{embed}}$ is defined as*

$$A_{K,Q,V}(H) = VH\sigma((KH)^\top QH). \tag{3}$$

*The multi-head attention (MHA) with $m$ heads is given by*

$$\text{MHA}(H) = \sum_{j=1}^{m} V_j H\sigma((K_j H)^\top Q_j H). \tag{4}$$

We want to point out that in this paper we apply ReLU as the activation function of the attention modules from the first to the penultimate layers in the transformer, and apply Softmax for the last layer. A transformer block is a residual composition of the form

$$B(\theta; H) = \text{FFN}(\text{MHA}(H) + H) + \text{MHA}(H) + H. \tag{5}$$

where FFN is a feed-forward neural network operating tokenwise on the input.

## 3 IN-CONTEXT REGRESSION ON MANIFOLD

**Problem Setup.** Empirical evidence from image (Roweis & Saul, 2000; Tenenbaum et al., 2000; Pope et al., 2021) and language datasets (Sharma & Kaplan, 2022; Havrilla & Liao, 2024). suggests the

presence of underlying low-dimensional geometric structures in high-dimensional data. Motivated by this observation, our study adopts a geometric prior by assuming that the data $\mathbf{x}$ lies on a Riemannian manifold $\mathcal{M}$ of intrinsic dimension $d$, isometrically embedded in $\mathbb{R}^D$ with $d \ll D$.

With this geometric prior, we consider in-context learning for regression of functions defined on $\mathcal{M}$. More precisely, given a prompt/task as

$$\mathfrak{s} = \{\mathbf{x}_1, y_1, \mathbf{x}_2, y_2, \ldots, \mathbf{x}_n, y_n; \mathbf{x}_{n+1}\} \text{ with } y_i = f(\mathbf{x}_i), \tag{6}$$

where $\mathbf{x}_i$'s are i.i.d. samples from a distribution $\rho_{\mathbf{x}}$ supported on $\mathcal{M}$ and $f$ is sampled from $\rho_f$, a distribution in the function space $\{f : \mathcal{M} \to \mathbb{R}\}$, the goal is to predict $f(\mathbf{x}_{n+1})$ based on the following in-context learning problem.

Given $\{f^\gamma\}_{\gamma=1,\ldots,\Gamma} \overset{i.i.d.}{\sim} \rho_f$ and the corresponding training set $\mathfrak{S} := \{\mathfrak{s}^\gamma\}_{\gamma=1}^\Gamma$ provided by $\mathfrak{s}^\gamma = \{\mathbf{x}_1^\gamma, y_1^\gamma, \mathbf{x}_2^\gamma, y_2^\gamma, \ldots, \mathbf{x}_n^\gamma, y_n^\gamma; \mathbf{x}_{n+1}^\gamma, y_{n+1}^\gamma\}$ with $\{\mathbf{x}_i^\gamma\} \overset{i.i.d.}{\sim} \rho_{\mathbf{x}}$ and $y_i^\gamma = f^\gamma(\mathbf{x}_i^\gamma)$, we minimize the empirical risk:

$$\widehat{T} \in \underset{T_\theta \in \mathcal{T}}{\arg\min}\, \mathcal{R}_{n,\Gamma}(T_\theta) \text{ where } \mathcal{R}_{n,\Gamma}(T_\theta) := \frac{1}{\Gamma} \sum_{\gamma=1}^\Gamma \left( T_\theta(\{\mathbf{x}_i^\gamma, y_i^\gamma\}_{i=1}^n; \mathbf{x}_{n+1}^\gamma) - y_{n+1}^\gamma \right)^2 \tag{7}$$

where $T_\theta$ is a transformer neural network parameterized by $\theta$ and $\mathcal{T}$ is a transformer network class to be specified. Our goal is to study the squared generalization error of $\widehat{T}$ on a random test sample $\mathfrak{s}$ (independent of training data) in (6):

$$\mathcal{R}_n(\widehat{T}(\mathfrak{s})) := (\widehat{T}(\{\mathbf{x}_i, y_i\}_{i=1}^n; \mathbf{x}_{n+1}) - f(\mathbf{x}_{n+1}))^2. \tag{8}$$

This generalization error can be characterized by the mean squared generalization error defined as:

$$\mathcal{R}_n(\widehat{T}) = \mathbb{E}_{\mathfrak{S}} \mathbb{E}_{\mathfrak{s}} \left[ \mathcal{R}_n(\widehat{T}(\mathfrak{s})) \right] \tag{9}$$

where the expectation $\mathbb{E}_{\mathfrak{s}}$ is taken for the test sample $\mathfrak{s}$ and the expectation $\mathbb{E}_{\mathfrak{S}}$ is taken for the joint distribution of the training samples.

**Transformer Network Class.** To describe the ICL problem more precisely, let us specify the transformer network class. We define a transformer network $T_\theta(\cdot)$ with weights parametrized by $\theta$ as consisting of an embedding layer, a positional encoding module, a sequence of transformer blocks, and a decoding layer, i.e., for an input $\mathfrak{s}$ defined in (6)

$$T_\theta(\mathfrak{s}) := DE \circ B_{L_T} \circ \cdots \circ B_1 \circ (PE + E(\mathfrak{s})), \tag{10}$$

Here E is a linear embedding and PE is the operation of adding positional encoding (see their definitions in Appendix B.2). $PE + E(\mathfrak{s})$ embeds $\mathfrak{s}$ as a matrix $H$

$$H = PE + E(\mathfrak{s}) = \begin{bmatrix} \mathbf{x}_1 & \cdots & \mathbf{x}_n & \mathbf{x}_{n+1} & \mathbf{0} \\ y_1 & \cdots & y_n & 0 & \mathbf{0} \\ 0 & \cdots & \cdots & \cdots & 0 \\ \mathcal{I}_1 & \cdots & \cdots & \cdots & \mathcal{I}_\ell \\ 1 & \cdots & \cdots & \cdots & 1 \end{bmatrix} \in \mathbb{R}^{d_{embed} \times \ell} = \mathbb{R}^{(D+5) \times \ell}. \tag{11}$$

In matrix $H$, each column is a token, and each token has dimension $d_{embed} = D + 5$. The first $D + 2$ rows are data terms. The $(D + 3)$-th and $(D + 4)$-th rows contain the well-known sinusoidal positional encodings $\mathcal{I}_j = (\cos(\frac{j\pi}{2\ell}), \sin(\frac{j\pi}{2\ell}))^\top$, which determines how each token will interact with another through the attention mechanism. The last row contains the constant entries all equal to 1. It is crucial to note that the data terms are dynamic, whereas the positional encoding and constant terms remain static. Furthermore, $B_1, \cdots, B_{L_T} : \mathbb{R}^{d_{embed} \times \ell} \to \mathbb{R}^{d_{embed} \times \ell}$ are the transformer blocks (with ReLU activation from the first to the penultimate layers and Softmax activation for the last layer in the attention module) where each block consists of the residual composition of multi-head attention layers and feed-forward layers. $DE : \mathbb{R}^{d_{embed} \times \ell} \to \mathbb{R}$ is the decoding layer which outputs the element in the $(D + 1)$-th row and $(n + 1)$-th column as the final output.

Our ICL problem is considered in the following networks class:

**Definition 3** (Transformer Network Class). *The transformer network class with weights $\theta$ is*

$$\mathcal{T}(L_{\mathrm{T}}, m_{\mathrm{T}}, d_{embed}, \ell, L_{\mathrm{FFN}}, w_{\mathrm{FFN}}, R, \kappa)$$

$$= \Big\{ \mathrm{T}_\theta(\cdot) \mid \mathrm{T}_\theta(\cdot) \text{ has the form } (10) \text{ with } L_{\mathrm{T}} \text{ transformer blocks, at most } m_{\mathrm{T}} \text{ attention heads in}$$

*each block, embedded dimension $d_{embed}$, number of hidden tokens $\ell$, and $L_{\mathrm{FFN}}$ layers of feed-forward networks with hidden width $w_{\mathrm{FFN}}$, with output $\|\mathrm{T}_\theta(\cdot)\|_{L^\infty(\mathbb{R}^D)} \leq R$*

*and weight magnitude $\|\theta\|_\infty \leq \kappa \Big\}$.*

Throughout the paper, we will shorten the notation $\mathcal{T}(L_{\mathrm{T}}, m_{\mathrm{T}}, d_{embed}, \ell, L_{\mathrm{FFN}}, w_{\mathrm{FFN}}, R, \kappa)$ as $\mathcal{T}$ as long as there is no ambiguity in the context.

## 4 BRIDGING ATTENTION TO KERNEL METHODS

One key insight of this paper is to interpret transformers used in ICL as a mechanism for learning kernel methods in function regression. This interpretation not only illuminates the internal workings of transformers in the in-context regression setting, but also motivates a theoretical framework to understand transformers in ICL.

**Constructing a Transformer to Implement Kernel Method.** The classical (Nadaraya–Watson) kernel estimator (Nadaraya, 1964; Watson, 1964) is a well-established way for the estimation of $f(\mathbf{x}_{n+1})$ given $\{(\mathbf{x}_i, f(\mathbf{x}_i))\}_{i=1}^n$. It outputs

$$\mathcal{K}_h(\mathfrak{s}) := \frac{\sum_{i=1}^n K_h(\mathbf{x}_{n+1} - \mathbf{x}_i) y_i}{\sum_{i=1}^n K_h(\mathbf{x}_{n+1} - \mathbf{x}_i)}, \quad \text{with } y_i = f(\mathbf{x}_i). \tag{12}$$

where we choose $K_h(u) = e^{-\frac{\|u\|^2}{h^2}}$ to be the unnormalized Gaussian kernel with bandwidth $h > 0$. The transformer's attention mechanism can be interpreted as a form of kernel method, where the attention scores function analogously to kernel-based importance weights over input tokens. Our idea is to first use the interaction mechanism in attention to construct several layers of transformer blocks which takes the input $H$ in (11) and outputs the following matrix:

$$H = \begin{bmatrix} \mathbf{x}_1 & \cdots & \mathbf{x}_n & \mathbf{x}_{n+1} & \mathbf{x}_{n+1} - \mathbf{x}_1 & \cdots & \mathbf{x}_{n+1} - \mathbf{x}_n \\ y_1 & \cdots & y_n & 0 & -\frac{\|\mathbf{x}_{n+1} - \mathbf{x}_1\|^2}{h^2} & \cdots & -\frac{\|\mathbf{x}_{n+1} - \mathbf{x}_n\|^2}{h^2} \\ 0 & \cdots & \cdots & \cdots & y_1 & \cdots & y_n \\ \mathcal{I}_1 & \cdots & \cdots & \cdots & \cdots & \cdots & \mathcal{I}_{2n+1} \\ 1 & \cdots & \cdots & \cdots & \cdots & \cdots & 1 \end{bmatrix} \in \mathbb{R}^{(D+5) \times (2n+1)}. \tag{13}$$

We will present the construction details which operates on the $H$ in (11) and gives rise to the $H$ in (13) in Appendix E.1. This operation accounts for the first to the penultimate layer in our transformer network.

In the final layer, we apply a single-head attention A with a mask from the $(n+2)$-th to the $(2n+1)$-th token with value matrix $V = \mathbf{e}_{D+1} \mathbf{e}_{D+2}^\top$, and sparse query, key matrices $Q, K$ such that $Q^{data} \in \mathbb{R}^{(D+2) \times (D+5)}$ and $K^{data} \in \mathbb{R}^{(D+2) \times (D+5)}$. The exact form of $Q^{data}$ and $K^{data}$ are given in the last step of the proof of Lemma 1.

Then, the $(n+1)$-th output token is

$$[\mathrm{A}(H)]_{n+1} = \sum_{j=n+2}^{2n+1} \mathrm{softmax}\left(\langle Q^{data} h_{n+1}, K^{data} H \rangle\right)_j V h_j$$

$$= \sum_{j=1}^n \frac{e^{-\|\mathbf{x}_{n+1} - \mathbf{x}_j\|^2 / h^2}}{\sum_{j=1}^n e^{-\|\mathbf{x}_{n+1} - \mathbf{x}_j\|^2 / h^2}} \cdot (y_j \mathbf{e}_{D+1}) = \mathcal{K}_h(\mathfrak{s}) \cdot \mathbf{e}_{D+1}.$$

Here, we denote $\mathbf{e}_j$ as the elementary vector with all entries zero except for the $j$-th entry, which is 1. Therefore, the residual attention gives

$$\mathrm{A}(H) + H = \begin{bmatrix} \mathbf{x}_1 & \cdots & \mathbf{x}_n & \mathbf{x}_{n+1} & \mathbf{x}_{n+1} - \mathbf{x}_1 & \cdots & \mathbf{x}_{n+1} - \mathbf{x}_n \\ y_1 & \cdots & y_n & \mathcal{K}_h(\mathfrak{s}) & -\frac{\|\mathbf{x}_{n+1} - \mathbf{x}_1\|^2}{h^2} & \cdots & -\frac{\|\mathbf{x}_{n+1} - \mathbf{x}_n\|^2}{h^2} \\ 0 & \cdots & \cdots & \cdots & y_1 & \cdots & y_n \\ \mathcal{I}_1 & \cdots & \cdots & \cdots & \cdots & \cdots & \mathcal{I}_{2n+1} \\ 1 & \cdots & \cdots & \cdots & \cdots & \cdots & 1 \end{bmatrix} \in \mathbb{R}^{(D+5) \times (2n+1)},$$

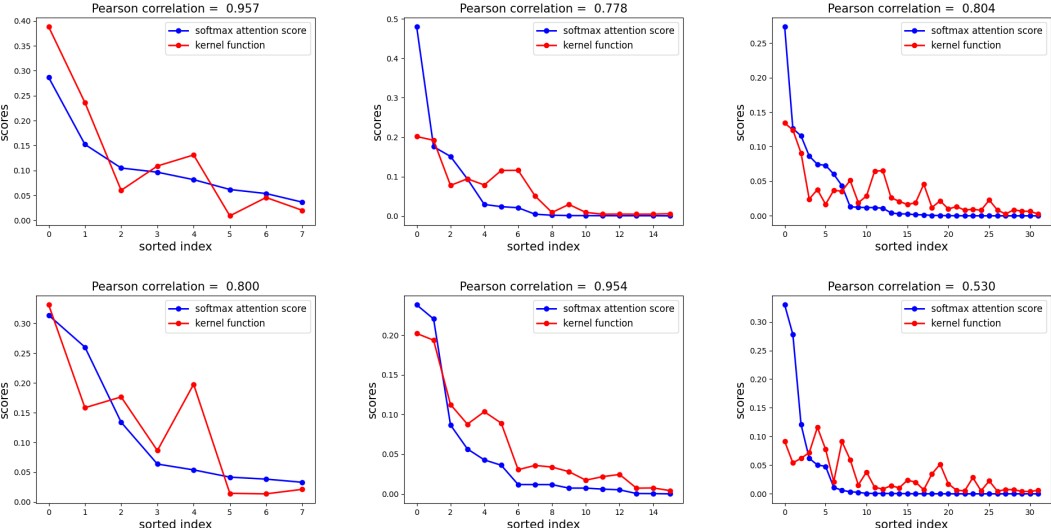

Figure 1: Examples of attention scores and Gaussian kernel with in-context length $n = 8$ (first column), $n = 16$ (second column), $n = 32$ (third column) respectively. The top and bottom rows are the plots at two different samples. This figure shows a strong correlation between attention scores and Gaussian kernel.

where the kernel estimator $\mathcal{K}_h(\mathfrak{s})$ is realized in $(D+1)$-th row and $(n+1)$-th column. Finally, the decoding operation produces this element in the $(D+1)$-th row and $(n+1)$-th column as the output.

This connection between the transformer network and the kernel estimator in (12) can be rigorously established, that is, *we prove that transformers can exactly implement the kernel estimator* (12) *without any error.* We summarize it as the following lemma, whose proof is in Appendix E.1.

**Lemma 1.** *Let $\mathcal{M} \subset [-b, b]^D$. Suppose the prompt $\mathfrak{s}$ in* (6) *satisfies: the $\mathbf{x}_i$'s are i.i.d. samples from a distribution $\rho_{\mathbf{x}}$ supported on $\mathcal{M}$ and $f : \mathcal{M} \to \mathbb{R}$ is bounded, i.e. $\|f\|_{L^\infty(\mathcal{M})} \le R$. Let $\mathcal{K}_h(\cdot)$ be the empirical kernel estimator defined in* (12). *Then there exists a transformer network $\mathrm{T}_h^* \in \mathcal{T}(L_T, m_T, d_{embed}, \ell, L_{FFN}, w_{FFN}, R, \kappa)$ with parameters*

$$L_\mathrm{T} = 5, \ m_\mathrm{T} = nD, \ d_{embed} = D + 5, \ \ell = 2n + 1,$$
$$L_{\mathrm{FFN}} = O(1), \ w_{\mathrm{FFN}} = D + 5, \ \kappa = O\left(D^8 n^2 b^8 R^4 / h^8\right)$$

*such that for any sample $\mathfrak{s}$ in the form of* (6), *we have*

$$\mathrm{T}_h^*(\mathfrak{s}) = \mathcal{K}_h(\mathfrak{s}). \tag{14}$$

*The notation $O(\cdot)$ hides absolute constants.*

**Remark 1** (Universality). *In Lemma 1, the network architecture and weight parameters of $\mathrm{T}_h^*$ are universal for different functions $f$ and points $\{\mathbf{x}_i\}_{i=1}^{n+1}$. The weight parameters only depend on $D, n, b, R, h$. This construction indicates that transformer can universally implement the kernel regression algorithm with zero approximation error.*

**Validating the Correlation between Attention Scores and Kernel Function.** To validate that transformer does indeed perform kernel regression implicitly, we conduct simulated experiment to compare the attention scores in the last layer of the trained transformer and the Gaussian kernel $e^{-\|\mathbf{x}_{n+1} - \mathbf{x}_i\|^2}$ to see if there is a strong correlation between the two.

In this simulation, we fix $\mathcal{M} = S^2$ (the 2-dimensional sphere), and we consider the target function $f : S^2 \to \mathbb{R}$ to be the linear combination of the real part of the first 10 spherical harmonics on the two-dimensional sphere $S^2$. More precisely, let $s_1(\theta, \phi), \cdots, s_{10}(\theta, \phi)$ be the real part of the first 10 spherical harmonics on $S^2$. For each task, we uniformly random sample the coefficients $w_k^\gamma \in [0, 1]$, and $\theta_i^\gamma \in [0, \pi]$, $\phi_i^\gamma \in [0, 2\pi]$, and generate $y_i^\gamma = \sum_{k=1}^{10} w_k^\gamma s_k(\theta_i^\gamma, \phi_i^\gamma)$, where $i = 1, \cdots, n$ (in-context length) and $\gamma = 1, \cdots, \Gamma$ (number of training tasks). Let $x_{1,i}^\gamma = \sin(\theta_i^\gamma) \cos(\phi_i^\gamma)$,

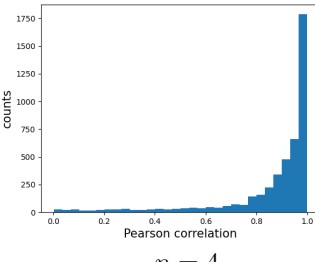 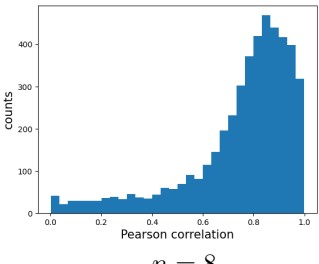 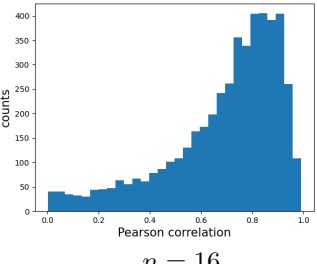

$n = 4$ $\qquad\qquad\qquad$ $n = 8$ $\qquad\qquad\qquad$ $n = 16$

Figure 2: Histograms of the Pearson correlation for $n = 4, 8, 16$ respectively. The ones with negative correlation are not included in this plot, while they only account for a small amount. The total counts for positive correlation are $4588, 4598, 4771$ out of a total of $5000$ samples in each case respectively.

Table 1: Average Pearson correlation coefficients ($\pm$ standard deviation) and the corresponding $p$-values ($\pm$ standard deviation)

| In-context length $n$ | Pearson correlation coefficient | $p$-value |
|:---:|:---:|:---:|
| 4 | $0.86 \pm 0.21$ | $0.14 \pm 0.21$ |
| 8 | $0.75 \pm 0.22$ | $0.09 \pm 0.19$ |
| 16 | $0.69 \pm 0.22$ | $0.06 \pm 0.17$ |
| 32 | $0.67 \pm 0.19$ | $0.03 \pm 0.12$ |

$x_{2,i}^{\gamma} = \sin(\theta_i^{\gamma})\sin(\phi_i^{\gamma})$, $x_{3,i}^{\gamma} = \cos(\theta_i^{\gamma})$. For each task, the training sample writes as the embedding matrix $H$ shown in (35) in the Appendix F. We fix the number of training and testing tasks $\Gamma = 50000$ and vary the in-context length $n = 4, 8, 16, 32$.

Figure 1 plots the attention scores (after sorting from the highest to the lowest value) in the last layer of the trained transformer and compares it against the Gaussian kernel (sorted according to the corresponding attention scores), which demonstrates a strong correlation between the two quantities. The distribution of the Pearson correlation values are plotted in Figure 2, we can see that most correlations are concentrated around $0.8$, showing that the attention score and Gaussian kernel are highly correlated with each other. More exemplar plots of attention scores and kernel function scores are provided in Figure 5 in Appendix F. The average Pearson correlation coefficients between the two scores and the corresponding $p$-values are also reported in Table 1. The results are averaged over $5000$ independent random testing samples. More details of the experimental setup are provided in Appendix F.

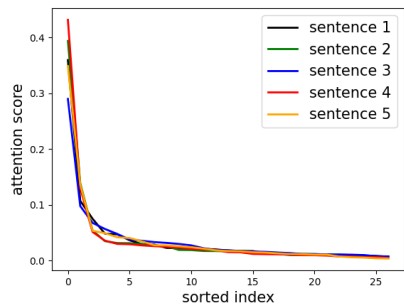

Figure 3: Softmax attention scores for real language data.

To further test how the curve of attention scores look like for real language data, we input five user generated sentences (with length about 20 - 30) into the pretrained GPT2 (Radford et al., 2019) and then plot the attention score for one of the heads in the model's last layer after sorting the score of each word from highest to the lowest value. The curves in Figure 3 shows that the attention scores for the real language data do exhibit some kernel shape.

## 5 TRANSFORMER-BASED ICL GENERALIZATION ERROR BOUND

Based on the connection between transformer and kernel methods, we derive a generalization error bounds for transformer-based ICL involving structured data. By imposing a geometric prior, we assume that $\mathbf{x}$ is sampled on a low-dimensional manifold $\mathcal{M}$, and $f$ is a function on the manifold $\mathcal{M}$.

This assumption leverages low-dimensional geometric structures in data which have been empirically observed in image (Roweis & Saul, 2000; Tenenbaum et al., 2000; Pope et al., 2021) and language datasets (Sharma & Kaplan, 2022; Havrilla & Liao, 2024).

**Assumption 1.** *Let $\mathcal{M}$ be a compact $d$-dimensional Riemannian manifold isometrically embedded in $\mathbb{R}^D$, $\mathcal{M} \subset [-b, b]^D$ for some $b > 0$, and $\mathcal{M}$ has a positive reach $\tau_{\mathcal{M}} > 0$ (reach is defined in Appendix B.1). Suppose $\rho_{\mathbf{x}}$ is the uniform distribution on $\mathcal{M}$.*

**Assumption 2.** *Let $\alpha \in (0, 1]$, $R, L > 0$, and $\rho_f$ be a probability distribution in the function space*

$$\mathcal{F} := \{f : \mathcal{M} \to \mathbb{R} : f \text{ is } \alpha\text{-Hölder with Hölder constant no more than } L, \text{ and } \|f\|_{L^\infty(\mathcal{M})} \leq R\}.$$

Our main theorem about the generalization error of transformer-based ICL is given below.

**Theorem 1.** *Suppose $\mathcal{M}$, $\rho_{\mathbf{x}}$ and $f$, $\rho_f$ satisfy Assumptions 1 and 2 respectively. If we choose the transformer network class $\mathcal{T}(L_T, m_T, d_{embed}, \ell, L_{FFN}, w_{FFN}, R, \kappa)$ with parameters*

$$L_T = 5, \ m_T = O(Dn), \ d_{embed} = D + 5, \ \ell = 2n + 1,$$
$$L_{FFN} = O(1), \ w_{FFN} = D + 5, \ \kappa = O\left(D^8 n^{\frac{4\alpha + 2d + 8}{2\alpha + d}} b^8 R^2\right),$$

*where $O(\cdot)$ hides the dependency on the absolute constants. Then the minimizer $\widehat{T}$ defined in (7) satisfies the squared generalization error bound:*

$$\mathcal{R}_n(\widehat{T}) \leq C_1 \left(nD^3 \Gamma^{-\frac{1}{2}} \sqrt{\log(nD\Gamma)}\right) + C_2 \left(n^{-\frac{2\alpha}{2\alpha + d}} \log^{1 + \frac{3d}{2}} n\right), \tag{15}$$

*where the constant $C_1$ depends on absolute constants, and the constant $C_2$ depends on $d, \alpha, L, R, \tau_{\mathcal{M}}$.*

This paper focuses on the scaling between the generalization error and $\Gamma, n$, and the constants $C_1, C_2$ may not be optimal. Note that in the worst case, $C_2$ can depend on $d^{d/2}$. The proof roadmap of Theorem 1 is presented in Appendix C and more details of the proof are provided in Appendix E. Lemma 1 is utilized as a key step to prove Theorem 1. Theorem 1 also offers insights into several key aspects of transformer-based ICL, which is discussed in the introduction.

**Validating the Generalization Error Bound.** We conduct simulated experiments to validate our generalization error bound (15) in Theorem 1 while varying $n$ (prompt length) and $\Gamma$ (number of training tasks). The data generating procedure is the same as the experiments in Section 4. Figure 4 plots the average Mean Squared Error (MSE) over 30 repetitions on the testing data against the number of tasks $\Gamma$ and the prompt length $n$. More details of the experiments are given in Appendix F.

The top row of Figure 4 shows the testing MSE with respect to $\Gamma$ in log-log scale when the prompt length is fixed to be $n = 16, 64, 256$ respectively. In log-log scale, the slope initially coincides with the theoretical slope of $-0.5$ in the first term of (15), and then slightly shifts above it. This is consistent with our error bound in (15), as when $\Gamma$ increases and $n$ is fixed, the second term starts to dominate the total error. In the bottom row of Figure 4, we plot the logarithm of testing MSE in terms of the prompt length $n$ when the number of tasks is fixed to be $\Gamma = 400, 1600, 6400$ respectively. In our error bound (15), both terms depend on $n$ while the first term increases and the second term decreases as $n$ increases. The testing MSE decays as $n$ increases, and the rate of decay depends on the balance of the two terms in (15). The larger $\Gamma$ is, the more dominant the second term in (15) is, and therefore the rate of convergence of the testing MSE is faster as $n$ increases. By comparing the three plots in the bottom row of Figure 4 with $\Gamma = 400, 1600, 6400$ respectively, we observe a faster rate of convergence with respect to $n$ when $\Gamma$ is larger, which is consistent with our theory.

## 6 ROADMAP FOR THE PROOF OF THEOREM 1

In this section, we present a roadmap for the proof of our main result in Theorem 1. We defer a more detailed discussion of the roadmap to Appendix C and all the proof details to Appendix E.

**Bias-Variance Error Decomposition.** We first decompose the squared generalization error at the test sample $\mathfrak{s}$ in (8) as follows:

$$\mathcal{R}_n(\widehat{T}(\mathfrak{s})) = \mathcal{R}_n(\widehat{T}(\mathfrak{s})) - \mathcal{R}_{n,\Gamma}(\widehat{T}) + \mathcal{R}_{n,\Gamma}(\widehat{T}) - \mathcal{R}_{n,\Gamma}(T^*) + \mathcal{R}_{n,\Gamma}(T^*) - \mathcal{R}_n(T^*(\mathfrak{s})) + \mathcal{R}_n(T^*(\mathfrak{s})),$$

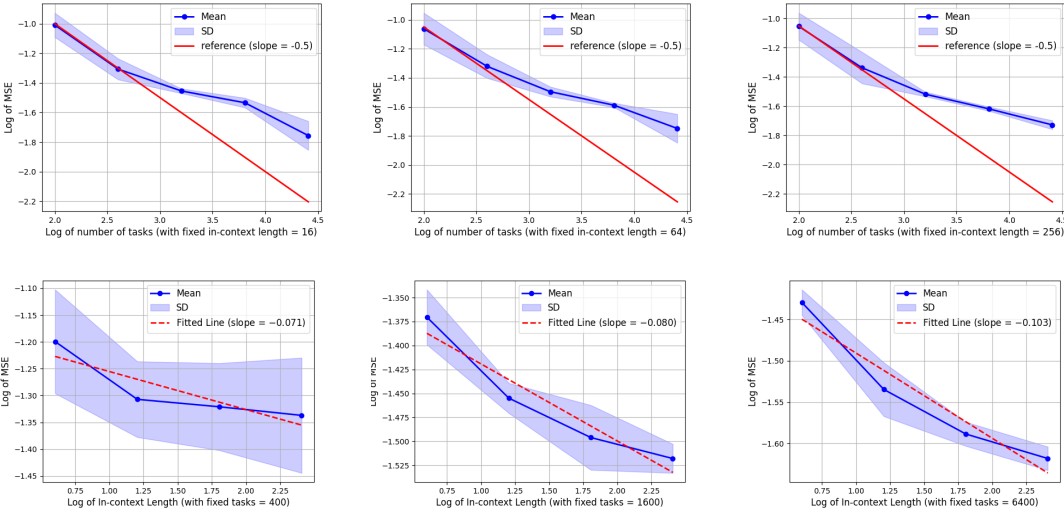

Figure 4: Top row: MSE v.s. number of tasks $\Gamma$ (with fixed prompt length $n = 16, 64, 256$, respectively). Bottom row: MSE v.s. prompt length $n$ (with fixed tasks $\Gamma = 400, 1600, 6400$ respectively). All plots are in log10-log10 scale.

where $T^*$ is a transformer network which approximates the in-context kernel estimator $\mathcal{K}_h$ in (12). One term satisfies $\mathcal{R}_{n,\Gamma}(\widehat{T}) - \mathcal{R}_{n,\Gamma}(T^*) \leq 0$ since $\widehat{T}$ is the minimizer of $\mathcal{R}_{n,\Gamma}$ given in (7). After taking expectations, we can decompose the mean squared generalization error (9) as follows:

$$\mathcal{R}_n(\widehat{T}) = \mathbb{E}_{\mathfrak{S}}\mathbb{E}_{\mathfrak{s}}\left[\mathcal{R}_n(\widehat{T}(\mathfrak{s}))\right] \tag{16}$$

$$\leq \underbrace{\mathbb{E}_{\mathfrak{S}}\left(\mathbb{E}_{\mathfrak{s}}\left[\mathcal{R}_n(\widehat{T}(\mathfrak{s}))\right] - \mathcal{R}_{n,\Gamma}(\widehat{T})\right)}_{\text{I}} + \underbrace{\mathbb{E}_{\mathfrak{S}}\left(\mathcal{R}_{n,\Gamma}(T^*) - \mathbb{E}_{\mathfrak{s}}\left[\mathcal{R}_n(T^*(\mathfrak{s}))\right]\right)}_{\text{II}} + \underbrace{\mathbb{E}_{\mathfrak{s}}\left[\mathcal{R}_n(T^*(\mathfrak{s}))\right]}_{\text{III}}.$$

In this error decomposition, error III denotes the approximation error which we will analyze in Section C.1. The errors in I and II denote the statistical error , which we will analyze in Section C.2.

In C.1 and C.2, we obtain

$$\text{III} \leq O\left(\frac{\left[\log\left(h^{-1}\right)\right]^{1+\frac{3d}{2}}}{nh^d} + h^{2\alpha}[\log(h^{-1})]^2\right) \quad \text{and} \quad \text{I+II} \leq O\left(\frac{nD^3\sqrt{\log(nD\Gamma/h)}}{\sqrt{\Gamma}} + h^2\right)$$

respectively. Putting them together, we get

$$\mathcal{R}_n(\widehat{T}) \leq \text{I} + \text{II} + \text{III} \leq C_1\left(\frac{nD^3\sqrt{\log(nD\Gamma/h)}}{\sqrt{\Gamma}}\right) + C_2\left(\frac{\left[\log\left(h^{-1}\right)\right]^{1+\frac{3d}{2}}}{nh^d} + h^{2\alpha}[\log(h^{-1})]^2\right).$$

Finally, choosing $h = n^{-\frac{1}{2\alpha+d}}$ gives rise to (15) in Theorem 1.

## 7 RELATED WORKS

We next discuss some connection and comparison of our result with existing works. This paper highlights bridging the attention mechanism and classical kernel methods. It provides a new interpretation of transformers in ICL, and new tools to address nonlinear models in transformer-based ICL, which allows us to move beyond linear models studied in Bai et al. (2023); Zhang et al. (2024); Von Oswald et al. (2023); Akyürek et al. (2022); Cole et al. (2024). This novel tool also allows us to address multiple tasks in ICL, in contrast to single task learning by transformers studied in Yun et al. (2019); Takakura & Suzuki (2023); Gurevych et al. (2022); Havrilla & Liao (2024); Shen et al. (2025).

The most closely related work to this paper is Kim et al. (2024), which studied in-context regression of Besov functions in $\mathbb{R}^D$. Kim et al. (2024) derived approximation and generalization error bounds for a transformer composed of a deep feedforward network and one linear attention layer. There are two main differences between this paper and Kim et al. (2024): 1) Our transformer network has 5 layers of multi-head attention, and each multi-head attention can be wide, i.e. with $nD$ attention heads. The feedforward layers in each attention is of a constant order. Such a wide transformer architecture shares some similarity to those used in large language models (LLMs). For example, GPT-2 Small only has 12 layers with 117 million parameters (Radford et al., 2019). Our approximation theory is developed by fully leveraging the attention mechanism. In contrast, Kim et al. (2024) utilized one linear attention layer and a deep feedforward network for approximation. 2) By incorporating low-dimensional geometric structures of data, we prove error bounds which scale exponentially with respect to the prompt length with the exponent depending on the intrinsic dimension $d$, while the error bound in Kim et al. (2024) scales exponentially depending on the ambient dimension $D$.

Our work is also connected with Li et al. (2023), which derived generalization errors for transformers as in-context algorithm learners. While the framework in Li et al. (2023) is general, it does not address some key components in this paper, such as our novel approximation theory bridging the attention mechanism to kernel methods, and our covering number calculation.

To understand transformers, the attention mechanism has also been connected to traditional mathematical models, such as interacting particle systems (Geshkovski et al., 2025) and integral-differential equations (Tai et al., 2025). The connection between the attention mechanism and kernel methods has been explored in Tsai et al. (2019); Song et al. (2021); Yu et al. (2024); Lu & Yu (2025); Cheng et al. (2024); Han et al. (2025). In particular, the work by Han et al. (2025) takes the kernel perspective to understand ICL and empirically demonstrates that the attention and hidden features in LLMs match the behaviors of a kernel regression. While our work and these prior studies all draw on the connection between the attention mechanism and kernel methods, our theoretical justification is novel. In particular, the construction of transformers to implement the kernel method in Lemma 1 and the generalization error bound in Theorem 1 have not been addressed in literature. Our paper provides a theoretical framework to understand transformer-based ICL with geometric structures.

Besides transformers, in-context operator learning has also been studied using DeepONets based on feedforward neural networks (Chiu et al., 2025). In contrast, our paper focuses on the role of the attention mechanism in ICL.

## 8 CONCLUSION AND DISCUSSION

**Conclusion.** This work provides a theoretical foundation for understanding in-context learning (ICL) with transformers in the setting of manifold-structured regression tasks. By establishing a novel connection between the attention mechanism and classical kernel regression, we interpret transformers as implicitly learning kernel-based algorithms for function regression. Our findings offer new theoretical insights into the algorithmic nature of transformers in ICL, establish a rigorous approximation and generalization theory for manifold regression, and provide tools for analyzing nonlinear models under geometric structure.

Our analysis derives generalization error bounds for $\alpha$-Hölder functions on compact Riemannian manifolds, revealing how the performance of transformers in ICL depends on the prompt length $n$, the number of training tasks $\Gamma$, and the intrinsic geometry of the data. Notably, our results show that transformers can achieve the minimax optimal regression rate up to logarithmic factors when $\Gamma$ is sufficiently large. Furthermore, the derived bounds depend exponentially with respect to the prompt length with the exponent depending on the intrinsic dimension $d$ of the manifold, rather than the ambient dimension $D$, highlighting the critical role of low-dimensional geometric priors.

**Discussion.** Our theoretical analysis focuses on $\alpha$-Hölder regression with fixed-length prompts and a large number of training tasks, under idealized assumptions such as exact kernel implementation via attention. Extending the framework to broader function classes, variable prompt lengths, noisy data or limited data remains an open challenge. Despite these limitations, our work reveals how geometric structure can enhance generalization in ICL and draws a principled connection between attention mechanisms and kernel methods.

**Reproducibility statement** To support reproducibility of our work, we provide comprehensive details across the main paper and supplementary material. All theoretical results are accompanied by clear assumptions and complete proofs in the appendix. For experimental results, we describe the datasets used, preprocessing steps, and hyperparameter settings in both the main text and the appendix. The implementation code used to generate the experimental results in this paper is available in the supplementary material.

ACKNOWLEDGMENTS

Zhaiming Shen and Wenjing Liao are partially supported by National Science Foundation under the NSF DMS 2145167 and the U.S. Department of Energy under the DOE SC0024348. Alexander Hsu and Rongjie Lai's research are supported in part by NSF DMS-2401297.

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

## A  THE USE OF LARGE LANGUAGE MODELS (LLMs)

We used ChatGPT for minor language editing, such as grammar correction and improving sentence flow. The scientific content and all writing were created by the authors.

## B  MORE DEFINITIONS

### B.1  MANIFOLD, GEODESIC DISTANCE, REACH OF THE MANIFOLD AND COVERING NUMBER

**Definition 4** (Manifold). *An $d$-dimensional* manifold $\mathcal{M}$ *is a topological space where each point has a neighborhood that is homeomorphic to an open subset of $\mathbb{R}^d$. Further, distinct points in $\mathcal{M}$ can be separated by disjoint neighborhoods, and $\mathcal{M}$ has a countable basis for its topology.*

With the induced metric on $\mathcal{M}$, the *geodesic distance* on the manifold between $\mathbf{x}, \mathbf{x}' \in \mathcal{M}$ is defined as

$$d_{\mathcal{M}}(\mathbf{x}, \mathbf{x}') := \inf\{|\gamma| : \gamma \in C^1([t, t']), \gamma : [t, t'] \to \mathcal{M}, \gamma(t) = \mathbf{x}, \gamma(t') = \mathbf{x}'\},$$

where the length is defined by $|\gamma| := \int_t^{t'} \|\gamma'(s)\|_2 ds$. The existence of a length-minimizing geodesic $\gamma : [t, t'] \to \mathcal{M}$ between any two points $\mathbf{x} = \gamma(t), \mathbf{x}' = \gamma(t')$ is guaranteed by the Hopf–Rinow theorem (Hopf & Rinow, 1931).

**Definition 5** (Medial Axis). *Let $\mathcal{M} \subseteq \mathbb{R}^D$ be a connected and compact $d$-dimensional submanifold. Its* medial axis *is defined as*

$$\mathrm{Med}(\mathcal{M}) := \{\mathbf{x} \in \mathbb{R}^D \mid \exists \mathbf{p} \neq \mathbf{q} \in \mathcal{M}, \|\mathbf{p} - \mathbf{x}\|_2 = \|\mathbf{q} - \mathbf{x}\|_2 = \inf_{\mathbf{z} \in \mathcal{M}} \|\mathbf{z} - \mathbf{x}\|_2\},$$

*which contains all points $\mathbf{x} \in \mathbb{R}^D$ with set-valued orthogonal projection $\pi_{\mathcal{M}}(\mathbf{x}) = \operatorname{argmin}_{\mathbf{z} \in \mathcal{M}} \|\mathbf{x} - \mathbf{z}\|_2$.*

**Definition 6** (Local Reach and Reach of a Manifold). *The* local reach *for $\mathbf{v} \in \mathcal{M}$ is defined as $\tau_{\mathcal{M}}(\mathbf{v}) := \inf_{\mathbf{z} \in \mathrm{Med}(\mathcal{M})} \|\mathbf{v} - \mathbf{z}\|_2$, which describes the minimum distance needed to travel from $\mathbf{v}$ to the closure of medial axis. The smallest local reach $\tau_{\mathcal{M}} := \inf_{\mathbf{v} \in \mathcal{M}} \tau_{\mathcal{M}}(\mathbf{v})$ is called* reach *of $\mathcal{M}$.*

**Definition 7** (Covering Number). *Let $(\mathcal{H}, \rho)$ be a metric space, where $\mathcal{H}$ is the set of objects and $\rho$ is a metric. For a given $\epsilon > 0$, the* covering number *$\mathcal{N}(\epsilon, \mathcal{H}, \rho)$ is the smallest number of balls of radius $\epsilon$ (with respect to $\rho$) needed to cover $\mathcal{H}$. More precisely,*

$$\mathcal{N}(\epsilon, \mathcal{H}, \rho) := \min\{N \in \mathbb{N} \mid \exists\{h_1, \ldots, h_N\} \subseteq \mathcal{H}, \forall h \in \mathcal{H}, \exists h_i \text{ such that } \rho(h, h_i) \leq \epsilon\}.$$

### B.2  EMBEDDING, POSITIONAL ENCODING, FEED-FORWARD NETWORK CLASS AND TRANSFORMER BLOCK CLASS

**Definition 8** (Embedding Layer). *Given $\mathbf{x}_i \in \mathbb{R}^D$ and $y_i \in \mathbb{R}$, the* embedding layer *$E$ takes an input $\mathfrak{s} = \{\mathbf{x}_1, y_1, \mathbf{x}_2, y_2, \ldots, \mathbf{x}_n, y_n; \mathbf{x}_{n+1}\}$ and maps it to*

$$\mathrm{E}(\mathfrak{s}) = \begin{bmatrix} \mathbf{x}_1 & \cdots & \mathbf{x}_n & \mathbf{x}_{n+1} & \mathbf{0} \\ y_1 & \cdots & y_n & 0 & \mathbf{0} \\ 0 & \cdots & \cdots & \cdots & 0 \\ 0 & \cdots & \cdots & \cdots & 0 \\ 0 & \cdots & \cdots & \cdots & 0 \\ 1 & \cdots & \cdots & \cdots & 1 \end{bmatrix} \in \mathbb{R}^{(D+5) \times \ell}.$$

**Definition 9** (Positional Encoding). *The* positional encoding *takes an input $\mathfrak{s}$, maps it to $\mathcal{I}_j = (\cos(\frac{j\pi}{2\ell}), \sin(\frac{j\pi}{2\ell}))^\top$ and put those $\mathcal{I}_j$, $j = 1, \cdots, \ell$, into the third and second last row in the embedding matrix, i.e.,*

$$\mathrm{PE}(\mathfrak{s}) = \begin{bmatrix} 0 & \cdots & \cdots & \cdots & 0 \\ 0 & \cdots & \cdots & \cdots & 0 \\ 0 & \cdots & \cdots & \cdots & 0 \\ \mathcal{I}_1 & \cdots & \cdots & \cdots & \mathcal{I}_\ell \\ 0 & \cdots & \cdots & \cdots & 0 \end{bmatrix} \in \mathbb{R}^{(D+5) \times \ell}.$$

With these definitions, we have

$$
\mathrm{PE} + \mathrm{E}\left(\mathfrak{s}\right) = \begin{bmatrix} \mathbf{x}_1 & \cdots & \mathbf{x}_n & \mathbf{x}_{n+1} & \mathbf{0} \\ y_1 & \cdots & y_n & 0 & \mathbf{0} \\ 0 & \cdots & \cdots & \cdots & 0 \\ \mathcal{I}_1 & \cdots & \cdots & \cdots & \mathcal{I}_\ell \\ 1 & \cdots & \cdots & \cdots & 1 \end{bmatrix} \in \mathbb{R}^{(D+5)\times \ell},
$$

as defined in (11).

**Definition 10** (Feed-forward Network Class)**.** *The feed-forward neural network (FFN) class with weights $\theta$ is*

$$
\mathcal{FFN}(L_{\mathrm{FFN}}, w_{\mathrm{FFN}}) = \{\mathrm{FFN}(\theta; \cdot) \mid \mathrm{FFN}(\theta; \cdot) \text{ is a FNN with at most } L_{\mathrm{FFN}} \text{ layers and width } w_{\mathrm{FFN}}\}.
$$

We use ReLU function $\sigma(x) = \max(x, 0)$ as the activation function in the feed-forward network. Note that each feed-forward layer is applied tokenwise to an embedding matrix $H$.

**Definition 11** (Transformer Block Class)**.** *The transformer block class with parameters $\theta$ is*

$$
\mathcal{B}(m, L_{\mathrm{FFN}}, w_{\mathrm{FFN}}) = \{\mathrm{B}(\theta; \cdot) \mid \mathrm{B}(\theta; \cdot) \text{ a MHA with } m \text{ attention heads, and a FFN layer}
$$
$$
\text{with depth } L_{\mathrm{FFN}} \text{ and width } w_{\mathrm{FFN}}\}.
$$

## C  ROADMAP FOR THE PROOF OF THEOREM 1

In this section, we present a detailed roadmap for the proof of our main result in Theorem 1. We defer all the proof details to Appendix E.

**Bias-Variance Error Decomposition.** We first decompose the squared generalization error at the test sample $\mathfrak{s}$ in (8) as follows:

$$
\mathcal{R}_n(\widehat{\mathrm{T}}(\mathfrak{s})) = \mathcal{R}_n(\widehat{\mathrm{T}}(\mathfrak{s})) - \mathcal{R}_{n,\Gamma}(\widehat{\mathrm{T}}) + \mathcal{R}_{n,\Gamma}(\widehat{\mathrm{T}}) - \mathcal{R}_{n,\Gamma}(\mathrm{T}^*) + \mathcal{R}_{n,\Gamma}(\mathrm{T}^*) - \mathcal{R}_n(\mathrm{T}^*(\mathfrak{s})) + \mathcal{R}_n(\mathrm{T}^*(\mathfrak{s})),
$$

where $\mathrm{T}^*$ is a transformer network which approximates the in-context kernel estimator $\mathcal{K}_h$ in (12). One term satisfies $\mathcal{R}_{n,\Gamma}(\widehat{\mathrm{T}}) - \mathcal{R}_{n,\Gamma}(\mathrm{T}^*) \leq 0$ since $\widehat{\mathrm{T}}$ is the minimizer of $\mathcal{R}_{n,\Gamma}$ given in (7). After taking expectations, we can decompose the mean squared generalization error (9) as follows:

$$
\mathcal{R}_n(\widehat{\mathrm{T}}) = \mathbb{E}_{\mathfrak{S}}\mathbb{E}_{\mathfrak{s}}\left[\mathcal{R}_n(\widehat{\mathrm{T}}(\mathfrak{s}))\right]
$$
$$
\leq \underbrace{\mathbb{E}_{\mathfrak{S}}\left(\mathbb{E}_{\mathfrak{s}}\left[\mathcal{R}_n(\widehat{\mathrm{T}}(\mathfrak{s}))\right] - \mathcal{R}_{n,\Gamma}(\widehat{\mathrm{T}})\right)}_{\mathrm{I}} + \underbrace{\mathbb{E}_{\mathfrak{S}}\left(\mathcal{R}_{n,\Gamma}(\mathrm{T}^*) - \mathbb{E}_{\mathfrak{s}}\left[\mathcal{R}_n(\mathrm{T}^*(\mathfrak{s}))\right]\right)}_{\mathrm{II}} + \underbrace{\mathbb{E}_{\mathfrak{s}}\left[\mathcal{R}_n(\mathrm{T}^*(\mathfrak{s}))\right]}_{\mathrm{III}}.
$$

In this error decomposition, error III denotes the approximation error which we will analyze in Section C.1. The errors in I and II denote the statistical error , which we will analyze in Section C.2.

### C.1  APPROXIMATION ERROR: TRANSFORMERS CAN IMPLEMENT KERNEL ESTIMATOR

A key innovation in our proof is establishing an approximation theory for transformers to implement the classical kernel estimator in (12). Importantly, this implementation is universal for $f$ and the $\mathbf{x}_i$'s so that the weight matrices in transformer are independent of $f$ and the $\mathbf{x}_i$'s.

Our approximation error bound is given by Proposition 1 below. Since transformers can exactly implement the kernel estimator as shown in Lemma 1, the approximation error in Proposition 1 is the same as the mean squared error given by the kernel estimator.

**Proposition 1.** *Suppose $\mathcal{M}$, $\rho_{\mathbf{x}}$ and $f$, $\rho_f$ satisfy Assumptions 1 and 2 respectively. Let $\mathfrak{s}$ be a prompt in (6), where $\{\mathbf{x}_i\}_{i=1}^{n+1}$ are i.i.d. samples from $\rho_{\mathbf{x}}$ and $f$ is sampled from $\rho_f$. There exists a transformer network $\mathrm{T}^* \in \mathcal{T}(L_{\mathrm{T}}, m_{\mathrm{T}}, d_{embed}, \ell, L_{\mathrm{FFN}}, w_{\mathrm{FFN}}, R, \kappa)$ with parameters*

$$
L_{\mathrm{T}} = 5, \ m_{\mathrm{T}} = O(Dn), \ d_{embed} = D + 5, \ \ell = 2n + 1,
$$
$$
L_{\mathrm{FFN}} = O(1), \ w_{\mathrm{FFN}} = D + 5, \ \kappa = O\left(D^8 n^2 b^8 R^2 / h^8\right)
$$

*such that*

$$\mathbb{E}_{\mathfrak{s}}\left[\mathcal{R}_n(\mathrm{T}^*(\mathfrak{s}))\right] \leq C_3 \left( \frac{\left[\log\left(h^{-1}\right)\right]^{1+\frac{3d}{2}}}{nh^d} + h^{2\alpha}[\log(h^{-1})]^2 \right). \tag{17}$$

*The constant $C_3$ hides the constants depending on $d, L, R, \tau_{\mathcal{M}}$.*

Proposition 1 is proved in Appendix E.4. Here we illustrate our proof idea. The empirical kernel estimator in (12) is applied to $n$ samples in the prompt. When $n \to \infty$, the empirical kernel estimator in (12) converges to its integral counterpart. Given any $f \in \mathcal{F}$ and $\mathbf{x}_{n+1} \sim \rho_{\mathbf{x}}$, we define the integral form of the kernel estimator as

$$\bar{\mathcal{K}}_h(f; \mathbf{x}_{n+1}) := \frac{\mathbb{E}_{\mathbf{x}}\left[K_h(\mathbf{x}_{n+1} - \mathbf{x})f(\mathbf{x})\right]}{\mathbb{E}_{\mathbf{x}}\left[K_h(\mathbf{x}_{n+1} - \mathbf{x})\right]} = \frac{\int K_h(\mathbf{x}_{n+1} - \mathbf{x})f(\mathbf{x})d\mathbf{x}}{\int K_h(\mathbf{x}_{n+1} - \mathbf{x})d\mathbf{x}}, \tag{18}$$

where the integral about $\mathbf{x}$ is about the measure $d\mathbf{x} = d\rho_{\mathbf{x}}$. For any test sample $\mathfrak{s}$, the approximation error III can be further decomposed into three terms:

$$\mathcal{R}_n(\mathrm{T}_h^*(\mathfrak{s})) = (\mathrm{T}_h^*(\mathfrak{s}) - f(\mathbf{x}_{n+1}))^2 \tag{19}$$

$$= \left(\mathrm{T}_h^*(\mathfrak{s}) - \mathcal{K}_h(\mathfrak{s}) + \mathcal{K}_h(\mathfrak{s}) - \bar{\mathcal{K}}_h(f; \mathbf{x}_{n+1}) + \bar{\mathcal{K}}_h(f; \mathbf{x}_{n+1}) - f(\mathbf{x}_{n+1})\right)^2$$

$$\leq 3\underbrace{(\mathrm{T}_h^*(\mathfrak{s}) - \mathcal{K}_h(\mathfrak{s}))^2}_{\text{A1}} + 3\underbrace{\left(\mathcal{K}_h(\mathfrak{s}) - \bar{\mathcal{K}}_h(f; \mathbf{x}_{n+1})\right)^2}_{\text{A2}} + 3\underbrace{\left(\bar{\mathcal{K}}_h(f; \mathbf{x}_{n+1}) - f(\mathbf{x}_{n+1})\right)^2}_{\text{A3}}$$

where the error in A1 measures the bias of using a transformer network to implement the kernel regression estimator, the error in A2 measures the variance of kernel regression estimator, and the A3 error measures the bias of using kernel regression estimator to approximate the target function. The bias and variance of kernel estimators has been studied in the literature of nonparametric estimation (Tang et al., 2024). In our paper, these three error terms are bounded by Lemma 1, Lemma 2, and Lemma 3 respectively. Our Lemma 1 shows that the error in A1 equals to 0. Lemma 2 and Lemma 3 give rise to an upper bound of A2 and A3. We defer their proof in Appendix E.2 and Appendix E.3, respectively.

**Lemma 2.** *Let $\bar{\mathcal{K}}_h$ be the integral kernel estimator defined as in (18). For any $\mathbf{x}_{n+1} \sim \rho_{\mathbf{x}}$ and any $\mathcal{M}$ and $f$ satisfying Assumptions 1 and 2 respectively,*

$$\left|\bar{\mathcal{K}}_h(f; \mathbf{x}_{n+1}) - f(\mathbf{x}_{n+1})\right| \leq O(h^\alpha \log(h^{-1})). \tag{20}$$

*The constant hidden in $O(\cdot)$ depend on $d, L, R, \tau_{\mathcal{M}}$, where the dependency on $d$ can be exponential in the worse case.*

**Lemma 3.** *Suppose $\mathcal{M}, \rho_{\mathbf{x}}$ and $f, \rho_f$ satisfy Assumptions 1 and 2 respectively. Let $\mathfrak{s}$ be a prompt in (6), where $\{\mathbf{x}_i\}_{i=1}^{n+1}$ are i.i.d. samples from $\rho_{\mathbf{x}}$ and $f$ is sampled from $\rho_f$. Let $\mathcal{K}_h$ and $\bar{\mathcal{K}}_h$ be defined as in (12) and (18). Then with probability at least $1 - \delta$,*

$$\left|\mathcal{K}_h(\mathfrak{s}) - \bar{\mathcal{K}}_h(f; \mathbf{x}_{n+1})\right| \leq O\left(\log^{3d/4}\left(\frac{1}{h}\right)\sqrt{\frac{\log(4/\delta)}{nh^d}}\right) \tag{21}$$

*The constants hidden in $O(\cdot)$ depend on $d, R, \tau_{\mathcal{M}}$, where the dependency on $d$ can be $d^{d/2}$ in the worse case.*

### C.2 STATISTICAL ERROR

This section focuses on bounding the statistical errors in I and II in (16). We consider a transformer network class $\mathcal{T}$ with the architecture in Proposition 1.

**Bounding Error** I **in** (16). The error in I in (16) can be bounded by

$$\mathbb{E}_{\mathfrak{S}} \sup_{\mathrm{T} \in \mathcal{T}} \left[\mathbb{E}_{\mathfrak{s}}\left[\mathcal{R}_n(\mathrm{T}(\mathfrak{s}))\right] - \mathcal{R}_{n,\Gamma}(\mathrm{T})\right]$$

since $\widehat{\mathrm{T}}$ belongs to the network class $\mathcal{T}$. Any function $f \sim \rho_f$ is bounded, such that $\|f\|_{L^\infty} \leq R$. The transformer network T also yields bounded output such that $\|\mathrm{T}\|_{L^\infty} \leq R$. For the sample $\mathfrak{s}$ in (6), the transformer neural network T takes the input $(\{\mathbf{x}_i, y_i\}_{i=1}^n; \mathbf{x}_{n+1})$ and outputs $\mathrm{T}(\{\mathbf{x}_i, y_i\}_{i=1}^n; \mathbf{x}_{n+1}) \in [-R, R]$. In this paper, we take squared loss at each sample:

$$\mathcal{L}(\mathrm{T}, \mathfrak{s}, y_{n+1}) = (\mathrm{T}(\{\mathbf{x}_i, y_i\}_{i=1}^n; \mathbf{x}_{n+1}) - y_{n+1})^2,$$

which satisfies $|\mathcal{L}(\mathrm{T}, \mathfrak{s}, y_{n+1})| \leq 4R^2$. We define the Rademacher complexity of $\mathcal{L} \circ \mathcal{T}$ with respect to the training sample $\mathfrak{S}$ as

$$\mathrm{Rad}(\mathcal{L} \circ \mathcal{T} \circ \mathfrak{S}) := \frac{1}{\Gamma} \mathbb{E}_{\boldsymbol{\xi} \sim \{\pm 1\}^{\Gamma}} \left[ \sup_{\mathrm{T} \in \mathcal{T}} \sum_{\gamma=1}^{\Gamma} \xi_{\gamma} \Big( \mathrm{T}(\{\mathbf{x}_i^{\gamma}, y_i^{\gamma}\}_{i=1}^{n}; \mathbf{x}_{n+1}^{\gamma}) - y_{n+1}^{\gamma} \Big)^2 \right]. \quad (22)$$

According to Shalev-Shwartz & Ben-David (2014, Lemma 26.2), we have

$$\mathbb{E}_{\mathfrak{S}} \sup_{\mathrm{T} \in \mathcal{T}} \left[ \mathbb{E}_{\mathfrak{s}} \left[ \mathcal{R}_n(\mathrm{T}(\mathfrak{s})) \right] - \mathcal{R}_{n,\Gamma}(\mathrm{T}) \right] \leq 2 \mathbb{E}_{\mathfrak{S}} \left[ \mathrm{Rad}(\mathcal{L} \circ \mathcal{T} \circ \mathfrak{S}) \right].$$

To bound the expectation of Rademacher complexity $\mathrm{Rad}(\mathcal{L} \circ \mathcal{T} \circ \mathfrak{S})$, we apply the well known Dudley entropy integral (Dudley, 1967), which we state in Lemma 8. By Lemma 8, we have

$$\mathbb{E}_{\mathfrak{S}} \left[ \mathrm{Rad}(\mathcal{L} \circ \mathcal{T} \circ \mathfrak{S}) \right] \leq \inf_{\epsilon > 0} \left( 2\epsilon + \frac{12}{\sqrt{\Gamma}} \int_{\epsilon}^{4R^2} \sqrt{\log \mathcal{N}(\delta, \mathcal{L} \circ \mathcal{T}, \|\cdot\|_{L^\infty})} \, d\delta \right) \quad (23)$$

where $\mathcal{N}(\delta, \mathcal{L} \circ \mathcal{T}, \|\cdot\|_{L^\infty})$ is the covering number (defined in Appendix B.1) of the function class $\mathcal{L} \circ \mathcal{T}$ under the $L^\infty$ norm. We follow the proof idea in Havrilla & Liao (2024) to bound its covering number as follows.

**Lemma 4.** *For a transformer class* $\mathcal{T}(L_\mathrm{T}, m_\mathrm{T}, d_{embed}, \ell, L_{\mathrm{FFN}}, w_{\mathrm{FFN}}, R, \kappa)$ *with input* $\|H\|_{\infty,\infty} \leq U$. *Let* $\delta > 0$, *then the covering number of* $\mathcal{L} \circ \mathcal{T}$ *satisfies*

$$\mathcal{N}(\delta, \mathcal{L} \circ \mathcal{T}, \|\cdot\|_{L^\infty}) \leq \left( \frac{2^{L_\mathrm{T}^2+4} L_{\mathrm{FFN}} U^{3L_\mathrm{T}} d_{embed}^{18L_\mathrm{T}^2 L_{\mathrm{FFN}}} \kappa^{6L_\mathrm{T}^2 L_{\mathrm{FFN}}+1} m_\mathrm{T}^{L_\mathrm{T}^2} \ell^{L_\mathrm{T}^2} R}{\delta} \right)^{P_\mathrm{T}},$$

*where* $P_\mathrm{T} = d_{embed}(D+1) + L_\mathrm{T}(3d_{embed}^2 m_\mathrm{T} + L_{\mathrm{FFN}} w_{\mathrm{FFN}}^2)$.

Lemma 4 is proved in Appendix E.5. Taking $\epsilon = 1/\sqrt{\Gamma}$ in (23), we obtain a bound

$$\mathbb{E}_{\mathfrak{S}} \sup_{\mathrm{T} \in \mathcal{T}} \left[ \mathbb{E}_{\mathfrak{s}} \left[ \mathcal{R}_n(\mathrm{T}(\mathfrak{s})) \right] - \mathcal{R}_{n,T}(\mathrm{T}) \right] \leq 2 \mathbb{E}_{\mathfrak{S}} \left[ \mathrm{Rad}(\mathcal{L} \circ \mathcal{T} \circ \mathfrak{S}) \right] \leq O \left( \frac{nD^3 \sqrt{\log(nD\Gamma/h)}}{\sqrt{\Gamma}} \right),$$

where the $O(\cdot)$ hides the dependency on some absolute constants.

**Bounding Error** II **in** (16). We can directly apply Hoeffding's inequality to bound this term, with details in Appendix E.9.

$$\mathbb{E}_{\mathfrak{S}} \left( \mathcal{R}_{n,\Gamma}(\mathrm{T}^*) - \mathbb{E}_{\mathfrak{s}} \left[ \mathcal{R}_n(\mathrm{T}^*(\mathfrak{s})) \right] \right) \leq O \left( \sqrt{\frac{\log(h^{-1})}{\Gamma}} + h^2 \right).$$

### C.3 PUTTING APPROXIMATION ERROR AND STATISTICAL ERROR TOGETHER

Putting all the error terms together in (16), we get

$$\mathcal{R}_n(\widehat{\mathrm{T}}) \leq \mathrm{I} + \mathrm{II} + \mathrm{III} \leq C_1 \left( \frac{nD^3 \sqrt{\log(nD\Gamma/h)}}{\sqrt{\Gamma}} \right) + C_2 \left( \frac{\left[ \log\left(h^{-1}\right) \right]^{1+\frac{3d}{2}}}{nh^d} + h^{2\alpha} [\log(h^{-1})]^2 \right).$$

Finally, choosing $h = n^{-\frac{1}{2\alpha+d}}$ gives rise to (15) in Theorem 1.

## D FUNDAMENTAL LEMMAS

In this section, we present some fundamental lemmas which are crucial for constructing a transformer to represent the target function. Note that similar results of Lemma 5 and 6 have appeared in Havrilla & Liao (2024), but our results are more general in the sense that they accommodate general $d_{embded}$ and general rows for gating. The detailed proofs of Lemma 5, 6, and 7 are provided in Appendix E.6, E.7, and E.8, respectively.

In the lemma and the proof, we use subscript to denote column index and superscript to denote row index. For a matrix $H$, we use the notation $\|H\|_\infty := \|H\|_{\infty,\infty} = \max_{i,j} |H_{ij}|$ to denote the infinity-infinity norm of a matrix $H$. When $\theta$ denotes the weight parameters of a neural network, we use $\|\theta\|_\infty$ to denote the largest magnitude in the weight parameters.

**Lemma 5** (Interaction Lemma). *Let $H = [h_t]_{1 \leq t \leq \ell} \in \mathbb{R}^{d_{embed} \times \ell}$ be an embedding matrix such that $h_t^{(d_{embed}-2):(d_{embed}-1)} = \mathcal{I}_t$ and $h_t^{d_{embed}} = 1$. Fix $1 \leq t_1, t_2 \leq \ell, 1 \leq i \leq d_{embed}$, and $\ell \in \mathbb{N}$. Suppose $d_{embed} \geq 5$ and $\|H\|_{\infty,\infty} < U$ for some $U > 0$, and the data kernels $Q^{data} \in \mathbb{R}^{(d_{embed}-3) \times d_{embed}}$ (the first $(d_{embed}-3)$ rows in the query matrix $Q$) and $K^{data} \in \mathbb{R}^{(d_{embed}-3) \times d_{embed}}$ (the first $(d_{embed}-3)$ rows in the key matrix $K$) satisfy $\max\{\|Q^{data}\|_{\infty,\infty}, \|K^{data}\|_{\infty,\infty}\} \leq \kappa$. Then one can construct an attention head $A$ with ReLU activation ($\sigma = \mathrm{ReLU}$) such that*

$$[A(H)]_t = \begin{cases} \sigma(\langle Q^{data} h_t, K^{data} h_{t_2} \rangle) \mathbf{e}_i & \text{if } t = t_1, \\ 0 & \text{otherwise.} \end{cases}$$

*The weight parameters of this attention head satisfies $\|\theta_A\|_\infty = O(d_{embed}^4 \kappa^2 \ell^2 U^2)$.*

Lemma 5 is proved in Appendix E.6. Lemma 5 is called an interaction lemma, which allows tokens to interact and therefore outputs a pair-wise interaction result.

**Lemma 6** (Gating Lemma). *Let $d_{embed} \geq 5$ and $H = [h_t]_{1 \leq t \leq \ell} \in \mathbb{R}^{d_{embed} \times \ell}$, be an embedding matrix such that $h_t^{(d_{embed}-2):(d_{embed}-1)} = (\mathcal{I}_t^1, \mathcal{I}_t^2) = \mathcal{I}_t$ and $h_t^{d_{embed}} = 1$. Then for any $r_1$ and $r_2$ with $1 \leq r_1 \leq r_2 \leq d_{embed} - 3$ and any $k_1, k_2$ with $1 \leq k_1, k_2 \leq \ell$, there exist both two-layer feed-forward networks (FFN) such that*

$$\mathrm{FFN}_1(h_t) = \begin{cases} h_t & \text{if } t \in \{1, \cdots, k_1\} \\ \begin{bmatrix} (h_t)_1 \\ \vdots \\ (h_t)_{r_1-1} \\ \mathbf{0} \\ (h_t)_{r_2+1} \\ \vdots \\ (h_t)_{d_{embed}-3} \\ \mathcal{I}_t^1 \\ \mathcal{I}_t^2 \\ 1 \end{bmatrix} & \text{otherwise} \end{cases} \tag{24}$$

*and*

$$\mathrm{FFN}_2(h_t) = \begin{cases} h_t & \text{if } t \in \{k_2, \cdots, \ell\} \\ \begin{bmatrix} (h_t)_1 \\ \vdots \\ (h_t)_{r_1-1} \\ \mathbf{0} \\ (h_t)_{r_2+1} \\ \vdots \\ (h_t)_{d_{embed}-3} \\ \mathcal{I}_t^1 \\ \mathcal{I}_t^2 \\ 1 \end{bmatrix} & \text{otherwise} \end{cases} \tag{25}$$

*Additionally, we have $\|\theta_{\mathrm{FFN}}\|_\infty \leq O(\ell \|H\|_{\infty,\infty})$.*

Lemma 6 is proved in E.7. Lemma 6 uses the feedforward layers to set certain rows in specified tokens zero.

**Lemma 7** (Decrementing Lemma). *Let $d_{embed} \geq 5$ and $H = [h_t]_{1 \leq t \leq \ell} \in \mathbb{R}^{d_{embed} \times \ell}$, be an embedding matrix such that $h_t^{(d_{embed}-2):(d_{embed}-1)} = (\mathcal{I}_t^1, \mathcal{I}_t^2) = \mathcal{I}_t$ and $h_t^{d_{embed}} = 1$. Then for any $r_1, r_2$ with $1 \leq r_1 \leq r_2 \leq d_{embed} - 3$ and any $k_1, k_2$ with $1 \leq k_1, k_2 \leq \ell$ and any $M > 0$, there*

*exists a six-layer residual feed-forward network* (FFN) *such that*

$$
\mathrm{FFN}(h_t) + h_t = \begin{cases} h_t & \text{if } t \in \{1, \cdots, k_1\} \cup \{k_2, \cdots, \ell\} \\ \begin{bmatrix} (h_t)_1 \\ \vdots \\ (h_t)_{r_1-1} \\ (h_t)_{r_1} - M \\ \vdots \\ (h_t)_{r_2} - M \\ (h_t)_{r_2+1} \\ \vdots \\ (h_t)_{d_{embed}-3} \\ \mathcal{I}_t \\ 1 \end{bmatrix} & \text{otherwise} \end{cases}
$$

*Additionally, we have* $\|\theta_{\mathrm{FFN}}\|_\infty \leq O(\ell M)$.

Lemma 7 is proved in Appendix E.8. Lemma 7 utilizes feedforward layers to substract $M$ from certain rows in specified tokens.

We next state the Dudley Entropy Integral (Dudley, 1967) which is used to derive (23), and refer its proof to Chen et al. (2020) and Van Der Vaart & Wellner (1996).

**Lemma 8** (Dudley Entropy Integral). *Let* $M > 0$. *Suppose* $\sup_{f \in \mathcal{F}} \|f\|_{L^\infty} \leq M$ *for some function class* $\mathcal{F}$. *Then*

$$
\mathbb{E}_{x,\xi} \left[ \sup_{f \in \mathcal{F}} \frac{1}{n} \sum_{i=1}^n \xi_i f(x_i) \right] \leq \inf_{\epsilon > 0} \left( 2\epsilon + \frac{12}{\sqrt{n}} \int_\epsilon^M \sqrt{\log \mathcal{N}(\delta, \mathcal{F}, \| \cdot \|_{L^\infty})} \, d\delta \right). \tag{26}
$$

*where* $\mathcal{N}(\delta, \mathcal{T}, \| \cdot \|_{L^\infty})$ *is the* $\delta$-*covering number of* $\mathcal{F}$ *with respect to* $L^\infty$ *norm.*

## E    DEFERRED PROOFS

### E.1    PROOF OF LEMMA 1

*Proof of Lemma 1.* First, we embed the sample $\mathfrak{s} = \{(\mathbf{x}_i, y_i)_{i=1}^n; \mathbf{x}_{n+1}\}$ into the embedding matrix $H$ such that

$$
\mathrm{PE} + \mathrm{E}(\mathfrak{s}) = H = \begin{bmatrix} \mathbf{x}_1 & \cdots & \mathbf{x}_n & \mathbf{x}_{n+1} & \mathbf{0} \\ y_1 & \cdots & y_n & 0 & \mathbf{0} \\ 0 & \cdots & \cdots & \cdots & 0 \\ \mathcal{I}_1 & \cdots & \cdots & \cdots & \mathcal{I}_{2n+1} \\ 1 & \cdots & \cdots & \cdots & 1 \end{bmatrix} \in \mathbb{R}^{d_{embed} \times \ell}.
$$

We denote the $i$-th column/token by $h_i$ in the following proof. Throughout the proof, we let $U = \|H\|_{\infty, \infty}$, which is the largest entry-wise magnitude of the matrix.

Next, let us demonstrate the construction of $\mathcal{K}_h(\mathfrak{s})$ step-by-step using our fundamental lemma in Appendix D.

● Copying of $(\mathbf{x}_{n+1})_i$, $1 \leq i \leq D$, to the next column (constant multiplication by 1).
Let us define each attention head $A_i$, $1 \leq i \leq D$, with $V_i = \mathbf{e}_i \mathbf{e}_{d_{embed}}^\top$, and data kernel in the form

$$
Q_i^{data} = \begin{bmatrix} 0 & & & & & 0 & 0 & 0 \\ & \ddots & & & & \vdots & \vdots & \vdots \\ & & \ddots & & & 0 & 0 & 1 \\ & & & 0 & & \vdots & \vdots & \vdots \\ & & & & 0 & 0 & 0 & 0 \\ & & & & 0 & 0 & 0 & 1 \end{bmatrix}, \quad K_i^{data} = \begin{bmatrix} 1 & & & & & 0 & 0 & 0 \\ & \ddots & & & & \vdots & \vdots & \vdots \\ & & \ddots & & & 0 & 0 & 0 \\ & & & 1 & & \vdots & \vdots & \vdots \\ & & & & 0 & 0 & 0 & 0 \\ & & & & 0 & 0 & 0 & M \end{bmatrix}
$$

where $Q_i^{data}, K_i^{data} \in \mathbb{R}^{(D+2)\times(D+5)}$. The $Q_i^{data}$ has the $i$-th position and last position of the last column equal to 1. By the Interaction Lemma 5, we can construct $A_i$, $1 \le i \le D$, such that $h_{n+2}$ interacts with $h_{n+1}$ only, i.e.,

$$
[A_i(H)]_{n+2} = \sigma \langle Q_i^{data} h_{n+2}, K_i^{data} h_{n+1} \rangle V_i h_{n+1} = \sigma((\mathbf{x}_{n+1})_i + M)\mathbf{e}_i = ((\mathbf{x}_{n+1})_i + M)\mathbf{e}_i
$$

and $[A_i(H)]_t = 0$ when $t \ne n+2$. Then the residual multi-head attention yields

$$
\mathrm{MHA}(H) + H = \begin{bmatrix} \mathbf{x}_1 & \cdots & \mathbf{x}_n & \mathbf{x}_{n+1} & \mathbf{x}_{n+1}+M & \mathbf{0} \\ y_1 & \cdots & y_n & 0 & 0 & \mathbf{0} \\ 0 & \cdots & \cdots & \cdots & \cdots & 0 \\ \mathcal{I}_1 & \cdots & \cdots & \cdots & \cdots & \mathcal{I}_{2n+1} \\ 1 & \cdots & \cdots & \cdots & \cdots & 1 \end{bmatrix}.
$$

Similarly, we can copy $(\mathbf{x}_{n+1})_i$, $1 \le i \le D$, to the $k$-th column for $k = n+2, \cdots, 2n+1$. This gives

$$
\mathrm{MHA}(H) + H = \begin{bmatrix} \mathbf{x}_1 & \cdots & \mathbf{x}_n & \mathbf{x}_{n+1} & \mathbf{x}_{n+1}+M & \cdots & \mathbf{x}_{n+1}+M \\ y_1 & \cdots & y_n & 0 & 0 & \cdots & 0 \\ 0 & \cdots & \cdots & \cdots & \cdots & \cdots & 0 \\ \mathcal{I}_1 & \cdots & \cdots & \cdots & \cdots & \cdots & \mathcal{I}_{2n+1} \\ 1 & \cdots & \cdots & \cdots & \cdots & \cdots & 1 \end{bmatrix}.
$$

Then we can apply Lemma 7 to subtract off the constant $M$ to get

$$
H_1 := \mathrm{B}_1(H) = \begin{bmatrix} \mathbf{x}_1 & \cdots & \mathbf{x}_n & \mathbf{x}_{n+1} & \mathbf{x}_{n+1} & \cdots & \mathbf{x}_{n+1} \\ y_1 & \cdots & y_n & 0 & 0 & \cdots & 0 \\ 0 & \cdots & \cdots & \cdots & \cdots & \cdots & 0 \\ \mathcal{I}_1 & \cdots & \cdots & \cdots & \cdots & \cdots & \mathcal{I}_{2n+1} \\ 1 & \cdots & \cdots & \cdots & \cdots & \cdots & 1 \end{bmatrix}.
$$

In total, this process needs $\mathrm{B}_1 \in \mathcal{B}(nD, 6, D+5)$. The upper bound of the weights parameter in $\mathrm{B}_1$ is $\|\theta_{\mathrm{B}_1}\|_\infty \le O(D^4 \ell^2 U^2 b^2)$.

• Implementation of $(\mathbf{x}_{n+1})_i - (\mathbf{x}_j)_i$ for $1 \le i \le D$ and $1 \le j \le n$

Let us define each attention head $A_{i,j}$ with $V_i = \mathbf{e}_i \mathbf{e}_{d_{embed}}^\top$, and data kernel in the form

$$
Q_i^{data} = \begin{bmatrix} 0 & & & & & 0 & 0 & 0 \\ & \ddots & & & & \vdots & \vdots & \vdots \\ & & \ddots & & & 0 & 0 & -1 \\ & & & 0 & & \vdots & \vdots & \vdots \\ & & & & 0 & 0 & 0 & 0 \\ & & & & 0 & 0 & 0 & 1 \end{bmatrix}, \quad K_i^{data} = \begin{bmatrix} 1 & & & & & 0 & 0 & 0 \\ & \ddots & & & & \vdots & \vdots & \vdots \\ & & \ddots & & & 0 & 0 & 0 \\ & & & 1 & & \vdots & \vdots & \vdots \\ & & & & 0 & 0 & 0 & 0 \\ & & & & 0 & 0 & 0 & M \end{bmatrix}
$$

where $Q_i^{data}, K_i^{data} \in \mathbb{R}^{(D+2)\times(D+5)}$. The $Q_i^{data}$ has the $i$-th position of last column equals to $-1$ and last position of the last column equals to 1. By the Interaction Lemma 5, we can construct $A_{i,j}$ such that $h_{n+1+j}$ interacts with $h_j$ only, i.e.,

$$A_{i,j}(h_{n+1+j}) = \sigma\langle Q_i^{data} h_{n+1+j}, K_i^{data} h_j \rangle V_i h_j = \sigma(-(\mathbf{x}_j)_i + M)\mathbf{e}_i = (-(\mathbf{x}_j)_i + M)\mathbf{e}_i$$

and $A_{i,j}(h_t) = 0$ for $t \neq n + 1 + j$, where $M \geq b \geq \|\mathbf{x}\|_\infty$. Then the residual multi-head attention yields

$$\text{MHA}(H_1) + H_1 = \begin{bmatrix} \mathbf{x}_1 & \cdots & \mathbf{x}_n & \mathbf{x}_{n+1} & \mathbf{x}_{n+1} - \mathbf{x}_1 + M & \cdots & \mathbf{x}_{n+1} - \mathbf{x}_n + M \\ y_1 & \cdots & y_n & 0 & \cdots & \cdots & 0 \\ 0 & \cdots & \cdots & \cdots & \cdots & \cdots & 0 \\ \mathcal{I}_1 & \cdots & \cdots & \cdots & \cdots & \cdots & \mathcal{I}_{2n+1} \\ 1 & \cdots & \cdots & \cdots & \cdots & \cdots & 1 \end{bmatrix}.$$

Then we can apply Lemma 7 to subtract off the constant $M$ to get

$$H_2 := \text{B}_2(H_1) = \begin{bmatrix} \mathbf{x}_1 & \cdots & \mathbf{x}_n & \mathbf{x}_{n+1} & \mathbf{x}_{n+1} - \mathbf{x}_1 & \cdots & \mathbf{x}_{n+1} - \mathbf{x}_n \\ y_1 & \cdots & y_n & 0 & \cdots & \cdots & 0 \\ 0 & \cdots & \cdots & \cdots & \cdots & \cdots & 0 \\ \mathcal{I}_1 & \cdots & \cdots & \cdots & \cdots & \cdots & \mathcal{I}_{2n+1} \\ 1 & \cdots & \cdots & \cdots & \cdots & \cdots & 1 \end{bmatrix}.$$

In total, this process needs $\text{B}_2 \in \mathcal{B}(Dn, 6, D+5)$. The upper bound of the weights parameter in $\text{B}_2$ is $\|\theta_{B_2}\| \leq O(D^4 \ell^2 U^2 b^2)$.

- Implementation of $-\frac{\|\mathbf{x}_{n+1} - \mathbf{x}_j\|^2}{h^2}$ for $1 \leq j \leq n$

Let us define each attention head $A_j$ with $V_j = \mathbf{e}_{D+1}\mathbf{e}_{d_{embed}}^\top$, and data kernel in the form

$$Q_j^{data} = \begin{bmatrix} -\frac{1}{h} & & & & 0 & 0 & 0 \\ & \ddots & & & \vdots & \vdots & \vdots \\ & & \ddots & & 0 & 0 & 0 \\ & & & -\frac{1}{h} & & \vdots & \vdots & \vdots \\ & & & & 0 & 0 & 0 & 0 \\ & & & & 0 & 0 & 0 & 1 \end{bmatrix}, \quad K_j^{data} = \begin{bmatrix} \frac{1}{h} & & & & 0 & 0 & 0 \\ & \ddots & & & \vdots & \vdots & \vdots \\ & & \ddots & & 0 & 0 & 0 \\ & & & \frac{1}{h} & & \vdots & \vdots & \vdots \\ & & & & 0 & 0 & 0 & 0 \\ & & & & 0 & 0 & 0 & M \end{bmatrix}$$

where $Q_j^{data}, K_j^{data} \in \mathbb{R}^{(D+2)\times(D+5)}$. By the Interaction Lemma 5, we can construct $A_j$ such that $h_{n+1+j}$ interacts with itself only, i.e.,

$$A_j(h_{n+1+j}) = \sigma(\langle Q_j^{data} h_{n+1+j}, K_j^{data} h_{n+1+j}\rangle) V_j h_{n+1+j}$$
$$= \sigma\left(-\frac{1}{h^2}\langle \mathbf{x}_{n+1} - \mathbf{x}_j, \mathbf{x}_{n+1} - \mathbf{x}_j\rangle + M\right)\mathbf{e}_{D+1} = \left(-\frac{\|\mathbf{x}_{n+1} - \mathbf{x}_j\|^2}{h^2} + M\right)\mathbf{e}_{D+1}$$

and $A_j(h_t) = 0$ for $t \neq n + 1 + j$, where $M \geq \max\left(\frac{4b^2 D}{h^2}, R\right)$. Then the residual multi-head attention yields

$$H_3 := \text{B}_3(H_2) = \text{MHA}(H_2) + H_2$$
$$= \begin{bmatrix} \mathbf{x}_1 & \cdots & \mathbf{x}_n & \mathbf{x}_{n+1} & -\frac{\|\mathbf{x}_{n+1} - \mathbf{x}_1\|^2}{h^2} + M & \cdots & -\frac{\|\mathbf{x}_{n+1} - \mathbf{x}_n\|^2}{h^2} + M \\ y_1 & \cdots & y_n & 0 & \cdots & \cdots & 0 \\ 0 & \cdots & \cdots & \cdots & \cdots & \cdots & 0 \\ \mathcal{I}_1 & \cdots & \cdots & \cdots & \cdots & \cdots & \mathcal{I}_{2n+1} \\ 1 & \cdots & \cdots & \cdots & \cdots & \cdots & 1 \end{bmatrix},$$

where $\text{B}_3 \in \mathcal{B}(n, 1, D+5)$. The upper bound of the weights parameter in $\text{B}_3$ is $\|\theta_{\text{B}_3}\|_\infty \leq O(D^4 \ell^2 U^2 M^2)$.

- copying $y_1, \cdots, y_n$ from columns $1, \cdots, n$ to columns $n+2, \cdots, 2n+1$

Similar as before, there exists $B_4 \in \mathcal{B}(n, 1, D+5)$ such that

$$\text{MHA}(H_3) + H_3 = \begin{bmatrix} \mathbf{x}_1 & \cdots & \mathbf{x}_n & \mathbf{x}_{n+1} & \mathbf{x}_{n+1} - \mathbf{x}_1 & \cdots & \mathbf{x}_{n+1} - \mathbf{x}_n \\ y_1 & \cdots & y_n & 0 & -\frac{\|\mathbf{x}_{n+1} - \mathbf{x}_1\|^2}{h^2} + M & \cdots & -\frac{\|\mathbf{x}_{n+1} - \mathbf{x}_n\|^2}{h^2} + M \\ 0 & \cdots & \cdots & \cdots & y_1 + M & \cdots & y_n + M \\ \mathcal{I}_1 & \cdots & \cdots & \cdots & \cdots & \cdots & \mathcal{I}_{2n+1} \\ 1 & \cdots & \cdots & \cdots & \cdots & \cdots & 1 \end{bmatrix}.$$

Then we can apply Lemma 7 to subtract off the constant $M \geq \max\left(\frac{4b^2 D}{h^2}, R\right)$ to get

$$H_4 := B_4(H_3) = \begin{bmatrix} \mathbf{x}_1 & \cdots & \mathbf{x}_n & \mathbf{x}_{n+1} & \mathbf{x}_{n+1} - \mathbf{x}_1 & \cdots & \mathbf{x}_{n+1} - \mathbf{x}_n \\ y_1 & \cdots & y_n & 0 & -\frac{\|\mathbf{x}_{n+1} - \mathbf{x}_1\|^2}{h^2} & \cdots & -\frac{\|\mathbf{x}_{n+1} - \mathbf{x}_n\|^2}{h^2} \\ 0 & \cdots & \cdots & \cdots & y_1 & \cdots & y_n \\ \mathcal{I}_1 & \cdots & \cdots & \cdots & \cdots & \cdots & \mathcal{I}_{2n+1} \\ 1 & \cdots & \cdots & \cdots & \cdots & \cdots & 1 \end{bmatrix}.$$

In total, this process needs $B_4 \in \mathcal{B}(n, 6, D+5)$. The upper bound of the weights parameter in $B_4$ is $\|\theta_{B_4}\|_\infty \leq O(D^4 \ell^2 M^2 \cdot M^2) = O(D^4 \ell^2 M^4)$.

- Implementation of $\mathcal{K}_h(\mathfrak{s})$

In the last layer of transformer, we use Softmax instead of ReLU for this part of construction (with a mask of size $n$). Then, we can construct $V = \mathbf{e}_{D+1} \mathbf{e}_{D+2}^\top$ and

$$Q^{data} = \begin{bmatrix} 0 & & & & & & 0 & 0 & 0 \\ & \ddots & & & & & \vdots & \vdots & \vdots \\ & & 0 & & & & 0 & 0 & 1 \\ & & & \ddots & & & 0 & 0 & 0 \\ & & & & 0 & & \vdots & \vdots & \vdots \\ & & & & & 0 & 0 & 0 & 0 \\ & & & & & 0 & 0 & 0 & 0 \end{bmatrix}, \quad K^{data} = \begin{bmatrix} 0 & & & & & & 0 & 0 & 0 \\ & \ddots & & & & & \vdots & \vdots & \vdots \\ & & 1 & & & & 0 & 0 & 0 \\ & & & \ddots & & & 0 & 0 & 0 \\ & & & & 0 & & \vdots & \vdots & \vdots \\ & & & & & 0 & 0 & 0 & 0 \\ & & & & & 0 & 0 & 0 & 0 \end{bmatrix}$$

where $Q^{data} \in \mathbb{R}^{(D+2) \times (D+5)}$ has $(D+1)$-th position in the last column equals to 1 and all the other entries are 0, and $K^{data} \in \mathbb{R}^{(D+2) \times (D+5)}$ has $(D+1, D+1)$-th position equals to 1 and all the other entries are 0, such that

$$[A(H_4)]_{n+1} = \sum_{j=n+2}^{2n+1} \text{softmax}\left(\langle Q^{data} h_{n+1}, K^{data} H_4 \rangle\right)_j V h_j$$

$$= \sum_{j=1}^{n} \frac{y_j e^{-\|\mathbf{x}_{n+1} - \mathbf{x}_j\|^2 / h^2}}{\sum_{j=1}^{n} e^{-\|\mathbf{x}_{n+1} - \mathbf{x}_j\|^2 / h^2}} \cdot \mathbf{e}_{D+1} = \mathcal{K}_h(\{\mathbf{x}_i, y_i\}_{i=1}^n; \mathbf{x}_{n+1}) \cdot \mathbf{e}_{D+1} = \mathcal{K}_h(\mathfrak{s}) \cdot \mathbf{e}_{D+1}.$$

Therefore, there exists $B_5 \in \mathcal{B}(1, 1, D+5)$ such that

$$H_5 := B_5(H_4) = \begin{bmatrix} \mathbf{x}_1 & \cdots & \mathbf{x}_n & \mathbf{x}_{n+1} & \mathbf{x}_{n+1} - \mathbf{x}_1 & \cdots & \mathbf{x}_{n+1} - \mathbf{x}_n \\ y_1 & \cdots & y_n & \mathcal{K}_h(\mathfrak{s}) & -\frac{\|\mathbf{x}_{n+1} - \mathbf{x}_1\|^2}{h^2} & \cdots & -\frac{\|\mathbf{x}_{n+1} - \mathbf{x}_n\|^2}{h^2} \\ 0 & \cdots & \cdots & \cdots & y_1 & \cdots & y_n \\ \mathcal{I}_1 & \cdots & \cdots & \cdots & \cdots & \cdots & \mathcal{I}_{2n+1} \\ 1 & \cdots & \cdots & \cdots & \cdots & \cdots & 1 \end{bmatrix}.$$

The upper bound of the weights parameter in $B_5$ is $\|\theta_{B_5}\|_\infty \leq O(D^4 \ell^2 M^2 \cdot 1) = O\left(D^4 \ell^2 M^2\right)$.

Finally, we apply a decoding layer DE to output the element $\mathcal{K}_h(\mathfrak{s})$ as desired. The uniform upper bound for the weight parameters in $B_5 \circ B_4 \circ B_3 \circ B_2 \circ B_1$ is $\kappa \leq O\left(D^4 \ell^2 M^4\right) \leq O\left(\frac{D^8 \ell^2 b^8 R^4}{h^8}\right) = O\left(\frac{D^8 n^2 b^8 R^4}{h^8}\right).$ □

## E.2 PROOF OF LEMMA 2

*Proof of Lemma 2.* Lemma 2 estimates the bias of kernel manifold regression. Our kernel estimator uses the Gaussian kernel, which has infinite support. To deal with the infinite support of the Gaussian kernel, we decompose the integral to nearby regions and far-away regions. For the $\mathbf{x}$ close to the center $\mathbf{x}_{n+1}$, we use the Lipchitz property of $f$ to estimate the bias; For the $\mathbf{x}$ far from the center $\mathbf{x}_{n+1}$, we use the Gaussian tail to bound the bias.

We first rewrite the bias in an integral form:

$$
\begin{aligned}
\bar{\mathcal{K}}_h(f; \mathbf{x}_{n+1}) - f(\mathbf{x}_{n+1}) &= \frac{\mathbb{E}\left[K_h(\mathbf{x}_{n+1} - \mathbf{x})f(\mathbf{x})\right]}{\mathbb{E}\left[K_h(\mathbf{x}_{n+1} - \mathbf{x})\right]} - f(\mathbf{x}_{n+1}) \\
&= \frac{\int K_h(\mathbf{x}_{n+1} - \mathbf{x})f(\mathbf{x})d\mathbf{x}}{\int K_h(\mathbf{x}_{n+1} - \mathbf{x})d\mathbf{x}} - f(\mathbf{x}_{n+1}) \\
&= \frac{\int K_h(\mathbf{x}_{n+1} - \mathbf{x})(f(\mathbf{x}) - f(\mathbf{x}_{n+1}))d\mathbf{x}}{\int K_h(\mathbf{x}_{n+1} - \mathbf{x})d\mathbf{x}}.
\end{aligned}
$$

We consider the set of points on the manifold $\mathcal{M}$ which is $\widetilde{h}$ distance to $\mathbf{x}_{n+1}$:

$$
B_{\widetilde{h}}(\mathbf{x}_{n+1}) := \{\mathbf{x} \in \mathcal{M} : \|\mathbf{x} - \mathbf{x}_{n+1}\| \leq \widetilde{h} = Ch\}.
$$

The choice of $\widetilde{h}$ will be specified later in the proof.

So we can write

$$
\begin{aligned}
\bar{\mathcal{K}}_h(f; \mathbf{x}_{n+1}) - f(\mathbf{x}_{n+1}) &= \frac{\int_{B_{\widetilde{h}}(\mathbf{x}_{n+1})} K_h(\mathbf{x}_{n+1} - \mathbf{x})(f(\mathbf{x}) - f(\mathbf{x}_{n+1}))d\mathbf{x}}{\int K_h(\mathbf{x}_{n+1} - \mathbf{x})d\mathbf{x}} \\
&\quad + \frac{\int_{\mathcal{M} \backslash B_{\widetilde{h}}(\mathbf{x}_{n+1})} K_h(\mathbf{x}_{n+1} - \mathbf{x})(f(\mathbf{x}) - f(\mathbf{x}_{n+1}))d\mathbf{x}}{\int K_h(\mathbf{x}_{n+1} - \mathbf{x})d\mathbf{x}} \\
&\leq \frac{\int_{B_{\widetilde{h}}(\mathbf{x}_{n+1})} K_h(\mathbf{x}_{n+1} - \mathbf{x})L(2\widetilde{h})^\alpha d\mathbf{x}}{\int K_h(\mathbf{x}_{n+1} - \mathbf{x})d\mathbf{x}} + \frac{2R \int_{\mathcal{M} \backslash B_{\widetilde{h}}(\mathbf{x}_{n+1})} K_h(\mathbf{x}_{n+1} - \mathbf{x})d\mathbf{x}}{\int K_h(\mathbf{x}_{n+1} - \mathbf{x})d\mathbf{x}} \\
&\leq 4L\widetilde{h}^\alpha + \frac{2R \int_{\mathcal{M} \backslash B_{\widetilde{h}}(\mathbf{x}_{n+1})} K_h(\mathbf{x}_{n+1} - \mathbf{x})d\mathbf{x}}{\int K_h(\mathbf{x}_{n+1} - \mathbf{x})d\mathbf{x}}.
\end{aligned}
$$

In the calculation above, we used the Lipchistz property of $f$ for the integral inside the ball $B_{\widetilde{h}}(\mathbf{x}_{n+1})$, where the geodesic distance and Euclidean distance are equivalent metrics. By Proposition 11 in (Maggioni et al., 2016), when $\|\mathbf{x}_{n+1} - \mathbf{x}\| \leq \tau_{\mathcal{M}}/2$, we have $d_{\mathcal{M}}(\mathbf{x}_{n+1}, \mathbf{x}) \leq 2\|\mathbf{x}_{n+1} - \mathbf{x}\|$.

We next bound the integral outside the ball $B_{\widetilde{h}}(\mathbf{x}_{n+1})$. When $h$ is small, i.e. $h < \tau_{\mathcal{M}}/2$, the integral satisfies

$$
\begin{aligned}
\int_{\mathcal{M}} h^{-d} K_h(\mathbf{x}_{n+1} - \mathbf{x})d\mathbf{x} &= \int_{\mathcal{M}} h^{-d} e^{-\frac{\|\mathbf{x}_{n+1} - \mathbf{x}\|^2}{h^2}} d\mathbf{x} \geq \int_{B_h(\mathbf{x}_{n+1})} h^{-d} e^{-\frac{\|\mathbf{x}_{n+1} - \mathbf{x}\|^2}{h^2}} d\mathbf{x} \\
&\geq \int_{B_h(\mathbf{x}_{n+1})} h^{-d} e^{-1} d\mathbf{x} = h^{-d} e^{-1} C_B h^d = e^{-1} C_B
\end{aligned}
$$

with

$$
C_B \geq \cos^d(\arcsin(\frac{h}{2\tau})) \geq \cos^d(\arcsin(\frac{1}{4})), \tag{27}
$$

according to (Niyogi et al., 2008, Lemma 5.3).

Therefore, when $h < \tau/2$, the integral outside the ball $B_{\widetilde{h}}(\mathbf{x}_{n+1})$ can be bounded as follows:

$$
\begin{aligned}
\frac{2R \int_{\mathcal{M} \backslash B_{\widetilde{h}}(\mathbf{x}_{n+1})} K_h(\mathbf{x}_{n+1} - \mathbf{x}) d\mathbf{x}}{\int_{\mathcal{M}} K_h(\mathbf{x}_{n+1} - \mathbf{x}) d\mathbf{x}} &= \frac{2R \int_{\mathcal{M} \backslash B_{\widetilde{h}}(\mathbf{x}_{n+1})} h^{-d} K_h(\mathbf{x}_{n+1} - \mathbf{x}) d\mathbf{x}}{\int_{\mathcal{M}} h^{-d} K_h(\mathbf{x}_{n+1} - \mathbf{x}) d\mathbf{x}} \\
&\leq 2eRC_B^{-1} \int_{\mathcal{M} \backslash B_{\widetilde{h}}(\mathbf{x}_{n+1})} h^{-d} K_h(\mathbf{x}_{n+1} - \mathbf{x}) d\mathbf{x} \\
&\leq 2eRC_B^{-1} \cdot (\rho_{\mathbf{x}}(\mathcal{M}) - \rho_{\mathbf{x}}(B_{\widetilde{h}}(\mathbf{x}_{n+1}))) \cdot h^{-d} e^{-\frac{\widetilde{h}^2}{h^2}} \\
&\xlongequal{\text{let } \widetilde{h}=Ch} 2eRC_B^{-1} \cdot (\rho_{\mathbf{x}}(\mathcal{M}) - \rho_{\mathbf{x}}(B_{\widetilde{h}}(\mathbf{x}_{n+1}))) \cdot h^{-d} e^{-C^2} \\
&\leq 2eRC_B^{-1} \cdot h^{-d} e^{-C^2} \\
&= O(h^{-d} e^{-C^2}),
\end{aligned}
$$

where $O$ hides constants about $R$ and $d$.

Hence

$$
\bar{\mathcal{K}}_h(f; \mathbf{x}_{n+1}) - f(\mathbf{x}_{n+1}) \leq O(C^\alpha h^\alpha) + O(h^{-d} e^{-C^2}).
$$

For $h \in (0, 1)$, by choosing $C = \sqrt{(d+1) \log(\frac{1}{h})}$, we have $h^{-d} e^{-C^2} = h$. Therefore,

$$
\bar{\mathcal{K}}_h(f; \mathbf{x}_{n+1}) - f(\mathbf{x}_{n+1}) \leq O\left(h^\alpha \log\left(\frac{1}{h}\right)\right) + O(h) = O\left(h^\alpha \log\left(\frac{1}{h}\right)\right).
$$

as desired. The notation $O(\cdot)$ hides the constants depending on $d, L, R, \tau_{\mathcal{M}}$.

$\square$

### E.3 PROOF OF LEMMA 3

*Proof of Lemma 3.* Lemma 3 estimates the variance of kernel manifold regression. We prove it using a series of concentration inequalities (Hoeffding, 1994; Vershynin, 2018).

Let us define some empirical quantities used in kernel estimator and their counterparts in expectation.

$$
\widehat{N}_n(\mathbf{x}_{n+1}) := \frac{1}{n} \sum_{i=1}^n K_h(\mathbf{x}_{n+1} - \mathbf{x}_i) f(\mathbf{x}_i), \quad \widehat{D}_n(\mathbf{x}_{n+1}) := \frac{1}{n} \sum_{i=1}^n K_h(\mathbf{x}_{n+1} - \mathbf{x}_i)
$$

$$
N(\mathbf{x}_{n+1}) := \mathbb{E}_{\mathbf{x}}[K_h(\mathbf{x}_{n+1} - \mathbf{x}) f(\mathbf{x})], \quad D(\mathbf{x}_{n+1}) := \mathbb{E}_{\mathbf{x}}[K_h(\mathbf{x}_{n+1} - \mathbf{x})]
$$

We first decompose the variance as follows:

$$
\begin{aligned}
&|\mathcal{K}_h(\mathfrak{s}) - \bar{\mathcal{K}}_h(f; \mathbf{x}_{n+1})| \\
&\leq \frac{1}{|\widehat{D}_n(\mathbf{x}_{n+1})||D(\mathbf{x}_{n+1})|} \left(|D(\mathbf{x}_{n+1})||\widehat{N}_n(\mathbf{x}_{n+1}) - N(\mathbf{x}_{n+1})| + |N(\mathbf{x}_{n+1})||\widehat{D}_n(\mathbf{x}_{n+1}) - D(\mathbf{x}_{n+1})|\right) \\
&= \frac{1}{|\widehat{D}_n(\mathbf{x}_{n+1})|} \left(|\widehat{N}_n(\mathbf{x}_{n+1}) - N(\mathbf{x}_{n+1})|\right) + \frac{|N(\mathbf{x}_{n+1})|}{|\widehat{D}_n(\mathbf{x}_{n+1})| \cdot |D(\mathbf{x}_{n+1})|} \left(|\widehat{D}_n(\mathbf{x}_{n+1}) - D(\mathbf{x}_{n+1})|\right).
\end{aligned}
\tag{28}
$$

We will bound (28) in the following steps.

• **Estimating $\widehat{D}_n(\mathbf{x}_{n+1})$ in the denominator.** We consider the following ball

$$
B_{\widetilde{h}} := B_{\widetilde{h}}(\mathbf{x}_{n+1}) := \{\mathbf{x} \in \mathcal{M} : \|\mathbf{x} - \mathbf{x}_{n+1}\| \leq \widetilde{h}\}
\tag{29}
$$

with $\widetilde{h} = Ch$. Let $n_B$ be the number of samples in $B_{\widetilde{h}}(\mathbf{x}_{n+1})$. By Liao & Maggioni (2019, Lemma 30), we can estimate $n_B$ as follows:

$$
\mathbb{P}\left\{ \left| \frac{n_B}{n} - \rho_{\mathbf{x}}(B_{\widetilde{h}}(\mathbf{x}_{n+1})) \right| \geq \frac{1}{2} \rho_{\mathbf{x}}(B_{\widetilde{h}}(\mathbf{x}_{n+1})) \right\} \leq 2e^{-\frac{3n \cdot \rho_{\mathbf{x}}(B_{\widetilde{h}}(\mathbf{x}_{n+1}))}{28}},
$$

where $\rho_{\mathbf{x}}(B_{\widetilde{h}}(\mathbf{x}_{n+1})) = C_B \widetilde{h}^d = C_B C^d h^d$, for some constant $C_B$ which satisfies (27). Therefore, with probability at least $1 - 2e^{-\frac{3n \cdot \rho_{\mathbf{x}}(B_{\widetilde{h}}(\mathbf{x}_{n+1}))}{28}}$, it holds

$$\frac{1}{2}\rho_{\mathbf{x}}(B_{\widetilde{h}}(\mathbf{x}_{n+1})) \leq \frac{n_B}{n} \leq \frac{3}{2}\rho_{\mathbf{x}}(B_{\widetilde{h}}(\mathbf{x}_{n+1})).$$

We next re-write $\widehat{D}_n(\mathbf{x}_{n+1})$ as

$$\widehat{D}_n(\mathbf{x}_{n+1}) = \frac{n_B}{n} \cdot \frac{1}{n_B} \sum_{i=1}^{n} K_h(\mathbf{x}_{n+1} - \mathbf{x}_i) \geq \frac{n_B}{n} \cdot \frac{1}{n_B} \sum_{\mathbf{x}_i \in B_{\widetilde{h}}(\mathbf{x}_{n+1})} K_h(\mathbf{x}_{n+1} - \mathbf{x}_i)$$

where the continuous counterpart of $\frac{1}{n_B} \sum_{\mathbf{x}_i \in B_{\widetilde{h}}(\mathbf{x}_{n+1})} K_h(\mathbf{x}_{n+1} - \mathbf{x}_i)$ is

$$\Phi := \frac{1}{\rho_{\mathbf{x}}(B_{\widetilde{h}})} \int_{B_{\widetilde{h}}} K_h(\mathbf{x} - \mathbf{x}_{n+1}) d\mathbf{x} \geq \frac{1}{\rho_{\mathbf{x}}(B_{\widetilde{h}})} \int_{B_h} K_h(\mathbf{x} - \mathbf{x}_{n+1}) d\mathbf{x}$$

$$\geq \frac{e^{-1}\rho_{\mathbf{x}}(B_h)}{\rho_{\mathbf{x}}(B_{\widetilde{h}})} \geq C_\Phi C^{-d}$$

where $C_\Phi$ is a constant depending on $\tau_{\mathcal{M}}$.

By the Hoeffding's inequality (Hoeffding, 1994), with probability $1 - \delta$, $\widehat{D}_n(\mathbf{x}_{n+1})$ satisfies

$$\widehat{D}_n(\mathbf{x}_{n+1}) \geq \frac{n_B}{n} \cdot \left( \Phi - \sqrt{\frac{\log(2/\delta)}{n_B}} \right) \geq \frac{n_B}{n} \left( C_\Phi C^{-d} - \sqrt{\frac{\log(2/\delta)}{n_B}} \right).$$

**Bounding the first term in** (28). The numerator of the first term can be decomposed as

$$\left| \widehat{N}_n(\mathbf{x}_{n+1}) - N(\mathbf{x}_{n+1}) \right| \leq \left| \widehat{N}_n^{(1)}(\mathbf{x}_{n+1}) - N^{(1)}(\mathbf{x}_{n+1}) \right| + \left| \widehat{N}_n^{(2)}(\mathbf{x}_{n+1}) - N^{(2)}(\mathbf{x}_{n+1}) \right|,$$

where

$$\widehat{N}_n^{(1)}(\mathbf{x}_{n+1}) = \frac{1}{n} \sum_{\mathbf{x}_i \in B_{\widetilde{h}}(\mathbf{x}_{n+1})} K_h(\mathbf{x}_{n+1} - \mathbf{x}_i) f(\mathbf{x}_i)$$

and

$$N^{(1)}(\mathbf{x}_{n+1}) = \int_{\mathbf{x} \in B_{\widetilde{h}}(\mathbf{x}_{n+1})} K_h(\mathbf{x}_{n+1} - \mathbf{x}) f(\mathbf{x}) d\mathbf{x}$$

and

$$\widehat{N}_n^{(2)}(\mathbf{x}_{n+1}) = \frac{1}{n} \sum_{\mathbf{x}_i \in \mathcal{M} \setminus B_{\widetilde{h}}(\mathbf{x}_{n+1})} K_h(\mathbf{x}_{n+1} - \mathbf{x}_i) f(\mathbf{x}_i)$$

and

$$N^{(2)}(\mathbf{x}_{n+1}) = \int_{\mathbf{x} \in \mathcal{M} \setminus B_{\widetilde{h}}(\mathbf{x}_{n+1})} K_h(\mathbf{x}_{n+1} - \mathbf{x}) f(\mathbf{x}) d\mathbf{x}.$$

Therefore, we just need to bound

$$\frac{\left| \widehat{N}_n^{(1)}(\mathbf{x}_{n+1}) - N^{(1)}(\mathbf{x}_{n+1}) \right| + \left| \widehat{N}_n^{(2)}(\mathbf{x}_{n+1}) - N^{(2)}(\mathbf{x}_{n+1}) \right|}{\widehat{D}_n(\mathbf{x}_{n+1})}$$

$$\leq \frac{\left| \widehat{N}_n^{(1)}(\mathbf{x}_{n+1}) - N^{(1)}(\mathbf{x}_{n+1}) \right|}{\frac{n_B}{n} \left( C_\Phi C^{-d} - \sqrt{\frac{\log(2/\delta)}{n_B}} \right)} + \frac{\left| \widehat{N}_n^{(2)}(\mathbf{x}_{n+1}) - N^{(2)}(\mathbf{x}_{n+1}) \right|}{\frac{n_B}{n} \left( C_\Phi C^{-d} - \sqrt{\frac{\log(2/\delta)}{n_B}} \right)} \qquad (30)$$

The first term in (30) can be written as

$$
\frac{\left| \widehat{N}_n^{(1)}(\mathbf{x}_{n+1}) - N^{(1)}(\mathbf{x}_{n+1}) \right|}{\frac{n_B}{n} \left( C_\Phi C^{-d} - \sqrt{\frac{\log(2/\delta)}{n_B}} \right)}
$$

$$
= \frac{\left| \frac{1}{n_B} \sum_{\mathbf{x}_i \in B_{\widetilde{h}}(\mathbf{x}_{n+1})} K_h\left(\mathbf{x}_{n+1} - \mathbf{x}_i\right) f(\mathbf{x}_i) - \frac{n}{n_B} \int_{\mathbf{x} \in B_{\widetilde{h}}(\mathbf{x}_{n+1})} K_h(\mathbf{x}_{n+1} - \mathbf{x}) f(\mathbf{x})d\mathbf{x} \right|}{C_\Phi C^{-d} - \sqrt{\frac{\log(2/\delta)}{n_B}}}
$$

$$
\leq \frac{\left| \frac{1}{n_B} \sum_{\mathbf{x}_i \in B_{\widetilde{h}}(\mathbf{x}_{n+1})} K_h\left(\mathbf{x}_{n+1} - \mathbf{x}_i\right) f(\mathbf{x}_i) - \frac{1}{\rho_{\mathbf{x}}(B_{\widetilde{h}})} \int_{\mathbf{x} \in B_{\widetilde{h}}(\mathbf{x}_{n+1})} K_h(\mathbf{x}_{n+1} - \mathbf{x}) f(\mathbf{x})d\mathbf{x} \right|}{C_\Phi C^{-d} - \sqrt{\frac{\log(2/\delta)}{n_B}}}
$$

$$
+ \frac{\left| \frac{1}{\rho_{\mathbf{x}}(B_{\widetilde{h}})} \int_{\mathbf{x} \in B_{\widetilde{h}}(\mathbf{x}_{n+1})} K_h(\mathbf{x}_{n+1} - \mathbf{x}) f(\mathbf{x})d\mathbf{x} - \frac{n}{n_B} \int_{\mathbf{x} \in B_{\widetilde{h}}(\mathbf{x}_{n+1})} K_h(\mathbf{x}_{n+1} - \mathbf{x}) f(\mathbf{x})d\mathbf{x} \right|}{C_\Phi C^{-d} - \sqrt{\frac{\log(2/\delta)}{n_B}}}.
$$

By the Hoeffding's inequality, we know that with probability at least $1 - \delta$,

$$
\left| \frac{1}{n_B} \sum_{\mathbf{x}_i \in B_{\widetilde{h}}(\mathbf{x}_{n+1})} K_h\left(\mathbf{x}_{n+1} - \mathbf{x}_i\right) f(\mathbf{x}_i) - \frac{1}{\rho_{\mathbf{x}}(B_{\widetilde{h}})} \int_{\mathbf{x} \in B_{\widetilde{h}}(\mathbf{x}_{n+1})} K_h(\mathbf{x}_{n+1} - \mathbf{x}) f(\mathbf{x})d\mathbf{x} \right| \leq R\sqrt{\frac{\log(2/\delta)}{n_B}}.
$$

By Liao & Maggioni (2019, Lemma 30), with probability $1 - \delta$,

$$
\left| \rho_{\mathbf{x}}(B_{\widetilde{h}}(\mathbf{x}_{n+1})) - \frac{n_B}{n} \right| \leq O\left( \sqrt{\frac{\log(2/\delta)\rho_{\mathbf{x}}(B_{\widetilde{h}}(\mathbf{x}_{n+1}))}{n}} \right)
$$

which gives rise to

$$
\left| \frac{1}{\rho_{\mathbf{x}}(B_{\widetilde{h}})} \int_{\mathbf{x} \in B_{\widetilde{h}}(\mathbf{x}_{n+1})} K_h(\mathbf{x}_{n+1} - \mathbf{x}) f(\mathbf{x})d\mathbf{x} - \frac{n}{n_B} \int_{\mathbf{x} \in B_{\widetilde{h}}(\mathbf{x}_{n+1})} K_h(\mathbf{x}_{n+1} - \mathbf{x}) f(\mathbf{x})d\mathbf{x} \right|
$$

$$
\leq \left| \frac{1}{\rho_{\mathbf{x}}(B_{\widetilde{h}})} - \frac{n}{n_B} \right| \cdot \left| \int_{\mathbf{x} \in B_{\widetilde{h}}(\mathbf{x}_{n+1})} K_h(\mathbf{x}_{n+1} - \mathbf{x}) f(\mathbf{x})d\mathbf{x} \right| \leq R\rho_{\mathbf{x}}(B_{\widetilde{h}}) \left| \frac{1}{\rho_{\mathbf{x}}(B_{\widetilde{h}})} - \frac{n}{n_B} \right|
$$

$$
= R\left| 1 - \frac{n}{n_B}\rho_{\mathbf{x}}(B_{\widetilde{h}}) \right| = R\frac{n}{n_B}\left| \frac{n_B}{n} - \rho_{\mathbf{x}}(B_{\widetilde{h}}) \right| \leq O\left( \frac{R}{\rho_{\mathbf{x}}(B_{\widetilde{h}}(\mathbf{x}_{n+1}))} \sqrt{\frac{\log(2/\delta)\rho_{\mathbf{x}}(B_{\widetilde{h}}(\mathbf{x}_{n+1}))}{n}} \right)
$$

$$
= O\left( R\sqrt{\frac{\log(2/\delta)}{n\rho_{\mathbf{x}}(B_{\widetilde{h}}(\mathbf{x}_{n+1}))}} \right).
$$

Therefore, with probability at least $1 - 2\delta$, the first term in (30) satisfies

$$
\frac{\left| \widehat{N}_n^{(1)}(\mathbf{x}_{n+1}) - N^{(1)}(\mathbf{x}_{n+1}) \right|}{\widehat{D}_n(\mathbf{x}_{n+1})} \leq O\left( RC^d\sqrt{\frac{\log(2/\delta)}{n_B}} \right) + O\left( RC^d\sqrt{\frac{\log(2/\delta)}{n\rho_{\mathbf{x}}(B_{\widetilde{h}}(\mathbf{x}_{n+1}))}} \right)
$$

$$
= O\left( RC^d\sqrt{\frac{\log(2/\delta)}{n\rho_{\mathbf{x}}(B_{\widetilde{h}}(\mathbf{x}_{n+1}))}} \right) = O\left( \frac{RC^{d/2}}{\sqrt{C_B}}\sqrt{\frac{\log(2/\delta)}{nh^d}} \right),
$$

$$
\tag{31}
$$

where the constant $C_B$ satisfies (27).

For the second term in (30), it satisfies

$$\frac{\left|\widehat{N}_n^{(2)}(\mathbf{x}_{n+1}) - N^{(2)}(\mathbf{x}_{n+1})\right|}{\widehat{D}_n(\mathbf{x}_{n+1})}$$

$$\leq \frac{\left|\frac{1}{n}\sum_{\mathbf{x}_i \in \mathcal{M}\setminus B_{\widetilde{h}}(\mathbf{x}_{n+1})} K_h(\mathbf{x}_{n+1} - \mathbf{x}_i) f(\mathbf{x}_i) - \int_{\mathbf{x} \in \mathcal{M}\setminus B_{\widetilde{h}}(\mathbf{x}_{n+1})} K_h(\mathbf{x}_{n+1} - \mathbf{x}) f(\mathbf{x}) d\mathbf{x}\right|}{\frac{n_B}{n}\left(C_\Phi C^{-d} - \sqrt{\frac{\log(2/\delta)}{n_B}}\right)}$$

By the Hoeffding's inequality, we have, with probability at least $1 - \delta$,

$$\left|\frac{1}{n}\sum_{\mathbf{x}_i \in \mathcal{M}\setminus B_{\widetilde{h}}(\mathbf{x}_{n+1})} K_h(\mathbf{x}_{n+1} - \mathbf{x}_i) f(\mathbf{x}_i) - \int_{\mathbf{x} \in \mathcal{M}\setminus B_{\widetilde{h}}(\mathbf{x}_{n+1})} K_h(\mathbf{x}_{n+1} - \mathbf{x}) f(\mathbf{x}) d\mathbf{x}\right| \leq Re^{-C^2}\sqrt{\frac{\log(2/\delta)}{n}},$$

where we bound $|K_h(\mathbf{x}_{n+1} - \mathbf{x}) f(\mathbf{x})| \leq Re^{-C^2}$ for all $\mathbf{x} \in \mathcal{M}\setminus B_{\widetilde{h}}$. Therefore, the second term in (30) can be further bounded as

$$\frac{\left|\widehat{N}_n^{(2)}(\mathbf{x}_{n+1}) - N^{(2)}(\mathbf{x}_{n+1})\right|}{\widehat{D}_n(\mathbf{x}_{n+1})} \leq \frac{Re^{-C^2}\sqrt{\frac{\log(2/\delta)}{n}}}{\frac{n_B}{n}\left(C_\Phi C^{-d} - \sqrt{\frac{\log(2/\delta)}{n_B}}\right)} \leq O\left(C^d e^{-C^2} \frac{n}{n_B}\sqrt{\frac{\log(2/\delta)}{n}}\right)$$

$$= O\left(\frac{e^{-C^2}}{h^{d/2}}\sqrt{\frac{\log(2/\delta)}{nh^d}}\right). \tag{32}$$

In summary, the first term in (28) can be split into two terms according to (30): one term is inside the ball $B_{\widetilde{h}}$ and the other is outside the ball. We have bounded the term inside the ball $B_{\widetilde{h}}$ in (31) and the term outside the ball in (32). Combining (31) and (32) gives rise to

$$\frac{|\widehat{N}_n(\mathbf{x}_{n+1}) - N(\mathbf{x}_{n+1})|}{|\widehat{D}_n(\mathbf{x}_{n+1})|} \leq O\left(\frac{RC^{d/2}}{\sqrt{C_B}}\sqrt{\frac{\log(2/\delta)}{nh^d}}\right) + O\left(\frac{e^{-C^2}}{h^{d/2}}\sqrt{\frac{\log(2/\delta)}{nh^d}}\right)$$

$$= O\left(\left[\log\left(h^{-1}\right)\right]^{\frac{d}{4}}\sqrt{\frac{\log(2/\delta)}{nh^d}}\right).$$

where the last line results from choosing

$$C = \sqrt{d\log(1/h)}, \tag{33}$$

so that $e^{-C^2} = h^d < h^{d/2}$ when $h$ is small.

**Bounding the second term in** (28). The second term in (28) can be bounded similarly to the first term, with an additional estimate on $\frac{N(\mathbf{x}_{n+1})}{D(\mathbf{x}_{n+1})}$. We define the ball $B_h(\mathbf{x}_{n+1})$ and $B_{\widetilde{h}}(\mathbf{x}_{n+1})$ as in (29) with $\widetilde{h} = Ch$.

$$\frac{|N(\mathbf{x}_{n+1})|}{|D(\mathbf{x}_{n+1})|} = \frac{\left|\int_{\mathcal{M}} K_h(\mathbf{x}_{n+1} - \mathbf{x}) f(\mathbf{x}) d\mathbf{x}\right|}{\left|\int_{\mathcal{M}} K_h(\mathbf{x}_{n+1} - \mathbf{x}) d\mathbf{x}\right|}$$

$$\leq \frac{\left|\int_{B_{\widetilde{h}}(\mathbf{x}_{n+1})} K_h(\mathbf{x}_{n+1} - \mathbf{x}) f(\mathbf{x}) d\mathbf{x}\right| + \left|\int_{\mathcal{M}\setminus B_{\widetilde{h}}(\mathbf{x}_{n+1})} K_h(\mathbf{x}_{n+1} - \mathbf{x}) f(\mathbf{x}) d\mathbf{x}\right|}{\left|\int_{B_h(\mathbf{x}_{n+1})} K_h(\mathbf{x}_{n+1} - \mathbf{x}) d\mathbf{x}\right|}$$

$$\leq \frac{R\rho_{\mathbf{x}}(B_{\widetilde{h}}(\mathbf{x}_{n+1})) + Re^{-C^2}\rho_{\mathbf{x}}(\mathcal{M})}{e^{-1}\rho_{\mathbf{x}}(B_h(\mathbf{x}_{n+1}))} \leq O(RC^d),$$

where the last inequality holds with $C$ chosen according to (33) and when $h$ is sufficiently small.

Applying a similar argument above, the second term in (28) can be bounded by

$$\frac{|N(\mathbf{x}_{n+1})|}{|\widehat{D}_n(\mathbf{x}_{n+1})| \cdot |D(\mathbf{x}_{n+1})|}\left(|\widehat{D}_n(\mathbf{x}_{n+1}) - D(\mathbf{x}_{n+1})|\right) \leq O\left(\log^{3d/4}\left(\frac{1}{h}\right)\sqrt{\frac{\log(2/\delta)}{nh^d}}\right)$$

**Putting the two terms in** (28) **together.** Putting the two terms in in (28) together, for $\delta > 2e^{-\frac{3n \cdot \rho_{\mathbf{x}}(B_{\tilde{h}}(\mathbf{x}_{n+1}))}{28}}$, we have with probability at least $1 - 2\delta$,

$$|\mathcal{K}_h(\mathfrak{s}) - \bar{\mathcal{K}}_h(f; \mathbf{x}_{n+1})| \leq O\left( \log^{3d/4}\left(\frac{1}{h}\right) \sqrt{\frac{\log(2/\delta)}{nh^d}} \right).$$

By abusing the notation, rewrite $2\delta$ as $\delta$, we get with at least probability $1 - \delta$,

$$|\mathcal{K}_h(\mathfrak{s}) - \bar{\mathcal{K}}_h(f; \mathbf{x}_{n+1})| \leq O\left( \log^{3d/4}\left(\frac{1}{h}\right) \sqrt{\frac{\log(4/\delta)}{nh^d}} \right).$$

as desired. The notation $O(\cdot)$ hides constants depending on $d, R, \tau_{\mathcal{M}}$. $\qquad\square$

### E.4 PROOF OF PROPOSITION 1

*Proof of Proposition 1.* By Lemma 1, 2, 3 and equation (19),

$$\mathbb{E}_{\mathfrak{s}}\left[\mathcal{R}_n(\mathrm{T}^*(\mathfrak{s}))\right] \leq (1 - \delta) \cdot O\left( \left[\log\left(h^{-1}\right)\right]^{3d/2} \frac{\log(4/\delta)}{nh^d} \right) + \delta \cdot (2R)^2 + O(h^{2\alpha}[\log(h^{-1})]^2)$$

$$\xlongequal{\text{let } \delta = 4h^2} (1 - 4h^2) \cdot O\left( \left[\log\left(h^{-1}\right)\right]^{3d/2} \frac{\log(h^{-1})}{nh^d} \right) + 16h^2R^2 + O(h^{2\alpha}[\log(h^{-1})]^2)$$

$$\leq O\left( \frac{\left[\log\left(h^{-1}\right)\right]^{1+3d/2}}{nh^d} \right) + O(h^{2\alpha}[\log(h^{-1})]^2).$$

The last inequality holds because $0 < h < 1$. The notation $O(\cdot)$ hides the constants depending on $d, L, R, \tau_{\mathcal{M}}$. $\qquad\square$

### E.5 PROOF OF LEMMA 4

*Proof of Lemma 4.* Through the proof, we use the notation $\|H\|_\infty := \|H\|_{\infty,\infty}$ to denote the infinity-infinity norm of a matrix $H$.

Since our transformer has softmax as activation function in the last layer and ReLU as activation from the first to the penultimate layers, we need to consider those two cases separately.

Set $\eta > 0$, we choose T with parameters $\theta$, and T' with parameters $\theta'$ such that $\|\theta - \theta'\|_\infty \leq \eta$.

We first bound the Multi-head Attention (MHA) layer in a transformer block. For the ReLU activation layer, according to Havrilla & Liao (2024, Lemma 2), for $\|H\|_\infty \leq U$, we have

$$\|\mathrm{MHA}_1(H) - \mathrm{MHA}_2(H)\|_\infty^{\mathrm{ReLU}} \leq 3\kappa^3 d_{embed}^6 U^3 m_{\mathrm{T}} \ell \eta.$$

By the similar argument, for the softmax activation layer, since it takes normalization, we can bound

$$\|\mathrm{MHA}_1(H) - \mathrm{MHA}_2(H)\|_\infty^{\mathrm{softmax}} \leq \|\mathrm{MHA}_1(H) - \mathrm{MHA}_2(H)\|_\infty^{\mathrm{ReLU}} \leq 3\kappa^3 d_{embed}^6 U^3 m_{\mathrm{T}} \ell \eta.$$

Therefore, for the MHA layer, we have

$$\|\mathrm{MHA}_1(H) - \mathrm{MHA}_2(H)\|_\infty \leq 3\kappa^3 d_{embed}^6 U^3 m_{\mathrm{T}} \ell \eta.$$

Next, we bound the FFN layer. According to Havrilla & Liao (2024, Lemma 2), we have

$$\|\mathrm{FFN}_1(H + \mathrm{MHA}_1(H)) - \mathrm{FFN}_2(H + \mathrm{MHA}_2(\mathrm{H}))\|_\infty$$
$$\leq 3\kappa^{3+L_{\mathrm{FFN}}} w_{\mathrm{FFN}}^{2L_{\mathrm{FFN}}} d_{embed}^6 U^3 m_{\mathrm{T}} \ell \eta + L_{\mathrm{FFN}}(w_{\mathrm{FFN}}(2d_{embed}^6 \kappa^3 U m_{\mathrm{T}} \ell) + 2)(\kappa w_{\mathrm{FFN}})^{L_{\mathrm{FFN}}-1} \eta.$$

Therefore, putting together the MHA and FFN layer together, we get the estimate on the difference of the transformer block $\|B_1(H) - B_2(H)\|_\infty$ ( for both ReLU and softmax activation) as

$$\|B_1(H) - B_2(H)\|_\infty$$
$$=\|(H + \mathrm{MHA}_1(H) + \mathrm{FFN}_1(H + \mathrm{MHA}_1(H)))$$
$$\quad - (H + \mathrm{MHA}_2(H) + \mathrm{FFN}_2(H + \mathrm{MHA}_2(\mathrm{H})))\|_\infty$$
$$\leq\|\mathrm{MHA}_1(H) - \mathrm{MHA}_2(H)\|_\infty$$
$$\quad + \|\mathrm{FFN}_1(H + \mathrm{MHA}_1(H)) - \mathrm{FFN}_2(H + \mathrm{MHA}_2(\mathrm{H}))\|_\infty$$
$$\leq 3\kappa^3 d_{embed}^6 U^3 m_\mathrm{T} \ell \eta + 3\kappa^{3+L_{\mathrm{FFN}}} w_{\mathrm{FFN}}^{2L_{\mathrm{FFN}}} d_{embed}^6 U^3 m_\mathrm{T} \ell \eta$$
$$\quad + L_{\mathrm{FFN}}(w_{\mathrm{FFN}}(2d_{embed}^6 \kappa^3 U m_\mathrm{T} \ell) + 2)(\kappa w_{\mathrm{FFN}})^{L_{\mathrm{FFN}}-1}\eta$$
$$\leq (4\kappa^{3+L_{\mathrm{FFN}}} w_{\mathrm{FFN}}^{2L_{\mathrm{FFN}}} d_{embed}^6 U^3 m_\mathrm{T} \ell + L_{\mathrm{FFN}}(w_{\mathrm{FFN}}(2d_{embed}^6 \kappa^3 U m_\mathrm{T} \ell) + 2)(\kappa w_{\mathrm{FFN}})^{L_{\mathrm{FFN}}-1})\eta.$$

Then, we can chain the multi-block together and have the difference

$$\|B_{L_T} \circ \cdots B_1(H) - B'_{L_T} \circ \cdots \circ B'_1(H)\|_\infty \leq 2^{7L_T^2} L_{\mathrm{FFN}} U^{3L_T} d_{embed}^{18L_T^2 L_{\mathrm{FFN}}} \kappa^{6L_T^2 L_{\mathrm{FFN}}} m_\mathrm{T}^{L_T^2} \ell^{L_T^2} \eta.$$

Recall that the decoder layer $D : \mathbb{R}^{d_{embed} \times \ell} \to \mathbb{R}$ is fixed and it outputs the last element in the first row. For the encoding layer $H = \mathrm{PE} + \mathrm{E}(\mathfrak{s})$, both PE and E are fixed and we have $\|\mathrm{PE} + \mathrm{E}(\mathfrak{s})\|_\infty = \|\mathfrak{s}\|_\infty + 1 \leq U + 1$. Thus, together this gives the total error bound between $T, T' \in \mathcal{T}(L_\mathrm{T}, m_\mathrm{T}, d_{embed}, \ell, L_{\mathrm{FFN}}, w_{\mathrm{FFN}}, R, \kappa)$ with $\|\theta - \theta'\|_\infty \leq \eta$ as

$$\|T(\mathfrak{s}) - T'(\mathfrak{s})\|_\infty = \|D \circ B_{L_T} \circ \cdots B_1 \circ (\mathrm{PE} + \mathrm{E}(\mathfrak{s})) - D' \circ B'_{L_T} \circ \cdots \circ B'_1(\mathrm{PE} + \mathrm{E}'(\mathfrak{s}))\|_\infty$$
$$\leq 2^{L_T^2+1} L_{\mathrm{FFN}} U^{3L_T} d_{embed}^{18L_T^2 L_{\mathrm{FFN}}} \kappa^{6L_T^2 L_{\mathrm{FFN}}} m_\mathrm{T}^{L_T^2} \ell^{L_T^2} \eta.$$

Notice that the total number of parameters in the transformer class $\mathcal{T}$ is

$$|\theta| = |\theta_D| + \sum_{i=1}^{L_\mathrm{T}} |\theta_{B_i}| + |\theta_E| = d_{embed} + L_\mathrm{T}(3d_{embed}^2 m_\mathrm{T} + L_{\mathrm{FFN}} w_{\mathrm{FFN}}^2) + d_{embed} D$$
$$\leq d_{embed}(D + 1) + L_\mathrm{T}(3d_{embed}^2 m_\mathrm{T} + L_{\mathrm{FFN}} w_{\mathrm{FFN}}^2).$$

Since the number of steps for each parameter is $\frac{2\kappa}{\eta}$, then the covering number is

$$\mathcal{N}(\delta, \mathcal{T}, \|\cdot\|_\infty)$$
$$\leq \left( \frac{2\kappa \cdot 2^{L_T^2+1} L_{\mathrm{FFN}} U^{3L_T} d_{embed}^{18L_T^2 L_{\mathrm{FFN}}} \kappa^{6L_T^2 L_{\mathrm{FFN}}} m_\mathrm{T}^{L_T^2} \ell^{L_T^2}}{\delta} \right)^{d_{embed}(D+1)+L_T(3d_{embed}^2 m_\mathrm{T} + L_{\mathrm{FFN}} w_{\mathrm{FFN}}^2)}$$
$$= \left( \frac{2^{L_T^2+2} L_{\mathrm{FFN}} U^{3L_T} d_{embed}^{18L_T^2 L_{\mathrm{FFN}}} \kappa^{6L_T^2 L_{\mathrm{FFN}}+1} m_\mathrm{T}^{L_T^2} \ell^{L_T^2}}{\delta} \right)^{d_{embed}(D+1)+L_T(3d_{embed}^2 m_\mathrm{T} + L_{\mathrm{FFN}} w_{\mathrm{FFN}}^2)}.$$

For the covering number of $\mathcal{L} \circ \mathcal{T}$, we have

$$\|\mathcal{L}(T, \mathfrak{s}, y) - \mathcal{L}(T', \mathfrak{s}, y)\|_\infty = (T(\mathfrak{s}) - y_{n+1})^2 - (T'(\mathfrak{s}) - y_{n+1})^2 \leq 4R\|T(\mathfrak{s}) - T'(\mathfrak{s})\|_\infty.$$

Therefore, the covering number

$$\mathcal{N}(\delta, \mathcal{L} \circ \mathcal{T}, \|\cdot\|_\infty)$$
$$\leq \left( \frac{4R \cdot 2\kappa \cdot 2^{L_T^2+1} L_{\mathrm{FFN}} U^{3L_T} d_{embed}^{18L_T^2 L_{\mathrm{FFN}}} \kappa^{6L_T^2 L_{\mathrm{FFN}}} m_\mathrm{T}^{L_T^2} \ell^{L_T^2}}{\delta} \right)^{d_{embed}(D+1)+L_T(3d_{embed}^2 m_\mathrm{T} + L_{\mathrm{FFN}} w_{\mathrm{FFN}}^2)}$$
$$= \left( \frac{2^{L_T^2+4} L_{\mathrm{FFN}} U^{3L_T} d_{embed}^{18L_T^2 L_{\mathrm{FFN}}} \kappa^{6L_T^2 L_{\mathrm{FFN}}+1} m_\mathrm{T}^{L_T^2} \ell^{L_T^2} R}{\delta} \right)^{d_{embed}(D+1)+L_T(3d_{embed}^2 m_\mathrm{T} + L_{\mathrm{FFN}} w_{\mathrm{FFN}}^2)}$$

as desired.

$\square$

### E.6    PROOF OF LEMMA 5

*Proof of Lemma 5.* For convenience, we denote the $i$-th token in the output by

$$A(h_i) := [A(H)]_i = \sum_{j=1}^{\ell} \sigma(\langle Qh_i, Kh_j \rangle) Vh_j, \tag{34}$$

This formula illustrates that the attention mechanism performs a weighted average of token values based on their pairwise interactions.

Let us defined the query, key, and value matrices as

$$Q = \begin{bmatrix} & & & Q^{data} & & \\ 0 & \cdots & 0 & (Q^{\mathcal{I}})_{1,1} & (Q^{\mathcal{I}})_{1,2} & 0 \\ 0 & \cdots & 0 & (Q^{\mathcal{I}})_{2,1} & (Q^{\mathcal{I}})_{2,2} & 0 \\ 0 & \cdots & 0 & 0 & 0 & 1 \end{bmatrix} \quad \text{and} \quad K = \begin{bmatrix} & & & K^{data} & & \\ 0 & \cdots & 0 & (K^{\mathcal{I}})_{1,1} & (K^{\mathcal{I}})_{1,2} & 0 \\ 0 & \cdots & 0 & (K^{\mathcal{I}})_{2,1} & (K^{\mathcal{I}})_{2,2} & 0 \\ 0 & \cdots & 0 & 0 & 0 & -C \end{bmatrix}$$

and $V = \mathbf{e}_i \mathbf{e}_{d_{embed}}^{\top}$. Here we call $Q^{data}, K^{data} \in \mathbb{R}^{(d_{embed}-3) \times d_{embed}}$ the data kernels, $Q^{\mathcal{I}} := \begin{bmatrix} (Q^{\mathcal{I}})_{1,1} & (Q^{\mathcal{I}})_{1,2} \\ (Q^{\mathcal{I}})_{2,1} & (Q^{\mathcal{I}})_{2,2} \end{bmatrix} \in \mathbb{R}^{2 \times 2}$ and $K^{\mathcal{I}} := \begin{bmatrix} (K^{\mathcal{I}})_{1,1} & (K^{\mathcal{I}})_{1,2} \\ (K^{\mathcal{I}})_{2,1} & (K^{\mathcal{I}})_{2,2} \end{bmatrix} \in \mathbb{R}^{2 \times 2}$ the interaction kernels, and $C > 0$ is a large positive number.

Let us choose $Q^{\mathcal{I}}, K^{\mathcal{I}}$ such that $K^{\mathcal{I}} = P_{\mathcal{I}_{t_2}}$ is a projection onto $\mathcal{I}_{t_2}$, and $Q^{\mathcal{I}}$ is a dilation and rotation of $\mathcal{I}_{t_1}$ onto $\mathcal{I}_{t_2}$, i.e., $Q^{\mathcal{I}} \mathcal{I}_{t_1} = C \mathcal{I}_{t_2}$. Now let us compute $A(h_t)$ for $t = t_1$ and $t \neq t_1$.

For any $1 \leq t \leq \ell$, we can write the action $A_h$ on $h_t$ as

$$A(h_t) = \sum_{k=1}^{\ell} \sigma(\langle Qh_t, Kh_k \rangle) Vh_k = \sum_{k=1}^{\ell} \sigma\left(\langle Q^{data} h_t, K^{data} h_k \rangle + \langle Q^{\mathcal{I}} \mathcal{I}_t, K^{\mathcal{I}} \mathcal{I}_k \rangle - C\right) \mathbf{e}_i.$$

**Case I:** $t = t_1$ and $k = t_2$. By construction, we have $\langle Q^{\mathcal{I}} \mathcal{I}_{t_1}, K^{\mathcal{I}} \mathcal{I}_{t_2} \rangle = \langle C \mathcal{I}_{t_2}, \mathcal{I}_{t_2} \rangle = C$. Therefore

$$\sigma\left(\langle Q^{data} h_{t_1}, K^{data} h_{t_2} \rangle + \langle Q^{\mathcal{I}} \mathcal{I}_{t_1}, K^{\mathcal{I}} \mathcal{I}_{t_2} \rangle - C\right) = \sigma\left(\langle Q^{data} h_{t_1}, K^{data} h_{t_2} \rangle + C - C\right)$$
$$= \sigma\left(\langle Q^{data} h_{t_1}, K^{data} h_{t_2} \rangle\right).$$

**Case II:** $t = t_1$ and $k \neq k_2$. We have $\langle Q^{\mathcal{I}} \mathcal{I}_{t_1}, K^{\mathcal{I}} \mathcal{I}_k \rangle \leq \|Q^{\mathcal{I}} \mathcal{I}_{t_1}\|_2 \|K^{\mathcal{I}} \mathcal{I}_k\|_2 = C \|P_{\mathcal{I}_{t_2}} \mathcal{I}_k\|_2 < C$. The last inequality holds since $\|P_{\mathcal{I}_{t_2}} \mathcal{I}_k\|_2 < 1$ when $k \neq t_2$. Thus, for large $C$, we have

$$\sigma\left(\langle Q^{data} h_{t_1}, K^{data} h_k \rangle + \langle Q^{\mathcal{I}} \mathcal{I}_{t_1}, K^{\mathcal{I}} \mathcal{I}_k \rangle - C\right) \leq \sigma\left(\langle Q^{data} h_{t_1}, K^{data} h_k \rangle + C \|P_{\mathcal{I}_{t_2}} \mathcal{I}_k\|_2 - C\right).$$

By choosing $\langle Q^{data} h_{t_1}, K^{data} h_k \rangle + C \|P_{\mathcal{I}_{t_2}} \mathcal{I}_k\|_2 - C < 0$, or equivalently, $C > \frac{\langle Q^{data} h_{t_1}, K^{data} h_k \rangle}{1 - \|P_{\mathcal{I}_{t_2}} \mathcal{I}_k\|_2}$, we get

$$\sigma\left(\langle Q^{data} h_{t_1}, K^{data} h_k \rangle + \langle Q^{\mathcal{I}} \mathcal{I}_{t_1}, K^{\mathcal{I}} \mathcal{I}_k \rangle - C\right) \leq \sigma\left(\langle Q^{data} h_{t_1}, K^{data} h_k \rangle + C \|P_{\mathcal{I}_{t_2}} \mathcal{I}_k\|_2 - C\right) = 0.$$

Combining Case I and II, we conclude $A(h_t) = \sigma\left(\langle Q^{data} h_t, K^{data} h_{t_2} \rangle\right) \mathbf{e}_i$ when $t = t_1$.

**Case III:** $t \neq t_1$ and $k = t_2$. We have

$$\langle Q^{\mathcal{I}} \mathcal{I}_t, K^{\mathcal{I}} \mathcal{I}_{t_2} \rangle = \|Q^{\mathcal{I}} \mathcal{I}_t\|_2 \|K^{\mathcal{I}} \mathcal{I}_{t_2}\|_2 \cos(\theta_{t,t_2}),$$

where $\theta_{t,t_2}$ is the angle between $Q^{\mathcal{I}} \mathcal{I}_t$ and $K^{\mathcal{I}} \mathcal{I}_{t_2}$. Since $t \neq t_1$, $Q^{\mathcal{I}} \mathcal{I}_t \neq C \mathcal{I}_{t_2}$, $\cos(\theta_{t,t_2}) < 1$. Then by choosing $C > \frac{\langle Q^{data} h_t, K^{data} h_{t_2} \rangle}{1 - \cos(\theta_{t,t_2})}$, we have

$$\sigma\left(\langle Q^{data} h_t, K^{data} h_{t_2} \rangle + \langle Q^{\mathcal{I}} \mathcal{I}_t, K^{\mathcal{I}} \mathcal{I}_{t_2} \rangle - C\right) = \sigma\left(\langle Q^{data} h_t, K^{data} h_{t_2} \rangle + C \cos(\theta_{t,t_2}) - C\right) = 0$$

**Case IV:** $t \neq t_1$ and $k \neq t_2$. In this case, we have $(\langle Q^{data} h_t, K^{data} h_k \rangle + \langle Q^{\mathcal{I}} \mathcal{I}_t, K^{\mathcal{I}} \mathcal{I}_k \rangle - C < 0$, so the argument follows the same way as Case 2.

Combining Case III and Case IV, we conclude $A(h_t) = 0$ when $t \neq t_1$.

To obtain the bound on the constant $C$, we need $C > \max\left(\frac{\langle Q^{data}h_{t_1}, K^{data}h_k\rangle}{1-\|P_{\mathcal{I}_{t_2}}\mathcal{I}_k\|_2}, \frac{\langle Q^{data}h_t, K^{data}h_{t_2}\rangle}{1-\cos(\theta_{t,t_2})}\right)$.

Both numerators can be bounded by

$$|\langle Q^{data}h_t, K^{data}h_k\rangle| \leq \|Q^{data}h_t\|_2 \|K^{data}h_k\|_2 \leq \|Q^{data}\|_{1,1}\|h_t\|_\infty\|K^{data}\|_{1,1}\|h_t\|_\infty$$
$$\leq \|Q^{data}\|_{\infty,\infty}d^2_{embed}\|K^{data}\|_{\infty,\infty}d^2_{embed}U^2 \leq d^4_{embed}\kappa^2 U^2.$$

The two denominators can be bounded by

$$1 - \|P_{\mathcal{I}_{t_2}}\mathcal{I}_k\|_2 \geq 1 - \cos(\frac{\pi}{2\ell}) \geq 1 - (1 - O(\ell^{-2})) = O(\ell^{-2}),$$

and

$$1 - \cos(\theta_{t,t_2}) = 1 - \langle \mathcal{I}_{t+t_2-t_1}, \mathcal{I}_{t_2}\rangle \geq 1 - \cos(\frac{\pi}{2\ell}) \geq 1 - (1 - O(\ell^{-2})) = O(\ell^{-2}).$$

The $O(\cdot)$ hides the dependency on some absolute constant. So we conclude $C = O(d^4_{embed}\kappa^2\ell^2 U^2)$.
$\square$

### E.7   PROOF OF LEMMA 6

*Proof of Lemma 6.* We denote the $i$-th column/token by $h_i$ and $j$-th component of $h_i$ by $(h_i)_j$ in the proof. Recall that $\mathcal{I}_t$ is the sinusoid positional encoding, it is easy to see there exists some $\mathbf{v} = (\mathbf{v}_1, \mathbf{v}_2) \in \mathbb{S}^1$ such that $\mathcal{I}_t \cdot \mathbf{v} > 0$ for $t = \{1, \cdots, k_1\}$ and $\mathcal{I}_t \cdot \mathbf{v} < 0$ for $t = \{k_1, \cdots, \ell\}$. Then for large $C$, we can construct (all the blank places are filled with zeros) with $C\mathbf{v}_1, C\mathbf{v}_2$ appears in $d_{embed-2}$-th and $d_{embed-3}$-th columns, from row $r_1$ to row $r_2$ for $1 \leq r_1 \leq r_2 \leq d_{embed-3}$.

$$W_1 = \begin{bmatrix} 1 & & & & & & & & \\ & \ddots & & & & & C\mathbf{v}_1 & C\mathbf{v}_2 & \\ & & \ddots & & & & \vdots & \vdots & \\ & & & 1 & & & C\mathbf{v}_1 & C\mathbf{v}_2 & \\ & & & & \ddots & & & & \\ & & & & & \ddots & & & \\ & & & & & & 1 & & \\ & & & & & & & 1 & \\ & & & & & & & & 1 \end{bmatrix} \in \mathbb{R}^{d_{embed} \times d_{embed}}, \quad \boldsymbol{b}_1 = \mathbf{0} \in \mathbb{R}^{d_{embed}}$$

$$W_2 = \begin{bmatrix} 1 & & & & & & & & \\ & \ddots & & & & & -C\mathbf{v}_1 & -C\mathbf{v}_2 & \\ & & \ddots & & & & \vdots & \vdots & \\ & & & 1 & & & -C\mathbf{v}_1 & -C\mathbf{v}_2 & \\ & & & & \ddots & & & & \\ & & & & & \ddots & & & \\ & & & & & & 1 & & \\ & & & & & & & 1 & \\ & & & & & & & & 1 \end{bmatrix} \in \mathbb{R}^{d_{embed} \times d_{embed}}, \quad \boldsymbol{b}_2 = \mathbf{0} \in \mathbb{R}^{d_{embed}},$$

so that

$$z_1 = \sigma(W_1 h_t + \boldsymbol{b}_1) = \begin{bmatrix} (h_t)_1 \\ \vdots \\ (h_t)_{r_1-1} \\ \sigma((h_t)_{r_1} + C\mathcal{I}_t \cdot \mathbf{v}) \\ \vdots \\ \sigma((h_t)_{r_2} + C\mathcal{I}_t \cdot \mathbf{v}) \\ (h_t)_{r_2+1} \\ \vdots \\ (h_t)_{d_{embed}-3} \\ \mathcal{I}_t^1 \\ \mathcal{I}_t^2 \\ 1 \end{bmatrix} \xmapsto{\text{if } \mathcal{I}_t \cdot \mathbf{v} < 0} \begin{bmatrix} (h_t)_1 \\ \vdots \\ (h_t)_{r_1-1} \\ 0 \\ \vdots \\ 0 \\ (h_t)_{r_2+1} \\ \vdots \\ (h_t)_{d_{embed}-3} \\ \mathcal{I}_t^1 \\ \mathcal{I}_t^2 \\ 1 \end{bmatrix}.$$

and

$$z_1 = \sigma(W_1 h_t + \boldsymbol{b}_1) = \begin{bmatrix} (h_t)_1 \\ \vdots \\ (h_t)_{r_1-1} \\ \sigma((h_t)_{r_1} + C\mathcal{I}_t \cdot \mathbf{v}) \\ \vdots \\ \sigma((h_t)_{r_2} + C\mathcal{I}_t \cdot \mathbf{v}) \\ (h_t)_{r_2+1} \\ \vdots \\ (h_t)_{d_{embed}-3} \\ \mathcal{I}_t^1 \\ \mathcal{I}_t^2 \\ 1 \end{bmatrix} \xmapsto{\text{if } \mathcal{I}_t \cdot \mathbf{v} > 0} \begin{bmatrix} (h_t)_1 \\ \vdots \\ (h_t)_{r_1-1} \\ (h_t)_{r_1} + C\mathcal{I}_t \cdot \mathbf{v} \\ \vdots \\ (h_t)_{r_2} + C\mathcal{I}_t \cdot \mathbf{v} \\ (h_t)_{r_2+1} \\ \vdots \\ (h_t)_{d_{embed}-3} \\ \mathcal{I}_t^1 \\ \mathcal{I}_t^2 \\ 1 \end{bmatrix}.$$

Then apply the second layer yields

$$z_2 = W_2 z_1 + \boldsymbol{b}_2 = \begin{bmatrix} (h_t)_1 \\ \vdots \\ (h_t)_{r_1-1} \\ 0 \\ \vdots \\ 0 \\ (h_t)_{r_2+1} \\ \vdots \\ (h_t)_{d_{embed}-3} \\ \mathcal{I}_t^1 \\ \mathcal{I}_t^2 \\ 1 \end{bmatrix} \quad \text{and} \quad z_2 = W_2 z_1 + \boldsymbol{b}_2 = \begin{bmatrix} (h_t)_1 \\ \vdots \\ (h_t)_{r_1-1} \\ (h_t)_{r_1} \\ \vdots \\ (h_t)_{r_2} \\ (h_t)_{r_2+1} \\ \vdots \\ (h_t)_{d_{embed}-3} \\ \mathcal{I}_t^1 \\ \mathcal{I}_t^2 \\ 1 \end{bmatrix}$$

respectively. This shows (24). Similarly, there exists some $\mathbf{v} = (\mathbf{v}_1, \mathbf{v}_2) \in \mathbb{S}^1$ such that $\mathcal{I}_t \cdot \mathbf{v} < 0$ for $t = \{1, \cdots, k_1\}$ and $\mathcal{I}_t \cdot \mathbf{v} > 0$ for $t = \{k_1, \cdots, \ell\}$. Applying the same argument we get (25).

To obtain a bound on the constant $C$, we need $|C\mathcal{I}_t \cdot \mathbf{v}| > \|H\|_\infty$. Hence $C > \frac{\|H\|_\infty}{|\mathcal{I}_t \cdot \mathbf{v}|} = O(\ell \|H\|_\infty)$.

$\square$

### E.8 PROOF OF LEMMA 7

*Proof of Lemma 7.* Given an $H = [h_t]_{1 \le t \le \ell}$, we apply the first layer of FFN with

$$
W_1 = \begin{bmatrix} & & & \\ & 1 & & \\ & & 1 & \\ & & & 1 \end{bmatrix} \in \mathbb{R}^{d_{embed} \times d_{embed}} \quad \text{and} \quad b_1 = \begin{bmatrix} M \\ \vdots \\ M \\ 0 \\ 0 \\ 0 \end{bmatrix} \in \mathbb{R}^{d_{embed}},
$$

so that the output after the first layer of FFN is

$$
H_1 = \begin{bmatrix} M & \cdots & M \\ \vdots & & \vdots \\ M & \cdots & M \\ \mathcal{I}_1 & \cdots & \mathcal{I}_\ell \\ 1 & \cdots & 1 \end{bmatrix} \in \mathbb{R}^{d_{embed} \times \ell}.
$$

Then by Lemma 6, there exists a two-layer FFN such that the output after applying these two layers become

$$
H_3 = \begin{bmatrix} M & \cdots & M & 0 & \cdots & 0 \\ \vdots & & \vdots & \vdots & & \vdots \\ M & \cdots & M & 0 & \cdots & 0 \\ \mathcal{I}_1 & \cdots & \mathcal{I}_{k_2} & \mathcal{I}_{k_2+1} & \cdots & \mathcal{I}_\ell \\ 1 & \cdots & 1 & 1 & \cdots & 1 \end{bmatrix} \in \mathbb{R}^{d_{embed} \times \ell}.
$$

Again by Lemma 6, there exists a two-layer FFN such that the output after applying these two layers become

$$
H_5 = \begin{bmatrix} 0 & \cdots & 0 & M & \cdots & M & 0 & \cdots & 0 \\ \vdots & & \vdots & \vdots & & \vdots & \vdots & & \vdots \\ 0 & \cdots & 0 & M & \cdots & M & 0 & \cdots & 0 \\ \mathcal{I}_1 & \cdots & \mathcal{I}_{k_1} & \mathcal{I}_{k_1+1} & \cdots & \mathcal{I}_{k_2} & \mathcal{I}_{k_2+1} & \cdots & \mathcal{I}_\ell \\ 1 & \cdots & 1 & 1 & \cdots & 1 & 1 & \cdots & 1 \end{bmatrix} \in \mathbb{R}^{d_{embed} \times \ell}.
$$

Finally, we apply a FFN with

$$
W_6 = \begin{bmatrix} -1 & & & & & \\ & \ddots & & & & \\ & & -1 & & & \\ & & & & & \\ & & & & 1 & \\ & & & & & 1 \\ & & & & & & 1 \end{bmatrix} \in \mathbb{R}^{d_{embed} \times d_{embed}} \quad \text{and} \quad b_6 = 0 \in \mathbb{R}^{d_{embed}},
$$

where the entries $-1$ appear in $r_1$-th row to $r_2$-th row and $r_1$-th column to $r_2$-th column. Therefore, the output after applying $W_6$ and $\boldsymbol{b}_6$ is

$$H_6 = \begin{bmatrix} 0 & \cdots & 0 & 0 & \cdots & 0 & 0 & \cdots & 0 \\ \vdots & & \vdots & \vdots & & \vdots & \vdots & & \vdots \\ 0 & \cdots & 0 & 0 & \cdots & 0 & 0 & \cdots & 0 \\ 0 & \cdots & 0 & -M & \cdots & -M & 0 & \cdots & 0 \\ \vdots & & \vdots & \vdots & & \vdots & \vdots & & \vdots \\ 0 & \cdots & 0 & -M & \cdots & -M & 0 & \cdots & 0 \\ 0 & \cdots & 0 & 0 & \cdots & 0 & 0 & \cdots & 0 \\ \vdots & & \vdots & \vdots & & \vdots & \vdots & & \vdots \\ 0 & \cdots & 0 & 0 & \cdots & 0 & 0 & \cdots & 0 \\ \mathcal{I}_1 & \cdots & \mathcal{I}_{k_1} & \mathcal{I}_{k_1+1} & \cdots & \mathcal{I}_{k_2} & \mathcal{I}_{k_2+1} & \cdots & \mathcal{I}_\ell \\ 1 & \cdots & 1 & 1 & \cdots & 1 & 1 & \cdots & 1 \end{bmatrix} \in \mathbb{R}^{d_{embed} \times \ell},$$

where the entries $-M$ appear in $r_1$-th row to $r_2$-th row and $k_1$-th column to $k_2$-th column. Therefore, the residual FFN gives the output

$$\text{FFN}(h_t) + h_t = \begin{cases} h_t & \text{if } t \in \{1, \cdots, k_1\} \cup \{k_2, \cdots, \ell\} \\[2mm] \begin{bmatrix} (h_t)_1 \\ \vdots \\ (h_t)_{r_1-1} \\ (h_t)_{r_1} - M \\ \vdots \\ (h_t)_{r_2} - M \\ (h_t)_{r_2+1} \\ \vdots \\ (h_t)_{d_{embed}-3} \\ \mathcal{I}_t \\ 1 \end{bmatrix} & \text{otherwise} \end{cases}$$

as desired. $\qquad\square$

### E.9 BOUNDING THE ERROR II IN (16)

Since $0 \le (\text{T}^*(\{\mathbf{x}_i^\gamma, y_i^\gamma\}_{i=1}^n); \mathbf{x}_{n+1}^\gamma) - y_{n+1}^\gamma)^2 \le 4R^2$, by Hoeffding's inequality, for any $t > 0$, it satisfies

$$\mathbb{P}(\mathcal{R}_{n,\Gamma}(\text{T}^*) - \mathbb{E}_{\mathfrak{s}}[\mathcal{R}_n(\text{T}^*(\mathfrak{s}))] \ge t) \le e^{-\frac{t^2 \Gamma}{8R^4}}.$$

Hence with probability at least $1 - \delta$, it satisfies

$$\mathcal{R}_{n,\Gamma}(\text{T}^*) - \mathbb{E}_{\mathfrak{s}}[\mathcal{R}_n(\text{T}^*(\mathfrak{s}))] \le R^2 \sqrt{\frac{8\log(1/\delta)}{\Gamma}}.$$

Let $\delta = h^2$, we get

$$\mathbb{E}_{\mathfrak{S}}\left(\mathcal{R}_{n,\Gamma}(\text{T}^*) - \mathbb{E}_{\mathfrak{s}}[\mathcal{R}_n(\text{T}^*(\mathfrak{s}))]\right) \le O\left(\sqrt{\frac{\log(h^{-1})}{\Gamma}} + h^2\right),$$

where $O(\cdot)$ hides the dependency on $R$.

# F    ADDITIONAL EXPERIMENTS AND DETAILS

$$H = \begin{bmatrix} x_{1,1}^{\gamma} & \cdots & x_{1,n}^{\gamma} & x_{1,n+1}^{\gamma} & \mathbf{0} \\ x_{2,1}^{\gamma} & \cdots & x_{2,n}^{\gamma} & x_{2,n+1}^{\gamma} & \mathbf{0} \\ x_{3,1}^{\gamma} & \cdots & x_{3,n}^{\gamma} & x_{3,n+1}^{\gamma} & \mathbf{0} \\ y_1^{\gamma} & \cdots & y_n^{\gamma} & 0 & \mathbf{0} \\ 0 & \cdots & \cdots & \cdots & 0 \\ \mathcal{I}_1 & \cdots & \cdots & \cdots & \mathcal{I}_{\ell} \\ 1 & \cdots & \cdots & \cdots & 1 \end{bmatrix}, \tag{35}$$

## F.1    ADDITIONAL EXPERIMENTAL DETAILS

For the transformer architecture we used for the experiments in Section 4, we fix $d_{embed} = 8$, $L_{\mathrm{T}} = 5$, $L_{\mathrm{FNN}} = 6$. The number of attention heads is $m = 1$ for $n = 4, 8, 16, 32$. We generate $\Gamma = 50000$ for both training and testing. The model is trained with batch size 100, using Adam with learning rate 0.0005 for 100 epochs.

For experiments in Section 5, we fix $d_{embed} = 8$, $L_{\mathrm{T}} = 5$, $L_{\mathrm{FNN}} = 6$, and the number of attention heads $m = 2, 4, 8$ for $n = 16, 64, 256$ respectively. We generate $\Gamma = 400, 1600, 6400$ for both training and testing. The model is trained with batch size 100, using Adam with learning rate 0.0005 for 100 epochs.

To make the experiment setup the as close as to our theory suggests, we apply the softmax activation in the last layer of our transformer model, and ReLU activation in all the layers before the last layer. The activation function for the feed-forward components are ReLU activation.

The following sentences are used to generate the attention score curves in Figure 3. Sentences are cut in the end so that all the sentences have the same length.

Sentence 1: "In the quiet town by the river, a curious child spent the afternoon reading stories about distant galaxies and dreaming of becoming an astronaut one day."

Sentence 2: "The professor walked slowly across the lecture hall, carefully explaining how black holes bend space and time while students scribbled furiously in their notebooks."

Sentence 3: "On a rainy evening in Paris, a young artist painted the city's rooftops in dazzling colors, imagining how the world might look if dreams could shape reality."

Sentence 4: "The spacecraft drifted silently beyond the orbit of Saturn, transmitting faint signals back to Earth as scientists waited anxiously for news of its discoveries."

Sentence 5: "In the heart of the ancient forest, an owl watched quietly from a high branch, while a fox padded softly across the moss-covered ground below."

## F.2    ADDITIONAL EXPERIMENTAL RESULTS

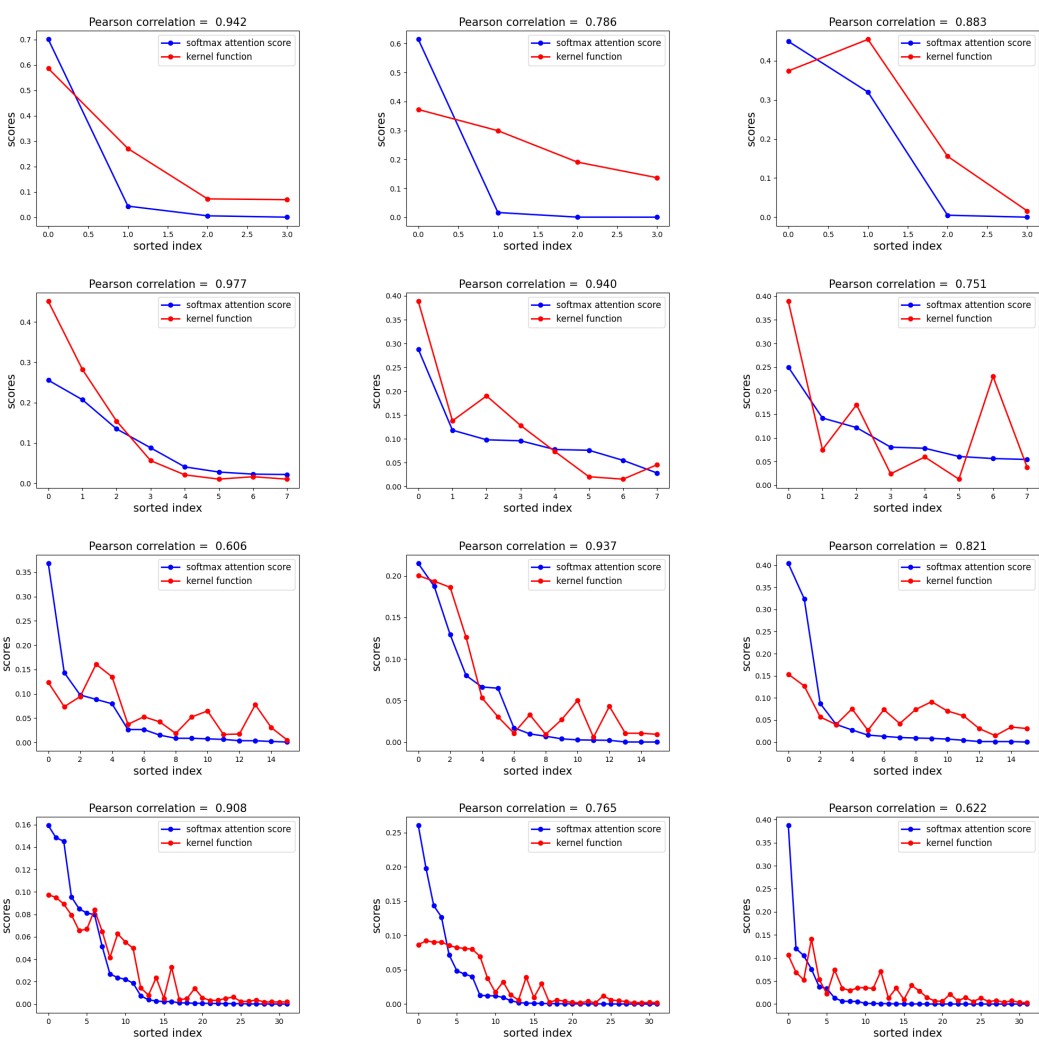

Figure 5: More examples of attention scores and Gaussian kernel function with in-context length $n = 4, 8, 16, 32$ respectively.

