# OpenReview forum: "Understanding In-Context Learning on Structured Manifolds: Bridging Attention to Kernel Methods"
_ICLR.cc/2026/Conference — ICLR 2026 Poster_

### Official Review · Reviewer_iBpf · 2025-10-28

**Soundness:** 2
**Presentation:** 3
**Contribution:** 2
**Rating:** 4
**Confidence:** 4

**Summary:**

This paper investigates the theoretical properties of in-context learning when applied to data residing on structured manifolds with a low intrinsic dimension. The authors begin by presenting an expressiveness result, demonstrating that a Transformer can utilize $O(n)$ hidden tokens to approximate a kernel smoothing operation. Building on this constructive proof, the paper derives a generalization bound for a function class of a comparable size to the Transformer construction. This bound is intended to clarify the role of the manifold's intrinsic dimension $d$ in the generalization of ICL. Finally, the paper includes synthetic experiments to validate these theoretical findings.

**Strengths:**

1. The paper frames an interesting and relevant problem: the theoretical analysis of in-context learning on high-dimensional data that possesses a low-dimensional intrinsic manifold structure. This setup is well-motivated and timely given current data regimes.

2. The theoretical claims regarding expressiveness and generalization are supported by corresponding synthetic experiments, which adds to the paper's completeness.

3. The paper is generally clearly written and well-organized.

**Weaknesses:**

1. The paper's discussion of novelty could be strengthened. The connection between attention mechanisms and kernel smoothing has been explored in prior work (e.g., [1, 2]). It would be beneficial for the authors to more clearly articulate the novelty of their specific formulation in light of these existing results.

2. The central claim regarding the generalization bound's (Eq. 15) dependence on the intrinsic dimension $d$ and ambient dimension $D$ requires further clarification.

   - Firstly, the claim that the bound scales exponentially with $d$ is difficult to verify, as the constants $C_1$ and $C_2$ are also noted to be dependent on $d$. To make this claim more concrete, it would be helpful to explicitly characterize the dependency of $C_1$ and $C_2$ on $d$.

   - Secondly, the bound's polynomial dependence on the ambient dimension ($D^3$) seems to obscure the intended message about the intrinsic dimension $d$. A key motivation for manifold-based analysis is often to obtain bounds that depend primarily on $d$ and are less sensitive to $D$. As written, the significance of the $D^3$ term on the bound's practical implications is unclear.

3. The theoretical setup, which relies on $l = O(n)$ hidden tokens for the expressiveness result, appears to differ from the standard ICL paradigm, where new outputs are typically generated without a set of internal "hidden tokens". The authors could provide a clearer justification for this modeling choice and discuss its implications for the relevance of the derived bounds to standard Transformer-based ICL.

[1] Transformer Dissection: A Unified Understanding of Transformer's Attention via the Lens of Kernel. https://arxiv.org/abs/1908.11775.

[2] Implicit Kernel Attention. https://arxiv.org/abs/2006.06147.

**Questions:**

1. Could the authors provide a more explicit characterization of the generalization bound (Eq. 15)? Specifically, how do the constants $C_1$ and $C_2$ depend on the intrinsic dimension $d$, and how should the $D^3$ term be interpreted in the context of a manifold-aware analysis?

2. What are the theoretical implications of the $l = O(n)$ hidden token requirement? Does the analysis hold for a more standard setting, such as $l = 0$? A clearer justification for the role of these hidden tokens and their necessity for the expressiveness result would be valuable.

---

> ### Author Response · Authors · 2025-11-21
> **Rebuttal**
>
> # Response to Reviewer iBpf
>
> We thank the reviewer for acknowledging that our study is interesting and relevant.
>
> >  ***Weakness 1:** The paper's discussion of novelty could be strengthened. The connection between attention mechanisms and kernel smoothing has been explored in prior work (e.g., [1, 2]). It would be beneficial for the authors to more clearly articulate the novelty of their specific formulation in light of these existing results.*
>
> **Response:** Thank you for pointing out the references [1,2]. We have acknowledged in related works that the connection between the attention mechanism and kernel methods has been observed in prior work. We already cited [1] in the original manuscript, and we added [2] in the revised manuscript. Our novelty is that we theoretically establish this connection in the manifold setting as well as numerically validate the connection. The construction of transformers to implement the kernel method in Lemma 1 and the generalization error bound in Theorem 1 have not been addressed in literature.
>
> [1] Transformer Dissection: A Unified Understanding of Transformer's Attention via the Lens of Kernel. https://arxiv.org/abs/1908.11775.
>
> [2] Implicit Kernel Attention. https://arxiv.org/abs/2006.06147.
>
>
> >  ***Weakness 2:** The central claim regarding the generalization bound's (Eq. 15) dependence on the intrinsic dimension and ambient dimension $D$ requires further clarification...*
>
> **Response:** Thank you for the question and we would like to clarify. The constants $C_1$ and $C_2$ are dependent on the complexity of the underlying input manifold (which in the worst case can be exponential in $d$) and the function (task) space, but are independent of $D$, $n$, and $\Gamma$. We are interested in studying the neural scaling laws in terms of $n$ and $\Gamma$, and this can be verifed by taking the log-log transformation. Having constants $C_1$ and $C_2$ only affect the intercept of the straight line in the log-log plot, but do not affect the slope.
>
> Regarding the curse of dimensionality (CoD), what is studied more often is the dependency with respect to the training data size. In our setting, the exponent in $\Gamma$ and $n$ is related to the curse of dimensionality (CoD). It important to show that the power in the exponential of $\Gamma$ and $n$ depends on the intrinsic dimension $d$. The constant of $D^3$ does not affect the slope in the log-log plot, hence does not affect the result of our neural scaling law. However, we do agree with the reviewer that this $D^3$ factor may not be tight, as the matrices Q, K, V in our construction are very sparse, so it is quite likely this factor can be improved.
>
>
>
> >  ***Weakness 3:** The theoretical setup, which relies on $l=O(n)$ hidden tokens for the expressiveness result, appears to differ from the standard ICL paradigm, where new outputs are typically generated without a set of internal "hidden tokens". The authors could provide a clearer justification for this modeling choice and discuss its implications for the relevance of the derived bounds to standard Transformer-based ICL.*
>
> **Response:** Our understanding of the standard in-context learning (ICL) paradigm is that predictions are made based on a few provided examples, commonly referred to as the prompt. We follow this standard paradigm, where we are given the prompt $(x_1, y_1, x_2, y_2, \cdots, x_n, y_n, x_{n+1})$, and the goal is to make the prediction on $y_{n+1}$. We are not entirely sure what the reviewer means by “without a set of internal hidden tokens,” and we would be happy to provide further clarification in the upcoming discussion.
>
>
> >  ***Question 1:** Could the authors provide a more explicit characterization of the generalization bound (Eq. 15)? Specifically, how do the constants $C_1$ and $C_2$ depend on the intrinsic dimension $d$, and how should the $D^3$ term be interpreted in the context of a manifold-aware analysis?*
>
> **Response:** We refer to our response to Weakness 2.
>
> >  ***Question 2:** What are the theoretical implications of the $l=O(n)$ hidden token requirement? Does the analysis hold for a more standard setting, such as $l=0$? A clearer justification for the role of these hidden tokens and their necessity for the expressiveness result would be valuable.*
>
> **Response:** In our ICL paradigm, a prompt of length $n$ is provided on the fly for the pre-trained transformer to select which task to perform. Then, given a new input $x_{n+1}$, the model is able to make the prediction $y_{n+1}$ based on the selected task. If $\ell=0$, then no prompt is provided, which is different from our setup. To be more precise, the $O(n)$ hidden tokens plays the role of guiding the pre-trained model to perform kernel regression for each individual task.

---

> > ### Comment · Reviewer_iBpf · 2025-11-24
> >
> > Thank you for your response and clarification. While I appreciate the updates, I still have several concerns regarding the generalization bounds and the specific ICL setup used in the analysis.
> >
> > 1. Regarding the generalization bound in Eqn. 15, I remain concerned about the presentation of constants $C_1$ and $C_2$. The paper claims that the bounds depend exponentially on the intrinsic dimension $d$. However, subsuming terms dependent on $d$ within $C_1$ and $C_2$ -- while labeling them "constants" -- obscures the true complexity of the bound. To improve transparency and precision, I strongly recommend one of the following revisions:
> >
> >    - Explicitly define the functional dependence of $C_1$ and $C_2$ with respect to $d$.
> >    - Restate the claim to expose the full dependency on $d$ in the main equation, reserving the term "constants" for terms (e.g., $C_1', C_2'$) that are strictly independent of the dimension.
> >
> > 2. I would like to discuss the ICL formulation further, as the current setup appears distinct from established conventions. Standard ICL literature typically follows one of two tokenization strategies:
> >
> >    - Tokenizing $x_i$ and $y_i$ separately, resulting in $2n+1$ tokens (where $n$ is the sample size) with a single output token $y_{\text{query}}$. See [1, Theorem 1].
> >    - Tokenizing each pair $(x_i, y_i)$ into a single token, resulting in $n+1$ tokens (where the final token represents the query), with the output being $(x_{\text{query}}, y_{\text{query}})$. See [2, Proposition 1].
> >
> >    The setup in Eqn. 11 deviates from these standards by introducing $n$ additional tokens that function effectively as padding. These extra tokens appear central to the construction in Eqn. 13, specifically for computing the difference vectors $x_{n+1} - x_1, \dots, x_{n+1} - x_n$. Consequently, this formulation does not seem to align with standard ICL or CoT frameworks, as neither typically relies on $n$ padding tokens following the query input $x_{n+1}$.
> >
> >    I acknowledge a misunderstanding regarding the definition of $l$ in my initial review. With that clarified, I have a specific follow-up: Does your analysis hold for the case where $l=n+1$? In other words, is the theoretical result valid without the $n$ extra padding tokens following $x_{n+1}$ in Eqn. 11?
> >
> > [1] What learning algorithm is in-context learning? Investigations with linear models. https://arxiv.org/abs/2211.15661.
> >
> > [2] Transformers learn in-context by gradient descent. https://arxiv.org/abs/2212.07677

---

> > > ### Author Response · Authors · 2025-11-25
> > > **Response to the comments by Reviewer iBpf**
> > >
> > > We appreciate the reviewer's response to our rebuttal, and we would like to further address the concern.
> > >
> > > >  ***Concern 1:** Regarding the constants $C_1$ and $C_2$.*
> > >
> > > **Response:** Thank you for emphasizing this. We re-examined our proof and found that the constant $C_1$ depends on $d$ and $\alpha$ via the multiplication factor $\frac{4\alpha+2d+8}{2\alpha+d}$ which can be uniformly bounded by a constant, e.g., $\frac{4\alpha+2d+8}{2\alpha+d}\leq 10$. For constant $C_2$, our proof of Lemma 2 and Lemma 3 shows that the dependency on $d$ could be $d^{d/2}$ in the worst case scenario. We have now addressed this change in the statement of Theorem 1, Lemma 2 and Lemma 3 in the revised manuscript.
> > >
> > > >  ***Concern 2:** Regarding the tokenization of ICL.*
> > >
> > > **Response:** Thank you for the clarification. Our construction still works if with $\ell=n+1$. In the $\ell=n+1$ case, instead of having matrix of size ${d_{embed}\times (2n+1)}=(D+5)\times (2n+1)$, we will have the matrix of size $(d_{embed}+D)\times (n+1)=(2D+5)\times (n+1)$ in order to store the additional paddings in the rows instead of columns. To be more precise, instead of having
> > >
> > > \begin{equation*}
> > > H_4 = \begin{bmatrix}
> > >     x_{1} & \cdots & x_{n} & x_{n+1} & x_{n+1}-x_1 & \cdots & x_{n+1}-x_n \cr
> > >     y_1 & \cdots & y_n & 0 & -\frac{||x_{n+1}-x_1||^2}{h^2} & \cdots & -\frac{||x_{n+1}-x_n||^2}{h^2}  \cr
> > >     0 & \cdots & \cdots & \cdots & y_1 & \cdots & y_n \cr
> > >     I_1 & \cdots & \cdots & \cdots & \cdots & \cdots & I_{2n+1} \cr
> > >     1 & \cdots & \cdots & \cdots & \cdots & \cdots & 1
> > > \end{bmatrix} \in\mathbb{R}^{(D+5)\times (2n+1)}
> > > \end{equation*}
> > >
> > >
> > > as in our original construction, we will have
> > >
> > > \begin{equation*}
> > >     H_4 = \begin{bmatrix}
> > >     x_{1} & \cdots & x_{n} & x_{n+1} \cr
> > >     y_1 & \cdots & y_n & 0   \cr
> > >     x_{n+1}-x_1 & \cdots & x_{n+1}-x_n & 0  \cr
> > >  -\frac{||x_{n+1}-x_1||^2}{h^2} & \cdots & -\frac{||x_{n+1}-x_n||^2}{h^2} & 0 \cr
> > >     I_1 & \cdots & I_{n} & I_{n+1}  \cr
> > >     1 & \cdots & 1 & 1
> > > \end{bmatrix}\in\mathbb{R}^{(2D+5)\times (n+1)},
> > > \end{equation*}
> > >
> > > and then
> > >
> > > \begin{equation*}
> > >     H_5:={\rm B_5}(H_4) = \begin{bmatrix}
> > >     x_{1} & \cdots & x_{n} & x_{n+1} \cr
> > >     y_1 & \cdots & y_n & K_h(s)   \cr
> > >     x_{n+1}-x_1 & \cdots & x_{n+1}-x_n & 0  \cr
> > >  -\frac{||x_{n+1}-x_1||^2}{h^2} & \cdots & -\frac{||x_{n+1}-x_n||^2}{h^2} & 0 \cr
> > >     I_1 & \cdots & I_{n} & I_{n+1}  \cr
> > >     1 & \cdots & 1 & 1
> > > \end{bmatrix}\in\mathbb{R}^{(2D+5)\times (n+1)}.
> > > \end{equation*}
> > >
> > > As guaranteed by the Interaction Lemma (Lemma 5), we can easily build this construction with $\ell=n+1$ in a similar way to the original setting with $\ell=2n+1$. Mathematically speaking, there are no essential difference between the two.

---

> > > > ### Comment · Reviewer_iBpf · 2025-11-27
> > > >
> > > > I thank the authors for their response. However, I remain concerned regarding the specific dependency of the constant $C_2$ on the intrinsic dimension $d$. Given that $C_2$ scales as $d^{d/2}$ in the worst case, the claim in the main paper that "the generalization bound scales exponentially with intrinsic dimension $d$" appears to be technically imprecise, as this scaling is superexponential. Could the authors please provide further clarification on how to interpret this theoretical result in light of this dependency of $d$?

---

> > > > > ### Author Response · Authors · 2025-11-28
> > > > > **Response to the comments by Reviewer iBpf**
> > > > >
> > > > > Thank you for the suggestion. The constants in our bound may not be optimal, however, the main interest of this paper is to study the scaling law of the generalization error in terms of $n$ and $\Gamma$. Note that the constant $d^{d/2}$ essentially becomes part of the intercept of a line equation after taking the log-log transformation.
> > > > >
> > > > > Let's say if
> > > > > \begin{equation*}
> > > > > {\rm Error} = d^{d/2}\cdot n^{-\frac{2\alpha}{2\alpha+d}}\log^{1+\frac{3d}{4}}(n).
> > > > > \end{equation*}
> > > > >
> > > > > After taking the log10-log10 transform, it becomes
> > > > >
> > > > > \begin{align*}
> > > > > \log_{10}({\rm Error}) = - \frac{2\alpha}{2\alpha+d}\log_{10}(n) + \frac{d}{2}\log_{10}(d) +\left(1+\frac{3d}{4}\right)\log_{10}(\log (n)) \approx -\frac{2\alpha}{2\alpha+d}\log_{10}(n) + \frac{d}{2}\log_{10}(d) = slope\cdot\log_{10}(n) + intercept.
> > > > > \end{align*}
> > > > >
> > > > > Hence, although the constant $d^{d/2}$ may not be ideal, it does not affect our analysis on studying the scaling law of $n$. In the revised manuscript, we have slightly modified our claim in the abstract and after Theorem 1 in Page 8 and also in Page 10 (highlighted in blue), to emphasize that we focus on studying the scaling law of the generalization error with respect to $n$ and $\Gamma$.

---

### Official Review · Reviewer_WAPE · 2025-10-28

**Soundness:** 4
**Presentation:** 2
**Contribution:** 3
**Rating:** 6
**Confidence:** 2

**Summary:**

This paper provides a bridge between Transformer attention mechanisms and classical kernel methods in in-context learning (ICL) over manifold-structured data. The authors prove that a Transformer can exactly implement Gaussian kernel regression and establish an approximation error bound for α-Hölder functions on compact Riemannian manifolds, which demonstrates dependence on intrinsic (manifold) rather than ambient dimension. They further validate their claims through simulations, observing high correlations between attention weights and Gaussian kernels.

**Strengths:**

+ The paper aims to establish connection between attention scores and kernel estimators (Nadaraya-Watson). This yields a new perspective on Transformers as in-context kernel learners, which conceptually advances and generalizes beyond prior linear-model analyses.

+ The construction of Lemma 1 remarking that a Transformer network that exactly realizes kernel regression with zero approximation error is nontrivial. The explicit architectural specification further concretizes its theoretical claim.

+ Theorem 1’s bound  separates the task-level and prompt-level effects, showing minimax-optimal scaling with intrinsic dimension d.

**Weaknesses:**

- The theoretical results rely on exact kernel implementation via attention and perfect manifold sampling. These assumptions may obscure how robust the results remain under approximate or noisy conditions typical in practice.

- Some key proof components, particularly how the softmax attention with masking produces the Gaussian weights in Eqs. (12) - (14) are deferred to appendices. The main text should outline at least the constructive steps more transparently for better readability.

- While the simulation results illustrate correlation trends between attention scores and Gaussian kernels, the experiments are synthetic and low-dimensional (S² sphere). There is no ablation on real-world datasets or on deviations when the model is not perfectly aligned with the Gaussian kernel hypothesis.

**Questions:**

1. On Remark 1 (Universality), the lemma asserts that the same parameterization works for all f and {x_i}. Could the authors clarify how positional encodings and attention masks encode x_i-dependent interactions without weight adaptation? Does this require scaling of input normalization with h or b?

2. The generalization rate depends on the kernel bandwidth h. How is it chosen or adapted across prompts/tasks? Is it optimized implicitly by the Transformer parameters, or assumed known? The statistical rate in Proposition 1 seems sensitive to this.

3. The analysis is limited to α-Hölder smooth functions, but why is it required? Could the framework be extended to broader classes such as Besov or Sobolev spaces, or to non-compact manifolds? Can you discuss how would that alter the covering number and rate derivations?

---

> ### Author Response · Authors · 2025-11-21
> **Rebuttal**
>
> # Response to Reviewer WAPE
>
> We would like to thank the reviewer for acknowledging that our theoretical contribution is nontrivial.
>
>
> >  ***Weakness 1:** The theoretical results rely on exact kernel implementation via attention and perfect manifold sampling...*
>
> **Response:** Our proof framework can be easily adapted to handle noise on the output $y$, according to Theorem 5.2 in [3], which demonstrates that the rate of convergence for kernel methods reimains the same if one adds i.i.d bounded noise on $y$. If we add noise on the input $x$, then the input domain becomes a tubular region around the manifold. We expect the upper bound to have an additional term on the magnitude square of the noise, but the theoretical analysis requires substantial addtional work to establish it rigorously. This can be completed in our future work.
>
>
> [3] Györfi, László, Michael Kohler, Adam Krzyżak, and Harro Walk. A distribution-free theory of nonparametric regression. New York, NY: Springer New York, 2002.
>
> >  ***Weakness 2:** Some key proof components, particularly how the softmax attention with masking produces the Gaussian weights in Eqs. (12) - (14) are deferred to appendices. The main text should outline at least the constructive steps more transparently for better readability.*
>
> **Response:** Thank you for the suggestion, we omitted some of the details in the main text mainly due to the page limit. We now added an additional page in the revised manuscript to include more details for the construction steps in Eqs. (12) - (14) in Sec. 4 of the main text.
>
>
>
> >  ***Weakness 3:** While the simulation results illustrate correlation trends between attention scores and Gaussian kernels, the experiments are synthetic and low-dimensional ($S^2$ sphere). There is no ablation on real-world datasets or on deviations when the model is not perfectly aligned with the Gaussian kernel hypothesis.*
>
> **Response:** We did try on real language data using GPT2-small (Figure 3). The learned score in language data exhibits a kernel shape, but the groundtruth function of the language data is unknown, so it is hard to validate the correlation. We considered a Guassian kernel since the Holder function class is uniformly regular and isotropic. If the function class has additional structures, it would be the best to consider special kernels which are adaptive the structures of the function class.
>
>
> >  ***Question 1:** On Remark 1 (Universality), the lemma asserts that the same parameterization works for all f and ${x_i}$. Could the authors clarify how positional encodings and attention masks encode $x_i$-dependent interactions without weight adaptation? Does this require scaling of input normalization with $h$ or $b$?*
>
> **Response:** Thank you for this great question. What we meant by Universality is that the transformer parameters (key, query, value matrices constructed in Lemma 1, and also the weight matrices in the FFN component) are universal for all $f$ and $\{x_i\}$. This is achieved by the Interaction Lemma 5 and the Decrementing Lemma 7. The parameter $b$ is the bound for input $x$ and $h$ is the kernel bandwidth; the transformer parameters do depend on $b$ and $h$, but are independent of the $x_i$'s and $f$.
>
> >  ***Question 2:** The generalization rate depends on the kernel bandwidth h. How is it chosen or adapted across prompts/tasks? Is it optimized implicitly by the Transformer parameters, or assumed known? The statistical rate in Proposition 1 seems sensitive to this.*
>
> **Response:** In general, we do not assume $h$ is known, and $h$ is choosen as $h=n^{-\frac{1}{2\alpha+d}}$ (stated in Appendix C.3) to balance the bias and variance terms in the kernel regression.
>
>
>
> >  ***Question 3:** The analysis is limited to $\alpha$-Hölder smooth functions, but why is it required? Could the framework be extended to broader classes such as Besov or Sobolev spaces, or to non-compact manifolds? Can you discuss how would that alter the covering number and rate derivations?*
>
> **Response:** We consider $\alpha$-Hölder functions which has practical insights that the output varies proportionally to the input variation. Our results can be applied to Besov or Sobolev spaces where the error is given by the kernel estimator for those functions. If the manifold is non-compact or unbounded, some steps in our proof become tricky since our current proof relies on some lower bound of the measure in any neighorhood of radius $h$. To tackle non-compact or unbounded manifolds, we will need the additional condition that the measure $\rho_x$ decays sufficiently fast. With this fast decay condition, we would truncate the manifold within a bounded region, and then bound the tail separately.

---

### Official Review · Reviewer_Sc1b · 2025-10-31

**Soundness:** 3
**Presentation:** 3
**Contribution:** 4
**Rating:** 8
**Confidence:** 4

**Summary:**

This paper theoretically studies in-context learning on structured data represented as manifolds pertaining to certain assumptions. They start with an explicit construction of a transformer that can perform Gaussian kernel regression without any error. They also give a theoretical framework for generalizing this result by analyzing transformer-based ICL to Reimannian manifolds, and give bounds on the error which are tighter than existing ones. Subsequently, they verify the theoretical claims with experiments that show that the bounds are indeed as one would expect.

**Strengths:**

- This is an important direction which theoretically studies theoretical representational strengths of transformer-based ICL, and relates it to the role of geometry of the data.
- The approximation bounds provided are tight.
- This will have several scopes for future work, where learning techniques can be studies using similar methods to relate to the structure of the data.
- The proofs are very nicely structured despite being long, which makes them easy to understand.

**Weaknesses:**

- Since the main purpose of the work is to provide a theoretical analysis, more emphasis should be given to the results and the proof techniques in the main part of the paper. Especially a proof/technique overview along with the first part of Section C would make more sense.
- Section 3- Please define the input tokens more precisely, for e.g. we are given with (n+1) tokens where the first n tokens contain (x_i, y_i) and the last token contains x_{n+1}, we expect the final output to be present in a certain token etc. Also is \Gamma the batch size? I thought later it is used as the size of the training data.
- The constructed transformer has 5 layers, it would be helpful to give an intuitive understanding of what those layers perform.
- The paper assumes some background on manifolds, but it would be better to explain the concepts more elaborately, especially in Sec B.1.

**Questions:**

- Line 790: What is P_f?
- What are hidden tokens in Def 3? And how are they represented in the input?
- Will the construction change if softmax was used as an activation in all the layers instead of ReLU? It would be nice to know what went wrong.

---

> ### Author Response · Authors · 2025-11-21
> **Rebuttal**
>
> # Response to Reviewer Sc1b
>
> We would like to thank the reviewer for acknowledging our theoretical contribution.
>
> >  ***Weakness 1:** Since the main purpose of the work is to provide a theoretical analysis, more emphasis should be given to the results and the proof techniques in the main part of the paper. Especially a proof/technique overview along with the first part of Section C would make more sense.*
>
> **Response:** Thanks for pointing out this. We didn't include the proof overview in the original manuscript mainly due to the page limit. We now added one additional page in the revised manuscript to include the proof overview in Section 6.
>
>
> >  ***Weakness 2:** Section 3- Please define the input tokens more precisely, for e.g. we are given with (n+1) tokens where the first n tokens contain (x_i, y_i) and the last token contains x_{n+1}, we expect the final output to be present in a certain token etc. Also is \Gamma the batch size? I thought later it is used as the size of the training data.*
>
> **Response:** We use a decoding layer to output the desired element, which is the element in the $(D + 1)$-th row and $(n + 1)$-th column (explained in lines 255-256 in the original manuscript). To improve readability, we also included this explanation in Section 3 of the revised manuscript. Thank you for this detailed observation. $\Gamma$ is the number of training tasks, not the batch size.
>
>
> >  ***Weakness 3:** The constructed transformer has 5 layers, it would be helpful to give an intuitive understanding of what those layers perform.*
>
> **Response:** We explicitly construct these $5$ layers in our proof of Lemma 1, but let us summarize the results here. The $5$ layers function in the following way. The first layer copies input $x_{n+1}$ for $n$ times to the $(n+2)$-th column and up to $(2n+1)$-th column; the second layer substract each $x_i$ from $x_{n+1}$ for $i=1,\cdots,n$; the third layer computes the norm $\|x_{n+1}-x_i\|^2$ and then normalized by $h^2$; the fourth layer copies the $y_i$; finally the fifth layer combines all the previous components together and computes the kernel regression result by using softmax activation.
>
>
> >  ***Weakness 4:** The paper assumes some background on manifolds, but it would be better to explain the concepts more elaborately, especially in Sec B.1.*
>
> **Response:** We added some more background on manifolds in Section B.1. We also added the definition of feed-forward network class in Section B.2.
>
> >  ***Question 1:** Line 790: What is P_f?*
>
> **Response:** $P_f$ was a typo and is supposed to be $\rho_f$ defined in Assumption 2.
>
>
> >  ***Question 2:** What are hidden tokens in Def 3? And how are they represented in the input?*
>
> **Response:** What we meant by hidden tokens is according to the representation in Eq. (11), so the number of hidden token $\ell$ is just the number of columns in the input matrix $H$.
>
>
> >  ***Question 3:** Will the construction change if softmax was used as an activation in all the layers instead of ReLU? It would be nice to know what went wrong.*
>
> **Response:** That's a great question. The construction will change if softmax activation are used in all the layers. Softmax does not do exact trunction as ReLU, so nontrivial additional work would need to be done. Our experiments on simulated data use the same activation function specified in our construction (softmax in the last layer and ReLU in other layers).

---

### Official Review · Reviewer_pVF9 · 2025-11-11

**Soundness:** 3
**Presentation:** 4
**Contribution:** 4
**Rating:** 6
**Confidence:** 2

**Summary:**

The paper analyzes in-context learning (ICL) for regression of Hölder functions on a compact d-dimensional Riemannian manifold. It establishes a connection between the attention mechanism and kernel methods, showcasing that attention interactions can be seen as regression kernels. The authors then construct a transformer for kernel regression. Building on this construction, the authors derive a generalization bound for transformer learnability and statistical difficulty.

I would like to not that I am not an expert on this topic (especially for proof details), and will defer to other reviewers for more in-depth judgement, and put a low confidence on my review as an indication of this.

**Strengths:**

- The paper is well-written and generally easy to follow with novel contributions.
- Lemma 1, where the authors explicitly construct a 5-block, multi-head transformer that exactly equals the Gaussian Nadaraya–Watson estimator on the prompt (no approximation error), is easy to follow and clearly makes the connection between the styles of methods and regression.
-The task-/prompt-level generalization analysis is interesting and very relevant for the field. The decomposition isolates (a) learning the algorithm across tasks and (b) per-prompt nonparametric difficulty on manifolds.

**Weaknesses:**

- The assumptions for the generalization error bound in Section 5 are quite idealised and strong. For example, the assumption that $p_x$ is uniform is unrealistic, but understandable for the work. A discussion of where this assumption might hold and where this framework may break down due to these assumptions would be helpful to add to the Appendix.
- Although the paper's main contributions are the theoretical bounds and connection to kernel methods, the empirical experiments are fairly weak to show the practicality of the kernel-attention connection. As such, the paper is mainly a conceptual contribution.
- The network class of $O(Dn)$ heads and length $2n+1$ is computationally heavy for any realistic $n$ and $D$ with a larger dataset and prompt. No efficiency argument or parameter reduction is provided, and no experiments report compute or runtime scaling. Therefore, the construction serves more as a conceptual bridge than as a scalable algorithm. As such, this makes the connections to typical LLM configurations weaker, and difficult to understand when practically useful. It may be helpful to provide a compute/memory profile for the constructive transformer and show that trained practical architectures (far fewer heads) empirically approach the same behavior. Larger context lengths in Table 1 would also help to clarify this.
- Additionally, the constructed architecture is a synthetic transformer for kernel simulation, not an explanation of why actual transformers implement kernel regression. Without ablation experiments showing convergence toward this construction under realistic constraints (fewer heads, smaller width, learned parameters), the claim of “bridging attention and kernels” remains largely conceptual.

**Questions:**

- I understand that this paper differs from Kim et al (2024) in the network size and exponential rate, but why is a comparison to their baseline not provided then, if it is the closest paper in the field for the empirical results?
- Why do the p-values have very large standard deviations in Table 1?

---

> ### Author Response · Authors · 2025-11-21
> **Rebuttal**
>
> # Response to Reviewer pVF9
>
> We would like to thank the reviewer for acknowledging our novel contribution.
>
> >  ***Weakness 1:** About the assumption that $\rho_x$ is uniform.*
>
> **Response:** We use a uniform distribution over a compact domain as a standard, tractable model to control covering numbers and concentration without additional density parameters. This assumption is often reasonable for synthetic benchmarks and simulation studies. The uniform assumption can be extended to distributions which are absolutely continuous with respect to the uniform distribution with the corresponding Radon-Nikodym derivative upper and lower bounded.
>
>
>
> >  ***Weakness 2:** About the empirical experiments and the the practicality of the kernel-attention connection.*
>
> **Response:** We have experimented on both the synthetic data and real language data. For synthetic data, we demonstrated the similarity between the learned attention score and the kernel function. For language data, we demonstrated the kernel shape of the attention score in Figure 3, although there is no ground truth to compare with.
>
>
>
> >  ***Weakness 3:** The network class of $O(DN)$ heads and length $2n+1$ is computationally heavy...*
>
> **Response:** We have token length of $2n+1$ because the context length is $n$. Our network has a small depth (5 layers) while each layer is wide. In terms of depth, our network is close to the practical transformer architecture (GPT2-small only has 12 layers). The width of $m=O(nD)$ provides an upper bound on the number of attention heads. The focus of this paper is not to provide a theoretical guarantee on the optimal number of parameters as a practical guide. In the experiment for verifying the correlation between attention scores and kernel (Figure 1 and 2), we used $m=1$ attention head and $L_{\rm T}=5$ transformer layers. In the experiment for verifying the generalization error bound curve (Figure 4), we used $m=2,4,8$ for in-context length $n=16,64,256$ respectively, also $L_{\rm T}=5$ layers in total. Those parameters are provided in appendix F.1 in the orginal manuscript.
>
>
>
> >  ***Weakness 4:** Additionally, the constructed architecture is a synthetic transformer for kernel simulation...*
>
> **Response:** The main goal of this paper is to theoretically establish the connection between the attention mechanism and kernel methods. We do have an experiment to show the kernel shape of the attention score on real language data (in Figure 3). In terms of depth, our synthetic transformer has $5$ layers, which is close to the practical model such as GPT2-small. In terms of width, our theory provides an upper bound $m=O(Dn)$, which may not be optimal at this point. In terms of the learned parameters, our generalization theorem (Theorem 1) applies to the empirical risk minimizer where the parameters are learned through optimization.
>
>
>
>
> >  ***Question 1:** I understand that this paper differs from Kim et al (2024)...*
>
> **Response:** We appreciate the reviewer for this question. We discussed the differences about our model with the one in Kim et al (2024) in the related work (section 6), but let us elaborate more here. Kim et al (2024) uses a transformer model which consists of one linear attention component plus a deep feed-forward component, so their model's expressivity mostly lies in the feed-forward component, while our model's expressivity mostly lies in the attention component. The depth of their feed-forward component is roughly $O(\log N)$ (from their results in Lemma 4.4), and the width of their feed-forward component is roughly $O(DN)$ (from their results in Lemma C.5 and Lemma 4.4), where $N$ is the number of B-spline basis they constructed. In Theorem 4.5, $N$ is chosen in the order of $n^{\frac{d}{2\alpha+d}}$. As a result, the network in Kim et al (2024) has width $Dn^{\frac{d}{2\alpha+d}}$ and depth $\log n$. In constrast, our transformer model has width $O(Dn)$ and the depth $5$.
>
> For the exponential rate, we both achieve the minimax optimal rate $O(n^{-\frac{2\alpha}{2\alpha+d}})$ up to log factors when the task size $\Gamma$ is large; However, their $d$ is the dimension of the ambient Euclidean space, while our $d$ is the intrinsic dimension of the underlying data manifold which might be embedded in a high dimension ambient space.
>
>
>
> >  ***Question 2:** Why do the p-values have very large standard deviations in Table 1?*
>
> **Response:** This is mainly due to the large number of functions in our task (function) space, each with its own local variation. In the experiment associated with Table 1, our task coefficients are sampled from $[0,1]$, and data are sampled from $[0,\pi]$ and $[0,2\pi]$. When the model is trained on data with variations, the variance can be high.

---

### Meta-Review · Area_Chair_5KNY · 2026-01-07

**Summary:**

This paper studies the theoretical properties of in-context learning (ICL) when applied to data residing on structured manifolds with a low intrinsic dimension. In particular, the authors show that transformer based ICL can perform kernel estimator with zero approximation error. The authors then prove a generalization error for transformer based ICL in terms of the prompt length, number of training tasks, and the intrinsic manifold dimension (instead of the ambient dimension). Synthetic experiments are provided to validate the theoretical results

Reviewers all appreciate the importance of the topic under study and the theoretical results presented in the paper. Despite some limitations (e.g. the generalization error bound can depend superexponentially on the intrinsic dimension), the results should be of interest to the community.

**Reviewer Concerns:**

A major concern made by Reviewer iBpf is on the constants $C_1$ and $C_2$ in the generalization error, which depend on the intrinsic dimension $d$. In the rebuttal, the authors reveal that the dependence of $C_2$ on $d$ could be $d^{d/2}$ in the worst case scenario, i.e. superexponential. This is thus a limitation of the current analysis.

**Reviewer Scores:**

The scores are 6,8,6,4

Reviewer iBpf (4) would likley have raised their score.

---

### Decision · Program_Chairs · 2026-01-26

Accept (Poster)